# Rethinking and Extending the Probabilistic Inference Capacity of GNNs

**Tuo Xu**
Wangxuan Institute of Computer Technology
Peking University
Beijing, China
`doujzc@stu.pku.edu.cn`

**Lei Zou**
Wangxuan Institute of Computer Technology
Peking University
Beijing, China
`zoulei@pku.edu.cn`

## Abstract

Designing expressive Graph neural networks (GNNs) is an important topic in graph machine learning fields. Despite the existence of numerous approaches proposed to enhance GNNs based on Weisfeiler-Lehman (WL) tests, what GNNs *can and cannot* learn still lacks a deeper understanding. This paper adopts a fundamentally different approach to examine the expressive power of GNNs from a probabilistic perspective. By establishing connections between GNNs' predictions and the central inference problems of probabilistic graphical models (PGMs), we can analyze previous GNN variants with a novel hierarchical framework and gain new insights into their node-level and link-level behaviors. Additionally, we introduce novel methods that can provably enhance GNNs' ability to capture complex dependencies and make complex predictions. Experiments on both synthetic and real-world datasets demonstrate the effectiveness of our approaches.

## 1 Introduction

Graph neural networks (GNNs) are the dominant approaches for learning graph-structured data, among which message passing neural networks (MPNNs) are the most promising variants, demonstrating remarkable success across various domains. The development of MPNNs can be classified into several distinct theoretical motivations. From the perspective of graph signal processing, MPNNs were proposed as a generalization of convolutions on non-Euclidean graph domains (Bruna et al., 2013). Also, MPNNs have been motivated by their connection to the Weisfeiler-Lehman (WL) graph isomorphism tests (Hamilton et al., 2017). At the same time, MPNNs were proposed by parameterizing mean-field inference in probabilistic graphical models (PGMs) (Dai et al., 2016). Although GNNs have been well studied, motivated and improved in terms of graph convolutions and WL tests, few works study GNN's probabilistic inference capacity. Many works (Dai et al., 2016; Satorras & Welling, 2021; Qu et al., 2022) implicitly assume that GNNs themselves fail to capture the complex joint dependencies among nodes, and consequently these works attempt to integrate GNNs with graphical models, where predictions are then obtained by variational inference algorithms such as loopy belief propagation (Murphy et al., 1999). The underlying belief behind these GNN-PGM models is that *leveraging GNNs to parameterize graphical models can enhance GNN's probabilistic inference capacity with belief propagation.* However, in this work, we contend that despite the empirical success of these methods, GNNs themselves in fact possess substantial approximation capabilities for posterior distributions in various graphical models.

In this study, our objective is to gain deeper understandings of the expressive power of GNNs in terms of approximate inference in graphical models. Unlike the $k$-WL hierarchy, graphical models provide a more intuitive framework for interpreting graph data and evaluating GNNs. Precisely, we propose an alternative expressive power hierarchy by introducing a family of Markov random fields (MRFs) with increasingly complex distribution and inference targets. Based on these metrics, we provide novel insights into various GNN variants including MPNNs, higher-order GNNs (Morris et al., 2018), subgraph GNNs (Bevilacqua et al., 2021), labeling trick (Zhang et al., 2020), etc. within the contexts of node classification and link prediction. Our finding confirm the progressively

increasing ability of existing GNN variants in terms of capturing higher-order dependencies, while also providing a novel perspective on their link prediction capacities.

In the second part of this study, we attempt to design a systematic and efficient framework for extending the capabilities of GNNs in modeling complex distributions and inference targets. By rethinking the inherent connection between 1-WL (MPNNs) and the Bethe approximation on pairwise Markov Random Fields (MRFs), we propose two methods, namely *phantom nodes* and *phantom edges*, each targeting at one of the problems. We then provide a formal analysis of the expressive power of these approaches. The primary advantage of our proposed methods, in contrast to other variants like higher-order GNNs, lies in their simplicity and efficiency, which can be easily implemented as graph preprocessing. Empirically, we demonstrate that our framework significantly enhances the capabilities of MPNNs in capturing complex distributions and inference targets, and also improve the performance of MPNNs on various real-world node classification and link prediction tasks.

## 2 BACKGROUND

We use $\{\}$ to denote sets and use $\{\{\}\}$ to denote multisets. The index set is denoted as $[n] = \{1, ..., n\}$. We consider node-attributed undirected graphs $G = (\boldsymbol{A}, \boldsymbol{X})$, where $\boldsymbol{A} \in \mathbb{R}^{n \times n}$ is the adjacency matrix representing the set of edges $\mathcal{E}_G$ over the nodes of $G$ indexed by $\mathcal{V}_G = [n]$, and $A_{i,j} = \mathbf{1}_{(i,j) \in \mathcal{E}_G}$. $\boldsymbol{X} \in \mathbb{R}^{n \times d}$ is the collection of node features, with its $i$-th entrance denoted as $\boldsymbol{x}_i \in \mathbb{R}^d$ corresponding to the feature of the node $i$. We use $\mathcal{N}_G(i)$ to denote the set of neighbors of the node $i$ in $G$. A clique $C$ is a fully-connect subset of nodes $C = \{i_1, ..., i_k\}$ and we refer to $k$ the order of $C$. A permutation $\pi$ is a bijective mapping $[n] \rightarrow [n]$. For convenience $\pi$ can act on graphs, adjacency matrices, node features and indices as usual by $\pi(G) = (\pi(\boldsymbol{A}), \pi(\boldsymbol{X}))$ with $\pi(\boldsymbol{A})_{\pi(i),\pi(j)} = \boldsymbol{A}_{i,j}, \pi(\boldsymbol{X})_{\pi(i),:} = \boldsymbol{X}_{i,:}. \pi(\{i_1, ..., i_k\}) = \{\pi(i_1), ..., \pi(i_k)\}$.

Given graphs of $n$ nodes we consider a set of random variables denoted as $\{\mathbf{x}_1, \mathbf{z}_1, ..., \mathbf{x}_n, \mathbf{z}_n\}$ and use $\boldsymbol{x}_1, \boldsymbol{z}_1, ..$ to denote the realization of random variables or non-probabilistic variables. To simplify notations, given a subsets of indices $A$ we let $\mathbf{x}_A$ to stand for $\{\mathbf{x}_i \mid i \in A\}$. For example, if $A = \{2, 4\}$ and $B = \{3\}$, we represent $p(\mathbf{x}_2 = \boldsymbol{x}_2, \mathbf{x}_4 = \boldsymbol{x}_4 \mid \mathbf{x}_3 = \boldsymbol{x}_3)$ as $p(\mathbf{x}_A = \boldsymbol{x}_A \mid \mathbf{x}_B = \boldsymbol{x}_B)$ and more compactly $p(\mathbf{x}_A \mid \mathbf{x}_B)$. We also will often use $\mathbf{X}$ to stand for $\{\mathbf{x}_1, ..., \mathbf{x}_n\}$ and $\mathbf{Z}$ for $\{\mathbf{z}_1, ..., \mathbf{z}_n\}$.

**Graph isomorphism and Weisfeiler-Lehman tests.** Two graphs $G = (\boldsymbol{A}_G, \boldsymbol{X}_G)$ and $H = (\boldsymbol{A}_H, \boldsymbol{X}_H)$ are isomorphic, denoted as $G \simeq H$, if they both have $n$ nodes and there exists a permutation $\pi : [n] \rightarrow [n]$ satisfying $(\boldsymbol{A}_G)_{i,j} = (\boldsymbol{A}_H)_{\pi(i),\pi(j)}$ and $(\boldsymbol{X}_G)_i = (\boldsymbol{X}_H)_{\pi(i)}$ for all $i, j \in [n]$. Such $\pi$ is an isomorphism. Weisfeiler-Lehman (WL) tests are a family of necessary tests for graph isomorphism. Apart from some corner cases (Cai et al., 1992), they are effective and computationally efficient tests for graph isomorphism. Its 1-dimensional variant is analogous to MPNNs, which iteratively aggregates the colors of nodes and their neighborhoods and then injectively hashes them into new colors. The algorithms decides two graphs non-isomorphic if the colors of two graphs are different. A detailed description of WL tests is in Appendix B. Due to space limits we leave a more detailed discussion of previous works in Appendix A.

## 3 A PROBABILISTIC INTERPRETATION OF GRAPHS

In this section we describe graphical models for graph data as well as highlighting the key problems of inference in graphical models. Given a graph $G = (\boldsymbol{A}, \boldsymbol{X})$ with $n$ nodes, we hypothesise each node $i$ is completely described by an (unknown) latent random variable $\mathbf{z}_i \in \mathcal{Z}$ and denote by $\mathbf{Z} = (\mathbf{z}_1, ..., \mathbf{z}_n) \in \mathcal{Z}^n$ the collection of the latent variables. For simplicity we may assume $\mathcal{Z}$ is a discrete space, but our results are also applicable to continuous $\mathcal{Z}$ as long as we generalize GNNs to output Hilbert space embeddings of distributions (Smola et al., 2007; Dai et al., 2016). $\boldsymbol{A}$ specifies the *conditional independence structure* of $\mathbf{Z}$: $\mathbf{z}_A \perp\!\!\!\perp \mathbf{z}_B \mid \mathbf{z}_C$ whenever there is no path from a node in $A$ to a node in $B$ which does not pass through a node in $C$. The node features are generated by these latent variables: for a node $i$, given its latent variable $\mathbf{z}_i$, the corresponding observed node feature $\boldsymbol{x}_i$ is sampled from $\mathbf{z}_i$: $p_{nf}(\mathbf{x}_i \mid \mathbf{z}_i)$. The graph learning target is, given observed node features $\mathbf{X}$, to infer the latent variables $\{\mathbf{z}_i\}$, which can be further used for tasks such as:

- Node classification: node label $\mathbf{y}_i$ is obtained by $p(\mathbf{y}_i \mid \mathbf{X}) = \sum_{\mathbf{z}_i \in \mathcal{Z}} p(\mathbf{y}_i \mid \mathbf{z}_i) p(\mathbf{z}_i \mid \mathbf{X})$.
- Link prediction: it can be naturally interpreted as asking "given that node $i$ is the source node, what is the possibility of $j$ being the target node?". Thus the link label $\mathbf{y}_{ij}$ for node pair $(i, j)$ is obtained from both $\mathbf{z}_i$ and $\mathbf{z}_j$: $p(\mathbf{y}_{ij} \mid \mathbf{X}) = \sum_{\mathbf{z}_i, \mathbf{z}_j \in \mathcal{Z}} p(\mathbf{y}_{ij} \mid \mathbf{z}_i, \mathbf{z}_j) p(\mathbf{z}_i, \mathbf{z}_j \mid \mathbf{X})$.

Obviously, once we obtained the marginals $p(\mathbf{z}_i \mid \mathbf{X})$ and more generally $p(\mathbf{z}_i, \mathbf{z}_j \mid \mathbf{X})$, the labels can be easily inferred. Thus the key problem lies in inferring marginals of the joint posteriori $p(\mathbf{Z} \mid \mathbf{X})$. Our target, therefore, is to investigate *whether and to what extend are GNNs expressive enough for inferring the marginals of the graphical models $p(\mathbf{Z} \mid \mathbf{X})$*. We now introduce details of our framework for investigating GNNs.

## 3.1 THE PROBABILISTIC FORMULATION

We first formally define the joint distribution we consider. Clifford & Hammersley (1971) states that a positive distribution satisfies the above conditional independence structure if and only if it can be represented by

$$p(\mathbf{X}, \mathbf{Z}) = \frac{1}{Z} \prod_{i \in \mathcal{V}} \Psi_i(\mathbf{z}_i, \mathbf{x}_i) \prod_{C \in \mathcal{C}} \Psi_C(\mathbf{z}_C), \tag{1}$$

where $\Psi_i, \Psi_C$ are potential functions, $\mathcal{C}$ is the set of cliques [1] in $G$, and $Z$ is the partition function $Z = \sum_{\mathbf{X}, \mathbf{Z}} \prod_{i \in \mathcal{V}} \Phi_i(\mathbf{z}_i, \mathbf{x}_i) \prod_{C \in \mathcal{C}} \Psi_C(\mathbf{z}_C)$ (assuming discrete node features). A potential function $\Psi_C(\mathbf{z}_C)$ is a nonnegative, real-valued function on the possible realizations of $\mathbf{z}_C$. Such a family of distributions is called Markov random fields (MRFs). Albeit powerful in their expressiveness, the formulation in Eq. 1 does not follow general invariant and equivariant assumptions in the graph data. First, each data instance in graphical models are assumed to share the same structure, while in graph machine learning fields each instance corresponds to different graphs with possibly different structures. Moreover, Eq. 1 might specifies different probabilistic distributions for two isomorphic graphs. To this end, we first define a family of MRFs that are applicable to different graph structures and also *invariant* to permutations.

**Lemma 1.** *Suppose there is an algorithm $\mathcal{F}$, given any graph $G = (\boldsymbol{A}, \boldsymbol{X})$, $\mathcal{F}$ maps the independence structure $\boldsymbol{A}$ to a MRF $p_G = \mathcal{F}(\boldsymbol{A})$. Then, $\mathcal{F}$ satisfies that for any permutation $\pi : [n] \to [n]$ and $\boldsymbol{A}$:*

$$p_{\pi(G)}(\pi(\mathbf{X}), \pi(\mathbf{Z})) = p_G(\mathbf{X}, \mathbf{Z}),$$

*if and only if $p_G$ can be represented in the form 1 with potentials given by $\Psi_C = \mathcal{A}(C, \boldsymbol{A})$, $\Psi_i = \mathcal{A}(i, \boldsymbol{A})$ where $\mathcal{A}$ is a permutation-invariant function that maps cliques (nodes) of graphs to potential functions:*

$$\mathcal{A}(C, \boldsymbol{A}) = \mathcal{A}(\pi(C), \pi(\boldsymbol{A})) \quad \forall C \in \mathcal{C} \cup \mathcal{V}, \boldsymbol{A} \in \{0, 1\}^{n \times n}, \text{ and } \pi : [n] \to [n].$$

Lemma 1 precisely describes the space of invariant MRFs: we can fully define such MRFs by specifying permutation-invariant $\mathcal{A}$. This enables us to further discuss the general relation between GNNs and graphical models. In reality, many works choose to parameterize the potential functions as above, with $\mathcal{A}$ being implemented as a constant function (Dai et al., 2016) or a GNN (Qu et al., 2022).

## 3.2 METRICS FOR EVALUATING GNNS

With the properly defined families of MRFs we now discuss our framework for evaluating the expressive power of GNNs.

**Complexity of distributions.** From the above discussions the complexity of the distribution is fully governed by $\mathcal{A}$. We consider two metrics for evaluating the complexity of distributions: the discriminating power of $\mathcal{A}$ and the order of the MRF. We will measure the discriminating power by the WL hierarchy, and define $\mathcal{A}$ is $k$-WL distinguishable if and only if for any cliques $C_1, C_2$ from two graphs $G = (\boldsymbol{A}_G, \boldsymbol{X}_G), H = (\boldsymbol{A}_H, \boldsymbol{X}_H), \mathcal{A}(C_1, \boldsymbol{A}_G) \neq \mathcal{A}(C_2, \boldsymbol{A}_H)$ only when $k$-WL

---

[1] We do not restrict $\mathcal{C}$ to be maximal cliques

distinguishes $C_1$ and $C_2$ (we assume $k$-WL does not input node features here; see Appendix B for detailed descriptions)[2]. The order of a MRF refers to the maximum order of cliques $\mathcal{C}$ we consider in Eq. 1. For example, in pairwise MRFs $\mathcal{C}$ is the set of cliques with orders no more than 2 (i.e., edges). Formally, a $k$-order MRF is defined by $\mathcal{A}$ that satisfies $\mathcal{A}(C, G) \equiv \Phi$ for all $|C| > k$ where $\Phi(\cdot) \equiv constant$. It is able to describe $k$-order dependencies over $k$-tuples of nodes.

**Target posteriori.** As previously discussed, we are interested in the inference problems of estimating posteriori including marginal distributions $p(\mathbf{z}_i \mid \mathbf{X})$ for node-level tasks and the joint distribution over parts of the variables $p(\mathbf{z}_i, \mathbf{z}_j \mid \mathbf{X})$ for link-level tasks. Note that since we can factorize the posteriori as

$$p(\mathbf{z}_i, \mathbf{z}_j \mid \mathbf{X}) = p(\mathbf{z}_i \mid \mathbf{z}_j, \mathbf{X})p(\mathbf{z}_j \mid \mathbf{X}),$$

the posteriori can also be estimated by repeatedly applying inference algorithms on the conditional marginals.

**Methods for inference.** Given the recent developments in GNNs, one may expect that GNNs might be able to perform *exact inference* in MRFs. However, our next theorem states that the exact inference of even a simple pair-wise MRF with merely 1-WL distinguishable potentials requires the ability of distinguishing all non-isomorphic graphs.

**Theorem 2.** *Given any connected graphs $G = (\boldsymbol{A}_G, \boldsymbol{X}_G), H = (\boldsymbol{A}_H, \boldsymbol{X}_H)$ and $i \in \mathcal{V}_G, j \in \mathcal{V}_H$. If $i$ and $j$ are not isomorphic, then there exists a 1-WL distinguishable 2-order $\mathcal{A}$ such that the pair-wise MRFs $p_G, p_H$ specified by $\mathcal{A}$ on $G$ and $H$ satisfy:*

$$p_G(\mathbf{z}_i = \boldsymbol{z} \mid \mathbf{X} = \boldsymbol{X}_G) \neq p_H(\mathbf{z}_j = \boldsymbol{z} \mid \mathbf{X} = \boldsymbol{X}_H).$$

Therefore, current GNNs fail to learn the exact inference on general graphs. This is not surprising as the exact inference on graphs with loops are often exponentially expensive, and we often resort to variational inference for estimating the marginals. Given the original intractable distribution $p$, variational inference aims to find a tractable distribution $q$ to approximate $p$: $\min_q D_{\mathrm{KL}}(q\|p)$, where $D_{\mathrm{KL}}$ is the Kullback-Leibler (Kullback & Leibler, 1951) divergence. In this paper we will focus on the successful approximate inference method belief propagation, which is also known as the *Bethe approximation* (Murphy et al., 1999; Yedidia et al., 2001). Bethe approximation on pairwise MRFs assumes a quasi distribution $q(\mathbf{Z}) = \prod_{i \in \mathcal{V}} q_i(\mathbf{z}_i) \prod_{(i,j) \in \mathcal{E}} \frac{q_{ij}(\mathbf{z}_i, \mathbf{z}_j)}{q_i(\mathbf{z}_i)q_j(\mathbf{z}_j)}$ thus considering the dependencies over nodes. (Yedidia et al., 2001). A successful algorithm for minimizing the Bethe free energy is the loopy belief propagation (Heskes, 2002b), which is a fixed iteration over messages of edges.

**Problem setup.** Our analysis aims to answer the question: *to what extent can GNNs approximate the inference of graphical models?* We tackle the problem by investigating GNN variants via the following criteria as discussed above:

- How complex the joint distribution can be? This includes the discriminating power and the maximum order of $\mathcal{A}$.
- What posteriori can GNNs approximate?
- If GNNs cannot perform exact inference over MRFs, how approximate can their predictions be?

We believe answering these questions would print a clearer picture about what GNNs can learn from a more intuitive, probabilistic perspective.

## 4 ANALYSING THE PROBABILISTIC INFERENCE CAPACITY OF GNNS

### 4.1 ON MPNNS AND PAIRWISE MRFS

Our first results aim to characterize the expressive power of MPNNs in terms of modeling probabilistic distributions over MRFs, as well as introducing our strategy for evaluating GNN models.

---

[2]We choose the WL hierarchy due to its prevalence in the GNN literature; other metrics are certainly acceptable. Note that MRFs are usually powerful even with less expressive $\mathcal{A}$: in reality, $\mathcal{A}$ is usually implemented simply producing identical potentials. Fully probability-based results without the utilization of WL tests is provided in Appendix K.2.

MPNNs are generally applied to learn node representations and are bounded by 1-WL which aggregates information from neighbors, thus in this section we consider pairwise MRFs defined by 1-WL distinguishable, 2-order $\mathcal{A}$.

We first discuss the equivalence between them in terms of discriminating power.

**Theorem 3.** *Given any graphs $G = (\boldsymbol{A}_G, \boldsymbol{X}_G), H = (\boldsymbol{A}_H, \boldsymbol{X}_H)$ and any 1-WL distinguishable 2-order $\mathcal{A}$. If at some iteration, the collection of the messages of parallel belief propagation on $G$ and $H$ are different, then 1-WL also distinguishes $G, H$.*

Surprisingly, although the message passing procedure in belief propagation operates on directed edges, we have shown that it is equivalent to 1-WL in terms of distinguishing power. Continuing from Theorem 3, it's obvious that MPNNs can also approximate the posteriori. In fact, it turns out that the correspondence reveals stronger facts about MPNNs.

**Theorem 4.** *MPNNs can learn marginals that are at least as accurate as belief propagation. Formally, there exists decoding functions $f, g$ such that given any 1-WL distinguishable 2-order $\mathcal{A}$, for arbitrary $\epsilon > 0$ and $n \in \mathbb{N}$, there exists a MPNN such that for any graphs with no more than $n$ nodes, such that:*

$$D_{\mathrm{KL}}(f(\boldsymbol{h}_i) \mid q_i) \leq \epsilon \text{ for } i \in \mathcal{V} \text{ and } D_{\mathrm{KL}}(g(\boldsymbol{h}_i, \boldsymbol{h}_j) \mid q_{ij}) \leq \epsilon \text{ for } (i, j) \in \mathcal{E},$$

*where $q_i, q_{ij}$ are node and edge marginals specified by a local extrema of Bethe approximation, $\boldsymbol{h}_i$ is the representation of node $i$ obtained by the MPNN, $D_{\mathrm{KL}}$ is the Kullback-Leibler divergence.*

Theorem 4 directly indicates that MPNNs are not only capable of approximating node marginals of Bethe approximation, but also *edge marginals*. In other words, if two nodes are connected, MPNNs are suitable for learning the joint representation of these two nodes! This implication will be exploited in Section 5 where we extend MPNNs for link prediction in a novel and simple manner.

Putting these together, we conclude the section by stating that MPNNs are able to perform exact inference on trees. This result naturally aligns with the fact that MPNNs, which shares the equivalent expressiveness with the 1-WL test, can exactly capture tree patterns Zhang et al. (2024).

**Corollary 5.** *Given any graph $G$ and 1-WL distinguishable 2-order $\mathcal{A}$, if $G$ is a tree, there is a MPNN that outputs true node and edge marginals of the MRF defined by $\mathcal{A}$.*

## 4.2 ANALYSING EXISTING GNN VARIANTS

In this section we switch to more complex and powerful GNN variants. From previous discussions, it is evident that current GNNs are likely capable of approximating variational methods including naive mean field (as shown by Dai et al. (2016)) and Bethe approximation. Thus, our analysis focuses on the remaining two metrics in Section 3, i.e. the complexities of the joint distribution and the target posteriori. For ease of discussion we summarize our metrics for evaluating the expressive power of GNNs as follows.

**Definition 6.** A class of GNN models can $k$-$l$ approximate some posteriori $p$ if and only if given arbitrary $k$-WL distinguishable $\mathcal{A}$ with maximum order being $l$, it satisfies:

- It can distinguish all graphs distinguished by iterations of belief propagation as in Theorem 3.
- It can provide marginals at least as accurate as Bethe approximation as in Theorem 4.

Therefore, $k$ and $l$ are corresponded to the complexity of the joint distribution and $p$ is corresponded to the target posteriori. For example, the results about MPNNs in Section 4.1 can be abbreviated as *MPNNs can 1-2 approximate $p(\mathbf{z}_i \mid \mathbf{X})$*. We can also derive upper bounds for MPNNs as follows.

**Theorem 7.** *MPNNs can at most 1-2 approximate $p(\mathbf{z}_i \mid \mathbf{X})$ for arbitrary $G$ and $i \in \mathcal{V}_G$.*

Similarly with our metrics, we notice that existing GNN variants also extend MPNNs mainly from two motivations: one to increase the graph-level expressive power and another to solve more complex tasks such as link prediction. Here we also discuss these variants separately according to their motivations.

### 4.2.1 GNNs that Focus on Expressive Power

In this section we investigate GNN variants that focus on increasing the expressive power beyond 1-WL. We shall see that most of them sufficiently improve expressiveness by approximating *more complex distributions*.

$k$**-GNNs.** The first GNN variants we consider are know as the higher-order GNNs, which directly corresponds to the $k$-WL hierarchy (Morris et al., 2018; Keriven & Peyré, 2019; Geerts & Reutter, 2022; Maron et al., 2019a). We will focus on $k$-GNNs that are bounded the by $k$-WL test. Note that $k$-GNNs generally computes the representation of a single node $j$ as the representation of $(j, j, ..., j)$.

**Proposition 8.** *$k$-GNNs can k-k approximate $p(\mathbf{z}_i \mid \mathbf{X})$ for arbitrary G.*

*Remark.* Unsurprisingly, $k$-GNNs provably capture $k$-order dependencies. At initialization, $k$-GNNs inject structural information of node tuples into their representations; At each layer, $k$-GNNs directly pass messages among $k$-tuples which share $k-1$ common nodes. This helps them to simulate the messages between factors and nodes in belief propagation.

**Subgraph GNNs.** Bevilacqua et al. (2021) proposed a new type of GNNs namely Equivariant Subgraph Aggregation Networks (ESANs), which provides a novel variant of GNNs namely subgraph GNNs. We consider its node-based variant, which are also studied in Frasca et al. (2022), showing a 3-WL *upper bound* for its expressiveness. Yet, this result is a limitation of ESANs and it's unclear how much the gap is between ESANs and 3-WL. Zhang et al. (2023a) showed that ESANs are equivalent to 3-WL in the very specific graph biconnectivity tasks.

**Proposition 9.** *ESANs with node marking policy can 1-3 approximate $p(\mathbf{z}_i \mid \mathbf{X})$ for arbitrary G.*

*Remark.* Here, we take a step forward and show that ESANs are equivalent to 3-WL in capturing higher-order dependencies. In fact, each subgraph with marked node $i$ can capture $i$'s adjacent 3-cliques, and by aggregating all subgraphs together at each layer, ESANs are able to capture all 3-cliques.

**GNNs with lifting transformations.** Bodnar et al. (2021b;a) designed new WL variants to consider graph structures such as cliques and cycles. We summarize their expressiveness as follows.

**Proposition 10.** *The GNN variants corresponding with SWL / CWL with k-clique simplex can 1-k approximate $p(\mathbf{z}_i \mid \mathbf{X})$.*

*Remark.* Similar with $k$-GNNs, this variant successfully exploits arbitrary order dependencies. With the specifically designed message passing paradigm, cliques can now send and receive messages as an ensemble. We believe this is the key to model complex distributions, and in Section 5 we will develop a novel framework to improve MPNNs' probabilistic inference capacity in a similar but simpler and more efficient manner.

**Other variants.** We notice that there are also other variants including ID-GNNs (You et al., 2021), Nested GNNs (Zhang & Li, 2021), etc. However, we summarize that they cannot improve MPNNs' probabilistic inference capacity.

**Proposition 11.** *The above GNNs can at most 1-2 approximate $p(\mathbf{z}_i \mid \mathbf{X})$.*

### 4.2.2 GNNs that Focus on Complex Prediction Tasks

We now list GNN variants that focus on *link prediction* and more generally joint prediction of multi-nodes. In this section we shall see that they sufficiently approximate *more complex posteriori*.

$k$**-GNNs.** Since $k$-GNNs learn representations for $k$-node tuples, it's natural to expect them being capable of approximating joint posteriori of $k$ nodes.

**Proposition 12.** *$k$-GNNs cannot 1-2 approximate $p(\mathbf{z}_{i_1}, ..., \mathbf{z}_{i_k} \mid \mathbf{X})$ for arbitrary $k \geq 2$.*

*Remark.* Surprisingly, our result states that $k$-GNNs are not capable of approximating $k$-posteriori even for simple pairwise MRFs. This explains why few works apply $k$-GNNs for link prediction, although 2-GNNs naturally produces node pair representations. Nevertheless, we notice that there is another line of expressive GNNs inspired by $k$-FWL, among which Edge Transformers (Bergen et al., 2021b) are 2-FWL-MPNNs designed for link prediction. Our next theorem verifies the effectiveness of these variants.

**Proposition 13.** *$k$-FWL-MPNNs can 1-2 approximate $p(\mathbf{z}_{i_1}, ..., \mathbf{z}_{i_k} \mid \mathbf{X})$.*

**Labeling trick.** To extend MPNNs for link prediction, SEAL (Zhang & Chen, 2018) adds labels to nodes to tag the target nodes equivalently and differently from the rest of the nodes. This technique was summarized and generalized in Zhang et al. (2020) known as labeling trick.

**Proposition 14.** *$k$-labeling trick MPNNs 1-2 approximate $f(\mathbf{z}) = p(\mathbf{z}_{i_1} = \cdots \mathbf{z}_{i_k} = \mathbf{z} \mid \mathbf{X})$ but not $p(\mathbf{z}_{i_1}, \mathbf{z}_{i_2} \mid \mathbf{X})$.*

*Remark.* The target posteriori might seem strange since it indicates all target nodes are indistinguishable from each other, which means that SEAL cannot learn representations of *ordered node tuples*: it cannot distinguish the target link $(i, j)$ between $(j, i)$. This aligns with the fact that SEAL is only applied on undirected link prediction problems.

**Ordered node pair labeling.** Similar to SEAL, GraIL (Teru et al., 2019), INDIGO (Liu et al., 2021), etc. add labels to target node pairs but now source and target nodes are labeled differently.

**Proposition 15.** *MPNNs with ordered node pair labeling 1-2 approximate $p(\mathbf{z}_i, \mathbf{z}_j \mid \mathbf{X})$.*

*Remark.* With different labels on the source and target nodes, these methods are able to learn representations of directed node pairs. This aligns with the fact that these methods are designed for predicting *directed links* in heterogeneous knowledge graphs.

**Source node labeling.** Another line of research (You et al., 2021; Zhu et al., 2021) also uses node labels for link prediction, but they only tag the source node, yielding a more efficient framework compared with labeling trick. Here we show that they are also able to learn node pair representations.

**Proposition 16.** *Source node labeling MPNNs 1-2 approximate $p(\mathbf{z}_i, \mathbf{z}_j \mid \mathbf{X})$.*

## 5 EXTENDING MPNNs FOR MODELING MORE COMPLEX DISTRIBUTIONS

After investigating previous GNN architectures, in this section we would like to study whether we can improve GNNs' expressive power for modeling MRFs as well as being more efficient compared with previous approaches. Formally, we also focus on the two targets: modeling more complex distribution and inferring more complex posteriori. We design novel methods namely *phantom nodes* and *phantom edges*, for provably lifting GNNs' expressive power of approximating complex distributions and complex posteriori respectively. These two approaches can be easily implemented as a preprocessing procedure of the input graphs.

### 5.1 PHANTOM NODES FOR MODELING HIGHER-ORDER DEPENDENCIES

We discuss extending MPNNs for learning more complex distributions. As previously discussed, the aggregation functions of MPNNs help them to be aware of edges, which naturally corresponds to *2-cliques* in MRFs. Intuitively, capturing higher-order cliques requires the aggregation function to pass messages within higher-order cliques of nodes, which calls for different network architectures. To avoid such inconvenience, inspired by belief propagation on factor graphs we add a phantom node $v_C$ for every maximum clique $C$ in $G$, and connect $v_C$ to all nodes in $C$. The phantom nodes then serve as midway of the cliques that stores the messages among cliques of nodes. In fact, we further tag all phantom nodes with an invented label $\hat{l}$ that distinguishes them from the ordinary nodes. By applying MPNNs on the augmented graph. We have the following result:

**Proposition 17.** *MPNNs with phantom nodes can 1-∞ approximate $p(\mathbf{z}_i \mid \mathbf{X})$.*

**Relaxation.** Sometimes it is unpractical to find maximum cliques. We can relax our method by only finding cliques with *no more than $k$ nodes* as an approximation to the original approach.

**Proposition 18.** *MPNNs with phantom nodes of cliques no more than $k$ can 1-$k$ approximate $p(\mathbf{z}_i \mid \mathbf{X})$.*

### 5.2 PHANTOM EDGES FOR MODELING JOINT POSTERIORI

We discuss learning joint posteriori $p(\mathbf{z}_i, \mathbf{z}_j \mid \mathbf{X})$ for node pairs $(i, j)$. Different from the previously discussed node labeling hierarchy, we provide an alternate method for link prediction inspired by

Table 1: Results on node classification. **Best results** of each category are bold. Results with * are taken from Qu et al. (2022). PN stands for phantom nodes.

| Algorithm | PPI-1 | | PPI-2 | | PPI-10 | |
|---|---|---|---|---|---|---|
| | Accuracy | Micro-F1 | Accuracy | Micro-F1 | Accuracy | Micro-F1 |
| GCN | 76.24±0.10 | 54.55±0.29 | 76.82±0.13 | 56.10±0.36 | 80.43±0.10 | 62.48±0.27 |
| +CRF* | 76.33±0.21 | 50.79±0.74 | 76.27±0.10 | 49.47±0.63 | 77.08±0.07 | 52.36±0.72 |
| +SPN* | 77.07±0.05 | 54.15±0.17 | 78.02±0.05 | 55.73±0.15 | 80.59±0.04 | 61.36±0.11 |
| +PN(ours) | **77.45**±0.07 | **60.17**±0.23 | **78.81**±0.08 | **61.94**±0.17 | **80.87**±0.03 | **64.85**±0.07 |
| SAGE | 79.32±0.07 | 62.25±0.11 | 84.13±0.04 | 72.93±0.04 | 92.13±0.04 | 87.72±0.05 |
| +CRF* | 77.43±0.28 | 54.57±1.07 | 76.27±0.10 | 49.47±0.63 | 77.65±0.38 | 54.44±1.34 |
| +SPN* | 82.11±0.03 | 68.56±0.07 | 85.40±0.05 | 74.45±0.07 | **95.28**±0.02 | **91.99**±0.04 |
| +PN(ours) | **82.30**±0.02 | **68.93**±0.04 | **85.60**±0.05 | **74.64**±0.04 | 92.29±0.03 | 86.75±0.06 |
| GCNII | 77.94±0.08 | 65.79±0.25 | 84.81±0.06 | 74.54±0.14 | 97.53±0.01 | 95.86±0.01 |
| +CRF* | 79.98±0.32 | 61.22±1.10 | 81.73±0.33 | 66.37±0.56 | 92.11±0.28 | 87.10±0.40 |
| +SPN* | **82.01**±0.03 | 67.80±0.11 | **85.83**±0.04 | **75.96**±0.05 | **97.55**±0.01 | **95.87**±0.02 |
| +PN(ours) | 81.76±0.02 | **69.07**±0.13 | 85.45±0.05 | 75.28±0.05 | 97.47±0.03 | 95.84±0.03 |

the specific formulation of the mean field and Bethe free energy, which optimize on the following variational free energy on pairwise MRFs respectively:

$$\min_q D_{\text{KL}}(q(\mathbf{Z})\|p(\mathbf{Z} \mid \mathbf{X})), \quad \text{where } q(\mathbf{Z}) = \prod_{i\in\mathcal{V}} q_i(\mathbf{z}_i) \text{ or } \prod_{i\in\mathcal{V}} q_i(\mathbf{z}_i) \prod_{(i,j)\in\mathcal{E}} \frac{q_{ij}(\mathbf{z}_i, \mathbf{z}_j)}{q_i(\mathbf{z}_i)q_j(\mathbf{z}_j)}.$$

Compared with mean field free energy, the edge terms $\frac{q_{ij}(\mathbf{z}_i, \mathbf{z}_j)}{q_i(\mathbf{z}_i)q_j(\mathbf{z}_j)}$ in Bethe free energy models the joint dependencies between node pairs thus can result in more accurate estimations of marginals. Inspired by this difference, to predict a set of target links $\hat{\mathcal{E}} = \{(u_1, v_1), (u_2, v_2), ...\}$ that do not appear in the original graph, we add a "phantom" term for each target link into the formulation of $q(\mathbf{Z})$, yielding the following quasi distribution:

$$q(\mathbf{Z}) = \prod_{i\in\mathcal{V}} q_i(\mathbf{z}_i) \prod_{(i,j)\in\mathcal{E}} \frac{q_{ij}(\mathbf{z}_i, \mathbf{z}_j)}{q_i(\mathbf{z}_i)q_j(\mathbf{z}_j)} \prod_{(u,v)\in\hat{\mathcal{E}}} \frac{q_{uv}(\mathbf{z}_u, \mathbf{z}_v)}{q_u(\mathbf{z}_u)q_v(\mathbf{z}_v)}. \tag{2}$$

Compared with the original formulation, Eq. 2 directly consider the dependence between $u$ and $v$, and by minimizing the KL divergence we can now also obtain the approximate marginal of $q_{uv}(\mathbf{z}_u, \mathbf{z}_v)$. To reflect this modification in GNNs, we add a *phantom edge* for each target link. The phantom edges are tagged with an invented label $\hat{l}$ that distinguishes them from the original edges. Then, the link representation of $(u, v)$ is computed as $f(\boldsymbol{h}_u, \boldsymbol{h}_v)$, where $\boldsymbol{h}_u, \boldsymbol{h}_v$ are representations of nodes $u, v$ learnt by MPNNs and $f$ is a MLP. The following proposition states the relation between phantom edges and the altered Bethe free energy.

**Proposition 19.** *MPNNs with phantom edges 1-2 approximates $p(\mathbf{z}_u, \mathbf{z}_v)$ given by the modified Bethe free energy with altered distribution in Eq. 2.*

One major drawback is that when initialized with multiple phantom edges, MPNNs might no longer preserve equivariant properties. Since DropGNN (Papp et al., 2021) uses a similar technique that deletes nodes randomly but instead increases the robustness and expressiveness of GNNs, we may also suppose that MPNNs can learn to handle with phantom edges properly. We can also use similar methods for predicting $p(\mathbf{z}_i, \mathbf{z}_j, \mathbf{z}_k, ... \mid \mathbf{X})$, but since it's less practical we left the description together with detailed and practical implementation at Appendix D.

## 6 EVALUATION

We conduct two sets of experiments to empirically validate our results and verify the probabilistic inference capacity of our methods. Firstly, we verify whether our proposed phantom nodes / edges

frameworks can systematically help MPNNs to learn higher-order dependencies and joint posteriori as our theory implies. Secondly, we investigate whether MPNNs with expressiveness for complex distributions and posteriori can also help real-world node classification and link prediction tasks.

**Compared algorithms.** Our phantom nodes / edges are frameworks for lifting the probabilistic inference capacities of general GNNs. We consider popular MPNNs GCN (Kipf & Welling, 2016b), SAGE (Hamilton et al., 2017) and GCNII (Chen et al., 2020a) on all tasks. For node classification tasks, we consider SPN (Qu et al., 2022), a recent GNN-CRF framework that also lifts GNNs' probabilistic inference capacity. For link prediction tasks, We also compare with recent state-of-the-art GNNs on undirected graphs ELPH (Chamberlain et al., 2023) and BUDDY (Chamberlain et al., 2023).

**Synthetic tasks.** We generate synthetic graphs and define pairwise and 3-order potential functions. Marginals are obtained by running loopy belief propagation and are used as the training target of GNNs. The results are in Table 3. The main metric that evaluates the similarity between GNNs and marginals is the KL divergence, where our methods systematically improve over base GNNs. We also notice that for predicting edge and node pair marginals, the KL divergence between our approaches and belief propagation does not reduce to 0. We hypothesis the reason to be the limited design of the output layers. Nevertheless, our approaches steadily enhance the performance of base GNNs in different tasks and metrics.

**Real-world tasks.** We consider two types of tasks. For *Node Classification* we consider the PPI (Zitnik & Leskovec, 2017; Hamilton et al., 2017) dataset, where each node has 121 labels. To make the dataset more challenging we try using only the first 1/2/10 training graphs, yielding PPI-1, PPI-2 and PPI-10. We have rerun GCNs, SAGE, GCNII and their phantom node enhanced versions on these datasets to jointly predict all the 121 labels at once, making it more challenging than Qu et al. (2022). The MPNNs and their phantom node enhanced versions share the same hyperparameters and network architectures. Empirically, the phantom nodes work well especially when training data is limited (PPI-1), due to their ability of performing data augmentation with the awareness of higher-order structures.

For *Link Prediction* we consider the Planetoid citation networks Cora (McCallum et al., 2000), Citeseer (Sen et al., 2008) and Pubmed (Namata et al., 2012). We apply our phantom edges on GCN, SAGE and GCNII. The results are in Table 2. Our methods can systematically improve MPNNs for real-world tasks. The effects of phantom edges are significant when the graphs are complex, which demonstrates phantom edges are able to model link-level structures by aggregating representations of nodes at each end of the target links.

| Algorithm | Cora | Citeseer | Pubmed |
|---|---|---|---|
| GCN | 40.57±1.65 | 51.21±1.73 | 29.97±1.13 |
| +PE(ours) | 54.69±1.07 | 64.29±1.08 | **41.28**±0.83 |
| SAGE | 43.03±2.39 | 43.22±1.92 | 26.16±0.65 |
| +PE(ours) | 46.64±2.61 | 39.45±2.48 | 34.91±1.71 |
| GCNII | 51.52±1.25 | 53.41±0.79 | 30.93±0.92 |
| +PE(ours) | **60.09**±3.13 | **67.80**±1.17 | 36.10±1.38 |
| ELPH | 50.84±1.93 | 64.35±1.53 | 32.94±1.41 |
| BUDDY | 52.02±1.37 | 58.23±1.44 | 26.56±1.34 |

| Algorithm | | Nodes | Edges | Node pairs |
|---|---|---|---|---|
| GCN | BCE loss | 0.045 | 1.008 | 1.373 |
| | KL div | 0.003 | 0.835 | 0.748 |
| +PNE | BCE loss | **0.044** | **0.931** | **1.368** |
| | KL div | **0.000** | **0.602** | 0.729 |
| GIN | BCE loss | 0.044 | 0.991 | **1.372** |
| | KL div | 0.001 | 1.021 | 0.765 |
| +PNE | BCE loss | **0.043** | **0.964** | 1.373 |
| | KL div | **0.000** | **0.947** | **0.694** |

Table 2: Results on link prediction tasks. The metrics are Hit@10.

Table 3: Results on syntactic tasks. PNE: phantom nodes / edges.

## 7 CONCLUSION

In this paper, we develop strong theoretical connections between GNNs and variational inference, show that MPNNs can achieve good estimations of node and edge marginals, and further investigate various GNN variants from the expressiveness and link prediction perspective while corresponding them to different inference problems in graphical models. We provide new understandings about how and why previous GNN variants works well under different settings. We develop new methods that provably lift MPNNs for modeling more complex distributions and predictions. Experiments on both synthetic and real-world datasets demonstrate the effectiveness of our approaches.

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

# Appendix

## Table of Contents

## A  EXTENDED RELATED WORKS

In this section we discuss more related works that aims to improve GNNs.

**Higher order GNNs.**   Since the works of Xu et al. (2018); Morris et al. (2018) that relate GNNs with the 1-WL tests, it is straightforward to extend GNNs by imitating higher-order WL tests. Precisely, $k$-order WL tests assign colors for $k$-tuples of nodes and perform color aggregation between different tuples. Similarly, instead of learning representations for nodes, many works choose to apply the message passing paradigm in higher-order WL tests to GNNs and directly learn representations for node tuples (Morris et al., 2018; Maron et al., 2019a; 2018; 2019b; Keriven & Peyré, 2019; Azizian & Lelarge, 2020; Geerts & Reutter, 2022).

**Subgraph GNNs.**   Since the higher order GNNs are often too expensive for larger graphs, many works try to find cheaper ways to design more expressive GNNs. A variety of works feed subgraphs to MPNNs. At each layer, a set of subgraphs is generated according to some predefined permutation-invariant policies, including node deletion (Cotta et al., 2021), edge deletion Bevilacqua et al. (2021), node marking (Papp & Wattenhofer, 2022), ego-networks (Zhao et al., 2021; Zhang & Li, 2021; You et al., 2021). We will focus on the unified ESAN framework proposed by Bevilacqua et al. (2021). Qian et al. (2022); Frasca et al. (2022) studied the expressive power of different branches of subgraph GNNs.

**Substructure counting GNNs.**   There is another way to design GNNs that surpass 1-WL by constructing structural features for GNNs. Chen et al. (2020b) showed that regular MPNNs cannot capture simple patterns such as cycles, cliques and paths. Bouritsas et al. (2020); Barcel'o et al. (2021) proposed to apply substructure counting as pre-processing, and add substructure information into node features. Bodnar et al. (2021b;a); Thiede et al. (2021); Horn et al. (2021) further designed novel WL variants and proposed fully-neural approaches that captures complex substructures.

**GNNs for link prediction.**   Standard GNNs learn representations for each node. Early methods such as GAE Kipf & Welling (2016a) use GNN as an encoder and decode link representations as a function over node representation pairs. These methods are problematic in capturing complex graph structures, and might lead to poor performance. Later on, labeling trick was introduced by SEAL Zhang & Chen (2018) and adopted by GraIL Teru et al. (2019), IGMC Zhang & Chen (2020), INDIGO Liu et al. (2021), etc. These methods encode source and target nodes to mark them differently from the rest of the graph, and are proved to be more powerful than GAE. ID-GNN You et al. (2021) and NBFNet Zhu et al. (2021) both augments GNNs with the identity of the source nodes. Besides, All-path Toutanova et al. (2016) encodes relations as linear projections and proposes to efficiently aggregate all paths with dynamic programming. However, All-Path is restricted to bilinear models, has limited link prediction capability and is also not inductive. EdgeTransformer Bergen et al. (2021a) utilizes attention mechanism to learn representations for nodes and links. While it also follows the 2-FWL message passing procedure, it operates directly on fully-connected graphs and have no proposals for simplifications as we do, thus it is not scalable to larger graphs. ELPH and BUDDY (Chamberlain et al., 2023) incorporate neighbor counting into node features to enhance the link prediction performance of MPNNs.

**PGM-GNN combined methods.**   There are also works that aim to combine GNNs and PGMs to obtain better graph representations. Dai et al. (2016) first proposed to parameterize the mean field iterations and loopy belief propagation of PGMs and construct two neural architectures, one of which is similar to MPNNs. Compared with our work, Dai et al. (2016) only showed that MPNNs

parameterize the mean field iterations of a simple pairwise MRF with identical potential functions. Satorras & Welling (2021); Zhang et al. (2023b) propose to parameterize the loopy belief propagation inference algorithm for higher-order PGMs. Satorras & Welling (2021) propose to directly parameterize the loopy belief propagation algorithm, which operates on edge-level messages. Zhang et al. (2023b) propose a different message passing paradigm which consists of two distinct modules namely factor-to-variable module and variable-to-factor module. To summarize, these frameworks are fundamentally different with typical GNNs. Compared with their work, our proposed phantom node framework can also capture higher-order PGMs, but is built on MPNNs and thus can provably improve the probabilistic inference capacity of different MPNN architectures. Qu et al. (2022) propose to parameterize the potential functions of CRFs with GNNs and propose a novel method for training the CRFs. Compared with our work, they are only able to handle pairwise potentials and GNNs only serve as a component of the training and inference framework while we aim to improve GNNs themselves to be able to capture higher-order MRFs.

## B  WEISFEILER-LEHMAN TESTS

In this section we introduce the Weisfeiler-Lehman (WL) tests and their variants.

### B.1  1-WL (COLOR REFINEMENT)

The classic 1-WL test (Weisfeiler & Leman, 1968) maintains a color for each node which is refined by aggregating the colors of their neighbors. It can be easily applied on node-featured graphs (Xu et al., 2018) as in Algorithm 1.

---

**Algorithm 1:** The 1-WL test (color refinement)

**Input** : $G = (\boldsymbol{A}, \boldsymbol{X})$
1  $l \leftarrow 0$;
2  $c_v^0 \leftarrow \text{hash}(\boldsymbol{x}_v)$ for all $v \in \mathcal{V}_G$;
3  **while** *not converge* **do**
4  $\quad$ $c_v^{l+1} \leftarrow \text{hash}(c_v^l, \{\{c_u^l \mid u \in \mathcal{N}(v)\}\})$;
5  $\quad$ $l \leftarrow l + 1$;
6  **end**
7  **return** $\{\{c_v^l \mid v \in \mathcal{V}_G\}\}$;

---

The iteration converges when the partitions of nodes no longer changes. The 1-WL test decides two graphs are non-isomorphic if the multisets of colors of the two graphs are different. The WL algorithm successfully distinguishes most pairs of graphs, apart from some special examples such as regular graphs. Similarly, given a subset of nodes $C$, 1-WL define its color as $\{\{c_v^l \mid v \in C\}\}$, and 1-WL distinguishes two set of nodes if the colors of them are differernt.

### B.2  $k$-WL

The $k$-WL tests extend 1-WL to coloring $k$-tuples of nodes as in Algorithm 2, where we use $\boldsymbol{v}$ to denote a tuple of nodes, $G[\boldsymbol{v}]$ for ordered subgraphs. The neighbors $\mathcal{N}^k(\boldsymbol{v})$ are defined as follows: assume $\boldsymbol{v} = (v_1, ..., v_k)$, then $\mathcal{N}^k(\boldsymbol{v}) = (\mathcal{N}_1^k(\boldsymbol{v}), \mathcal{N}_1^k(\boldsymbol{v}), ..., \mathcal{N}_k^k(\boldsymbol{v}))$, where

$$\mathcal{N}_i^k(\boldsymbol{v}) = \{\{(v_1, ..., v_{i-1}, u, v_{i+1}, ..., v_k) \mid u \in \mathcal{V}\}\}.$$

### B.3  $k$-FWL

The $k$-FWL (Cai et al., 1989) test is equally expressive with the $(k+1)$-WL test. It has the same initialization with $(k+1)$-WL. The neighbors $\mathcal{N}^k(\boldsymbol{v})$ are defined as follows: assume $\boldsymbol{v} = (v_1, ..., v_k)$, then $\mathcal{N}^k(\boldsymbol{v}) = \{\{\mathcal{N}_u^k(\boldsymbol{v}) \mid u \in \mathcal{V}\}\}$, where

$$\mathcal{N}_u^k(\boldsymbol{v}) = ((u, v_2, ..., v_k), (v_1, u, ..., v_k), ..., (v_1, ..., u, v_k)).$$

---

**Algorithm 2:** The $k$-WL tests

---

**Input** : $G = (\boldsymbol{A}, \boldsymbol{X})$

1   $l \leftarrow 0$;

2   $c_{\boldsymbol{v}}^0 \leftarrow \mathrm{hash}(G[\boldsymbol{v}])$ for all $\boldsymbol{v} \in \mathcal{V}_G^k$;

3   **while** *not converge* **do**

4     $c_{\boldsymbol{v}}^{l+1} \leftarrow \mathrm{hash}(c_{\boldsymbol{v}}^l, \{\!\{ c_{\boldsymbol{u}}^l \mid \boldsymbol{u} \in \mathcal{N}^k(\boldsymbol{v}) \}\!\})$;

5     $l \leftarrow l + 1$;

6   **end**

7   **return** $\{\!\{ c_{\boldsymbol{v}}^l \mid v \in \mathcal{V}_G \text{ for all } v \in \boldsymbol{v} \}\!\}$;

---

**Algorithm 3:** The $k$-FWL tests

---

**Input** : $G = (\boldsymbol{A}, \boldsymbol{X})$

1   $l \leftarrow 0$;

2   $c_{\boldsymbol{v}}^0 \leftarrow \mathrm{hash}(G[\boldsymbol{v}])$ for all $\boldsymbol{v} \in \mathcal{V}_G^k$;

3   **while** *not converge* **do**

4     $c_{\boldsymbol{v}}^{l+1} \leftarrow \mathrm{hash}(c_{\boldsymbol{v}}^l, \{\!\{ c_{\boldsymbol{u}}^l \mid \boldsymbol{u} \in \mathcal{N}^k(\boldsymbol{v}) \}\!\})$;

5     $l \leftarrow l + 1$;

6   **end**

7   **return** $\{\!\{ c_{\boldsymbol{v}}^l \mid v \in \mathcal{V}_G \text{ for all } v \in \boldsymbol{v} \}\!\}$;

---

### B.4   COLORS OF $k$-WL / $k$-FWL

From the previous discussions $k$-WL and $k$-FWL both assign colors for $k$-tuples of nodes. The color of the graph $G$ is defined by

$$c_G = \mathrm{Hash}(\{\!\{ c_{\boldsymbol{v}} \mid \boldsymbol{v} \in \mathcal{V}^k \}\!\}).$$

Similarly, given any subset of nodes $\mathcal{S} \subseteq \mathcal{V}$, we also define its color as

$$c_{\mathcal{S}} = \mathrm{Hash}(\{\!\{ c_{\boldsymbol{v}} \mid \boldsymbol{v} \in \mathcal{S}^k \}\!\}).$$

## C   DISCUSSIONS ABOUT THE FIXED POINT PROPERTIES OF MPNNS AND BETHE APPROXIMATION

In the previous discussion we have seen that MPNNs are capable of approximating local extrema of the Bethe approximation. To further extend our results we ask: can MPNNs capture *all* local extrema of the Bethe approximation? We first notice that recently a class of neural networks (Bai et al., 2019) treat the fixed points of their layers as outputs. To better discuss the properties of MPNNs, we try to establish a general correspondence between MPNN layers and the Bethe free energy in the perspective of fixed points.

**Theorem 20.** *Given a 1-WL distinguishable 2-order $\mathcal{A}$, there is a MPNN layer $\mathcal{L}$ such that all fixed points of $\boldsymbol{H} = \mathcal{L}(\mathrm{Concat}(\boldsymbol{H}, \boldsymbol{X}), \boldsymbol{A})$ implies local extrema of the Bethe free energy of the MRF given any graph $G$, and also all local extrema of Bethe approximation are implied in $\mathcal{L}$. That is, there exists a function $f$ such that given any graph $G$ (of the size $n$), let $\mathcal{H}$ be the set of fixed points of $\mathcal{L}$ and $\mathcal{S}$ be the set of local extrema of Bethe approximation, then $f$ is always surjective from $\mathcal{H}$ to $\mathcal{S}$.*

Recall that the fixed points of Bethe approximation contain not only node marginals but also edge marginals, thus Theorem 20 confirms the fact that MPNNs are actually able to learn edge representations. In practice, there are several fixed-point MPNNs (Gu et al., 2020; Park et al., 2021; Liu et al., 2022b;a) that work well on node classification, and Theorem 20 suggests the possibility of using them to learn node and edge marginals of Bethe approximation.

# D    EXTENDED DISCUSSIONS ABOUT PHANTOM NODES / EDGES

In this section we give extended description of the proposed phantom nodes / edges methods.

## D.1    PHANTOM NODES

We summarize the procedure in Algorithm 4. For the original phantom nodes, the function FindCliques finds maximum cliques in a given graph; for the relaxed version the function finds all cliques with no more than $k$ nodes.

---

**Algorithm 4:** The Phantom Node Procedure

**Input**  : $G = (\boldsymbol{A}, \boldsymbol{X})$
**Output:** $\hat{G} = (\hat{\boldsymbol{A}}, \hat{\boldsymbol{X}})$
1  $\mathcal{C} \leftarrow$ FindCliques$(G)$;
2  $(N, d) \leftarrow$ Shape$(\boldsymbol{X})$;
3  $\hat{N} \leftarrow N + |\mathcal{C}|$;
4  $\hat{d} \leftarrow d + 1$;
5  $\hat{\boldsymbol{A}} \leftarrow$ Zeros$(\hat{N}, \hat{N})$;
6  $\hat{\boldsymbol{X}} \leftarrow$ Zeros$(\hat{N}, \hat{d})$;
7  $\hat{\boldsymbol{A}}_{:N,:N} \leftarrow \boldsymbol{A}$;
8  $\hat{\boldsymbol{X}}_{:N,:d} \leftarrow \boldsymbol{X}$;
9  $p = N + 1$;
10 **for** $C \in \mathcal{C}$ **do**
11  $\quad$ $\hat{\boldsymbol{X}}_{p,\hat{d}} \leftarrow 1$;
12  $\quad$ **for** $i \in C$ **do**
13  $\quad\quad$ $\hat{\boldsymbol{A}}_{i,p} \leftarrow 1$;
14  $\quad\quad$ $\hat{\boldsymbol{A}}_{p,i} \leftarrow 1$;
15  $\quad$ **end**
16 **end**
17 **return** $\hat{G} = (\hat{\boldsymbol{A}}, \hat{\boldsymbol{X}})$;

---

The procedure is standard and easy to implement, but it might not suit most practical MPNN architectures such as GCNs, SAGE, etc. because the MPNNs will have to learn from data to treat the phantom nodes (tagged with only the last dimension of the input node feature) are differently from the regular nodes. A practical implementation is realized as follows, where we use $i, j$ to refer to regular nodes and $p$ to refer to phantom nodes.

$$\boldsymbol{h}_i^{t+1} = \mathrm{COM}^t \left( \boldsymbol{h}_i^t, \mathrm{AGG}_{PN}^t \left( \left\{\!\left\{ \mathsf{h}_p^t \mid p \in \mathcal{N}(i) \cap \hat{\mathcal{V}} \right\}\!\right\} \right), \mathrm{AGG}^t \left( \left\{\!\left\{ \boldsymbol{h}_j^t \mid j \in \mathcal{N}(i) \right\}\!\right\} \right) \right),$$

$$\boldsymbol{h}_p^{t+1} = \mathrm{C\hat{O}M}^t \left( \mathsf{h}_p^t, \mathrm{A\hat{G}G}^t \left( \left\{\!\left\{ \boldsymbol{h}_i^t \mid i \in \mathcal{N}(p) \right\}\!\right\} \right) \right),$$

where $\hat{\mathcal{V}}$ is the set of phantom nodes.

**Algorithm complexities.**    We now discuss the time and space complexities of the proposed Algorithm 4 and compare with $k$-GNNs.

The major complexities come from finding cliques in line 1. Suppose the input graph $G$ contains $N$ nodes. In worst cases when the graph is dense (contains $\mathcal{O}(N^2)$ edges), FindCliques would return $\mathcal{O}\left( \binom{N}{k} \right) = \mathcal{O}(N^k)$ results (suppose $k << N$) and takes no more than $\mathcal{O}(N^k)$ time (Nesetril & Poljak, 1985). In this case the augmented graph $\hat{G}$ contains $\hat{N} = \mathcal{O}(N^k)$ nodes and $\mathcal{O}(kN^k)$ edges. Applying MPNNs on such augmented graph then takes $\mathcal{O}(kN^k)$ time at each layer (for ease of comparison we only consider terms w.r.t. sizes of graphs). To summarize, the algorithm takes $\mathcal{O}(N^k)$ space and $\mathcal{O}(kN^k)$ time. $k$-GNNs takes $\mathcal{O}(N^k)$ space and $\mathcal{O}(kN^{k+1})$ time.

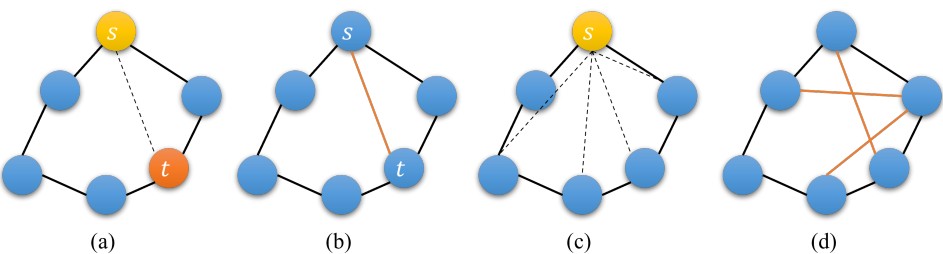

Figure 1: Relation between node labeling and phantom edges.

If the input graph is sparse, our method is far more efficient than $k$-GNNs when the number of $k$-cliques is much less than $N^k$. Suppose there are $C$ cliques, then the augmented graph contains $\mathcal{O}(N+C)$ nodes and $\mathcal{O}(M+kC)$ edges, where $M$ is the number of edges in the original graph, and each layer of the MPNNs only takes $\mathcal{O}(M+kC)$ time and $\mathcal{O}(N+C)$ space. The time complexity of finding cliques in sparse graphs is hard to estimate, but there are empirically efficient algorithms that can effective find $k$-cliques in large graphs (see, eg. Gianinazzi et al. (2021)). Since `FindCliques` is a one-time pre-processing procedure, the proposed algorithm is much more efficient than $k$-GNNs.

**Limitations.**     From the above analysis it is evident that when the input graph is large and dense, the proposed method might add a large amount of phantom nodes to the input graph, making the algorithm inefficient especially when $k$ is set to a large value. Even so, the proposed method is still more efficient than $k$-GNNs.

### D.2   PHANTOM EDGES

It's simple to extend MPNNs to learn edges of different types, and our practical implementation here follows the common approaches Schlichtkrull et al. (2017). Let $\mathcal{E}$ denote the set of regular edges and $\hat{\mathcal{E}}$ the set of phantom edges. Let $\mathcal{N}(i)$ denote the neighbors of $i$ via regular edges and $\hat{\mathcal{N}}(i)$ the neighbors of $i$ via phantom edges. Our practical implementation is

$$\boldsymbol{h}_i^{t+1} = \mathbf{h}_i^{t+1} + \lambda \mathbf{h}_i^{t+1},$$
$$\mathbf{h}_i^{t+1} = \mathrm{COM}^t\left(\boldsymbol{h}_i^t, \mathrm{AGG}^t\left(\{\{\boldsymbol{h}_j^t \mid j \in \mathcal{N}(i)\}\}\right)\right),$$
$$\mathbf{h}_i^{t+1} = \hat{\mathrm{COM}}^t\left(\boldsymbol{h}_i^t, \hat{\mathrm{AGG}}^t\left(\{\{\boldsymbol{h}_j^t \mid j \in \hat{\mathcal{N}}(i)\}\}\right)\right),$$

Note that, we can also implement phantom edges for predicting $p(\mathbf{z}_{i_1}, ..., \mathbf{z}_{i_k} \mid \mathbf{X})$. This is done by noticing that belief propagation on factor graphs also produces marginals for factors, i.e., cliques of nodes. By adding a phantom node which is connected with $i_1, ..., i_k$, we have shown in the proof of Proposition 17, 18 that by applying MPNNs we can also learn the posteriori $p(\mathbf{z}_{i_1}, ..., \mathbf{z}_{i_k} \mid \mathbf{X})$, even if the nodes $i_1, ..., i_k$ are not connected in the original graph.

**Algorithm complexities.**     We now discuss the time and space complexities of the proposed phantom edges and compare with node labeling methods. Given a graph $G$ with $N$ nodes and $M$ edges, we are asked to predict a batch of links $\hat{\mathcal{E}}$. Simultaneously predicting all links in $\hat{\mathcal{E}}$ is equivalent to applying MPNNs on an induced graph with $N$ nodes and $M + |\hat{\mathcal{E}}|$ edges thus requires $\mathcal{O}(N + M + |\hat{\mathcal{E}}|)$ time. For labeling trick methods, they need to rerun GNNs for each target link thus requires $\mathcal{O}(\hat{\mathcal{E}}(N + M))$ time. For source node labeling methods, they need to rerun GNNs for target links with different source nodes. Therefore, suppose the target links $\hat{\mathcal{E}}$ contains $S$ different source nodes, they require $\mathcal{O}(S(N + M))$ time.

**Relation with node labeling methods.**     We now discuss the relation between our methods and node labeling methods. We will mainly focus on the ordered labeling trick (Zhang et al., 2020; Liu et al., 2021) and the partial node labeling (Zhu et al., 2021).

Table 4: Statics of datasets.

| Dataset | Task | # Features | # Labels | # Nodes | # Edges |
|---------|------|-----------|----------|---------|---------|
| PPI | NC | 50 | 121 | 56944 | 818716 |
| Cora | LP | 1433 | 1 | 2708 | 5278 |
| Citeseer | LP | 3703 | 1 | 3327 | 4676 |
| Pubmed | LP | 500 | 1 | 18717 | 44237 |

We first show that labeling trick can be seen as a special case of the phantom edge. Suppose given a graph $G$, the target links $\hat{\mathcal{E}}$ only contain one link $\hat{\mathcal{E}} = \{(s,t)\}$. In this case, we only add one phantom edge $(s,t)$ into the original graph (Figure 1 (b)). Labeling trick methods add additional labels to mark $s, t$ differently from the rest of the nodes (Figure 1 (a)). As we shall see later, both the two methods aim to mark the target link $(s,t)$ uniquely: that is, for any graphs $G, H$ and $s, t \in \mathcal{V}_G, p, q \in \mathcal{V}_H$, obviously

$$\hat{G} \simeq \hat{H} \iff \mathcal{G} \simeq \mathcal{H} \iff \text{There exists an isomorphism } \pi \text{ from } G \text{ to } H, \pi(s) = p, \pi(t) = q,$$
(3)

where $\hat{G}, \hat{H}$ are graphs induced by phantom edges and $\mathcal{G}, \mathcal{H}$ are graphs induced by labeling trick. Therefore, both the methods aim to tag the target link uniquely.

Now we discuss how partial node labeling and general phantom edges extend the above methods to obtain more efficient variants. As we shall see later, both the methods aim to produce representations of multiple links within one run of GNNs. Since the labeling trick methods can only compute the representation of one link, partial node labeling methods relax this deficiency by modifying the node-level labels. By only considering the source nodes (or the tail nodes), they are able to produce representations of links that share the same source (or tail) node within one run of GNNs (Figure 1 (c)). In contrast, our methods work at the edge level: we extend (b) in Figure 1 by allowing more than one phantom edges to present simultaneously (Figure 1 (d)). Compared with the partial node labeling methods, the advantage of our method is that it is more flexible, since we allow the simultaneous computation of link representations that not necessary share the same source (tail) node. The limitation, however, is that our methods no longer strictly preserve the isomorphism property in Eq. 4.

## E    EXPERIMENTAL DETAILS

In this section we describe our experimental setup in more details.

### E.1    DATASETS

Statics of datasets are in Table 4.

For node classification tasks, the experiments follow the settings in Qu et al. (2022). For fairness we have rerun the results GCN (Kipf & Welling, 2016b), SAGE (Hamilton et al., 2017) and GCNII Chen et al. (2020a) on the datasets. We train the GNNs to output 121 dimensional vectors, with each dimension corresponds to one of the labels.

For link prediction tasks, the experiment configurations follow the settings in Chamberlain et al. (2023). To make the tasks more challenging we record the Hit@10 metric. For all tasks we use negative sampling to generate negative targets. At training time the message passing links are equal to the supervision links, while at test time disjoint sets of links are held out that are never seen at training time. We random generate 70-10-20 percent train-val-test splits which is the same as Chamberlain et al. (2023). The predictor of the model is designed as $p(u,v) = \text{MLP}(\boldsymbol{h}_u \odot \boldsymbol{h}_v)$ where $\boldsymbol{h}_u, \boldsymbol{h}_v$ are node representations of $u, v$ respectively.

For syntactic tasks, we first randomly generate a set of graphs. Then, we define potential functions of MRFs. For simplicity we select the following potential functions:

$$\Psi_2(\mathbf{z}_i, \mathbf{z}_j) = \begin{cases} 1, & \mathbf{z}_i = \mathbf{z}_j, \\ 2, & \text{else}, \end{cases}$$

for all edges, and select

$$\Psi_3(\mathbf{z}_i, \mathbf{z}_j, \mathbf{z}_k) = \begin{cases} 10, & \mathbf{z}_i = \mathbf{z}_j = \mathbf{z}_k, \\ 1, & \text{else,} \end{cases}$$

for all 3-cliques. We then run loopy belief propagation on the graphs and obtain node marginals and edge marginals. We also obtain node pair marginals by adding phantom terms as discussed in Section 5. These marginals are use as training targets for GNNs, where we optimize on the cross entropy loss. We also evaluate the KL divergence between the marginals produced by GNNs and the marginals produced by loopy belief propagation.

### E.2 CONFIGURATIONS

**Phantom nodes for node classification.**  We apply our phantom nodes on GNNs without changing hyperparameters or network structures. We choose the Adam (Kingma & Ba, 2014) optimizer, with learning rate $5 \times 10^{-3}$, weight decay 0.

**GCN (Kipf & Welling, 2016b).**  We set the numbers of hidden neurons to 128, and the number of layers to 2. We use ReLU (Nair & Hinton, 2010) as the activation function. We set dropout rate to be 0.5 and apply layer normalization.

**SAGE (Hamilton et al., 2017).**  We set the numbers of hidden neurons to 1024, and the number of layers to 2. We use ReLU (Nair & Hinton, 2010) as the activation function. We set dropout rate to be 0.5 and apply layer normalization.

**GCNII (Chen et al., 2020a).**  We set the numbers of hidden neurons to 1024, and the number of layers to 5. We use ReLU (Nair & Hinton, 2010) as the activation function. We set dropout rate to be 0.5 and do not apply layer normalization.

**Implementation with phantom nodes.**  The configurations of GNN layers are the same. For each input graph we find all maximum cliques. The layers are implemented as follows,

$$\boldsymbol{H}^{t+1} = \boldsymbol{H}_{reg}^{t+1} + \tilde{\boldsymbol{A}}_{ph}\boldsymbol{H}_{pht}^{t+1},$$
$$\boldsymbol{H}_{reg}^{t+1} = \text{Layer}_1(\boldsymbol{H}^t, \boldsymbol{A}_{reg}),$$
$$\boldsymbol{H}_{pht}^{t+1} = \text{Layer}_2(\boldsymbol{H}^t, \boldsymbol{A}_{pht}),$$

where $\boldsymbol{A}$ is the adjacency matrix, $\boldsymbol{A}_{pht}$ is the adjacency matrix w.r.t. phantom nodes, i.e. only edges connected with phantom nodes are considered. $\boldsymbol{A}_{reg}$ is the original adjacency matrix. $\tilde{\boldsymbol{A}_{pht}}$ is the normalized version of $\boldsymbol{A}_{ph}$.

**Phantom edges for link prediction.**  We apply our phantom nodes on GNNs without changing hyperparameters or network structures. We choose the Adam (Kingma & Ba, 2014) optimizer, with learning rate $1 \times 10^{-4}$, weight decay $5 \times 10^{-4}$.

**GCN (Kipf & Welling, 2016b).**  We set the numbers of hidden neurons to 1024, and the number of layers to 2. We use ReLU (Nair & Hinton, 2010) as the activation function. We set dropout rate to be 0.5 and apply layer normalization.

**SAGE (Kipf & Welling, 2016b).**  We set the numbers of hidden neurons to 1024, and the number of layers to 2. We use ReLU (Nair & Hinton, 2010) as the activation function. We set dropout rate to be 0.5 and apply layer normalization.

**GCNII (Chen et al., 2020a).**  We set the numbers of hidden neurons to 1024, and the number of layers to 2. We use ReLU (Nair & Hinton, 2010) as the activation function. We set dropout rate to be 0.5 and do not apply layer normalization.

Table 5: Results on the DBLP dataset built by Qu et al. (2022).

| Model | GAT | GAT-SPN | GAT-PN | GCNII | GCNII-SPN | GCNII-PN |
|---|---|---|---|---|---|---|
| **Accuracy** | 79.16 | 84.84 | 86.34 | 81.79 | 83.57 | 85.08 |

**Implementation with phantom edges.**  The configurations of GNN layers are the same. We set the batch size to 1024. At each iteration, we randomly select 1024 edges as positive and negative training targets, and add 1024 phantom edges to the original graphs. The layers are implemented as follows,

$$\boldsymbol{H}^{t+1} = \text{Layer}_1(\boldsymbol{H}^t, \boldsymbol{A}_{reg}) + \lambda \text{Layer}_2(\boldsymbol{H}^t, \boldsymbol{A}_{pht}),$$

where $\boldsymbol{A}_{reg}$ is the original adjacency matrix, $\boldsymbol{A}_{pht}$ is the phantom edge matrix. To predict the link $(i, j)$ we have

$$\boldsymbol{y}_{ij} = \text{MLP}(\boldsymbol{h}_i \odot \boldsymbol{h}_j),$$

where $\odot$ is the Hadamard production.

### E.3    ADDITIONAL EXPERIMENTS

We also conduct additional experiments on DBLP (Tang et al., 2008) datasets. The experimental setup follows Qu et al. (2022), where papers from eight conferences are treated as nodes, and are splitted them into three categories for classification according to conference domains. For each paper, the mean GloVe embedding (Pennington et al., 2014) of words in the title and abstract as node features. The training/validation/test graph is formed as the citation graph of papers published before 1999, from 2000 to 2009, after 2010 respectively. The results are in Table 5.

## F    PROOF W.R.T. INVARIANT MRFS

### F.1    PROOF OF LEMMA 1

**Lemma 1.**  *Suppose there is an algorithm $\mathcal{F}$, given any graph $G = (\boldsymbol{A}, \boldsymbol{X})$, $\mathcal{F}$ maps its structure $\boldsymbol{A}$ to a MRF $p_G = \mathcal{F}(\boldsymbol{A})$. Then, $\mathcal{F}$ satisfies that for any permutation $\pi : [n] \to [n]$ and $\boldsymbol{A}$:*

$$p_{\pi(G)}(\pi(\mathbf{X}), \pi(\mathbf{Z})) = p_G(\mathbf{X}, \mathbf{Z}),$$

*if and only if $p_G$ can be represented in the form 1 with potentials given by $\Psi_C = \mathcal{A}(C, \boldsymbol{A})$ where $\mathcal{A}$ is permutation-invariant:*

$$\mathcal{A}(C, \boldsymbol{A}) = \mathcal{A}(\pi(C), \pi(\boldsymbol{A})) \quad \forall\, C, G, \text{ and } \pi : [n] \to [n].$$

*Proof.*  Since MRFs can be written in the form of

$$p(\mathbf{X}, \mathbf{Z}) = \frac{1}{Z} \prod_{i \in \mathcal{V}} \Psi_i(\mathbf{z}_i, \mathbf{x}_i) \prod_{C \in \mathcal{C}} \Psi_C(\mathbf{z}_C),$$

therefore $\mathcal{F}$ can be seen as a function that maps a graph $G = (\boldsymbol{A}, \boldsymbol{X})$ to a set of potential functions $\{\Psi_C \mid C \in \mathcal{C}_G \cup \mathcal{V}_G\}$.

**1→2.**  Given any $\mathcal{F}$, we define the corresponding $\mathcal{A}$ as follows. Given a graph $G$, we can identify all graphs that are isomorphic to $G$, and we suppose these graphs compose a set $\mathcal{G}$ which we refer to as the isomorphism set of $G$. For each isomorphism set $\mathcal{G}$ of arbitrary graph, we arbitrarily specify an element $G_0$ in $\mathcal{G}$ and refer to it as the isomorphism prototype of all graphs in $\mathcal{G}$. Then, for any graph, $G$ we define $\mathcal{A}$ as

$$\mathcal{A}(C, G) = \Psi_{C_0},$$

where $\Psi_{C_0}$ is computed by $\mathcal{F}(G_0)$, $G_0$ is the isomorphism prototype of $G$ with $G_0 = \pi(G)$, and $C_0 = \pi(C)$. Now we show $\mathcal{A}$ satisfies the constraint. Given any graph $G$ and permutation $\pi : [n] \to [n]$, we have

$$\mathcal{A}(C, G) = \Psi_{C_0} = \mathcal{A}(\pi(C), \pi(G)).$$

Since $\mathcal{F}$ satisfies $p_{\pi(G),\mathcal{F}}(\pi(\mathbf{X}), \pi(\mathbf{Z})) = p_{G,\mathcal{F}}(\mathbf{X}, \mathbf{Z})$, we also have

$$p_{G,\mathcal{F}}(\mathbf{X}, \mathbf{Z}) = p_{G_0,\mathcal{F}}(\pi(\mathbf{X}), \pi(\mathbf{Z})),$$

thus $p_{G,\mathcal{F}}$ can be represented by $\mathcal{A}$.

**2→1.** For any $G$, we denote $\Psi_C = \mathcal{A}(C, G)$. Then, we have

$$
\begin{aligned}
p_{\pi(G),\mathcal{A}}(\pi(\mathbf{X}), \pi(\mathbf{Z})) &= \frac{1}{Z} \prod_{i \in \pi(\mathcal{V})} \mathcal{A}(i, \pi(G))(\mathbf{z}_{\pi(i)}, \mathbf{x}_{\pi(i)}) \prod_{C \in \pi(\mathcal{C})} \mathcal{A}(C, \pi(G))(\mathbf{z}_{\pi(C)}) \\
&= \frac{1}{Z} \prod_{i \in \mathcal{V})} \mathcal{A}(\pi(i), \pi(G))(\mathbf{z}_i, \mathbf{x}_i) \prod_{C \in \mathcal{C}} \mathcal{A}(\pi(C), \pi(G))(\mathbf{z}_C) \\
&= \frac{1}{Z} \prod_{i \in \mathcal{V}} \Psi_i(\mathbf{z}_i, \mathbf{x}_i) \prod_{C \in \mathcal{C}} \Psi_C(\mathbf{z}_C) \\
&= p_{G,\mathcal{A}}(\mathbf{X}, \mathbf{Z}).
\end{aligned}
$$

$\square$

### F.2 PROOF OF THEOREM 2.

**Theorem 2.** *Consider simple pairwise MRFs parameterized with:*

$$
\mathcal{A}(C, G) = \begin{cases} \Phi_1, & |C| = 1, \\ \Phi_2, & |C| = 2, \end{cases}
$$

*Given any connected graphs $G = (\boldsymbol{A}_G, \boldsymbol{X}_G)$, $H = (\boldsymbol{A}_H, \boldsymbol{X}_H)$ and $i \in \mathcal{V}_G$, $j \in \mathcal{V}_H$. If $i$ and $j$ are not isomorphic, then there exists $\Phi_1, \Phi_2$ and $\boldsymbol{z} \in \mathcal{Z}$ such that the pair-wise MRFs $p_G, p_H$ specified by $G$ and $H$ satisfy:*

$$p_G(\mathbf{z}_i = \boldsymbol{z} \mid \mathbf{X} = \boldsymbol{X}_G) \neq p_H(\mathbf{z}_j = \boldsymbol{z} \mid \mathbf{X} = \boldsymbol{X}_H). \tag{4}$$

*Proof.* We proof the theorem by manually constructing $\Phi_1, \Phi_2$ that distinguishes nodes $i, j$. Without loss of generality we may assume node features are taken from a finite discrete space $\mathbf{x}_i \in \{\boldsymbol{x}_1, ..., \boldsymbol{x}_l\}$. If not, we simply encode all features in $G$ and $H$ into such a discrete space.

By flatting Eq. 4 we have

$$
\begin{aligned}
& p_G(\mathbf{z}_i = \boldsymbol{z} \mid \mathbf{X} = \boldsymbol{X}_G) \neq p_H(\mathbf{z}_j = \boldsymbol{z} \mid \mathbf{X} = \boldsymbol{X}_H) \\
\iff & \frac{1}{Z_G} \sum_{\mathbf{Z}_{\backslash i}} \prod_{k \in [n]} \Phi_1(\mathbf{x}_{G,k}, \mathbf{z}_k) \prod_{(k,l) \in \mathcal{E}_G} \Phi_2(\mathbf{z}_k, \mathbf{z}_l) \\
& \neq \frac{1}{Z_H} \sum_{\mathbf{Z}_{\backslash i}} \prod_{k \in [n]} \Phi_1(\mathbf{x}_{H,k}, \mathbf{z}_k) \prod_{(k,l) \in \mathcal{E}_H} \Phi_2(\mathbf{z}_k, \mathbf{z}_l) \\
\iff & Z_H \sum_{\mathbf{Z}_{\backslash i}} \prod_{k \in [n]} \Phi_1(\mathbf{x}_{G,k}, \mathbf{z}_k) \prod_{(k,l) \in \mathcal{E}_G} \Phi_2(\mathbf{z}_k, \mathbf{z}_l) \\
& \neq Z_G \sum_{\mathbf{Z}_{\backslash i}} \prod_{k \in [n]} \Phi_1(\mathbf{x}_{H,k}, \mathbf{z}_k) \prod_{(k,l) \in \mathcal{E}_H} \Phi_2(\mathbf{z}_k, \mathbf{z}_l),
\end{aligned}
$$

where $\sum_{\mathbf{Z}_{\backslash i}}$ denotes summing over all $\mathbf{z}_1, \mathbf{z}_2...$ in $\mathbf{Z}$ except $\mathbf{z}_i$. If we combine $G, H$ into a huge graph $G' = (\boldsymbol{A}, \boldsymbol{X})$ with $(|\mathcal{V}_G| + |\mathcal{V}_H|)$ nodes such that

$$
\boldsymbol{A} = \begin{bmatrix} \boldsymbol{A}_G, \boldsymbol{O} \\ \boldsymbol{O}, \ \boldsymbol{A}_H \end{bmatrix},
$$

$$
\boldsymbol{X} = \begin{bmatrix} \boldsymbol{X}_G \\ \boldsymbol{X}_H \end{bmatrix},
$$

and denote its edges as $\mathcal{E}$ and nodes as $\mathcal{V}$, and let $i' = i, j' = j + |\mathcal{V}_G|$ be the nodes corresponding to $i, j$ in $G, H$, we can describe Eq. 4 over $G'$ as

$$p_G(\mathbf{z}_i = \boldsymbol{z} \mid \mathbf{X} = \boldsymbol{X}_G) \neq p_H(\mathbf{z}_j = \boldsymbol{z} \mid \mathbf{X} = \boldsymbol{X}_H)$$

$$\iff \sum_{\mathbf{Z}_{\backslash i'}} \prod_{k \in \mathcal{V}} \Phi_1(\mathbf{x}_k, \mathbf{z}_k) \prod_{(k,l) \in \mathcal{E}} \Phi_2(\mathbf{z}_k, \mathbf{z}_l)$$

$$\neq \sum_{\mathbf{Z}_{\backslash j'}} \prod_{k \in \mathcal{V}} \Phi_1(\mathbf{x}_k, \mathbf{z}_k) \prod_{(k,l) \in \mathcal{E}} \Phi_2(\mathbf{z}_k, \mathbf{z}_l)$$

$$\iff Z_{G'} p_{G'}(\mathbf{z}_{i'} = \boldsymbol{z} \mid \mathbf{X}) \neq Z_{G'} p_{G'}(\mathbf{z}_{j'} = \boldsymbol{z} \mid \mathbf{X})$$

To proceed, we assume latent variables $\mathbf{z}_k$ are taken from a discrete space $\mathbf{z}_k \in \mathcal{Z} = \{1, ..., nl^n 2^{n^2}\}$ where $n = |\mathcal{V}|$. Obviously we can encode a tuple $(b, \boldsymbol{A}, \boldsymbol{X})$ into the space of $\mathbf{z}_k$ where $b \in \{1, ..., n\}, \boldsymbol{A} \in \{0, 1\}^{n \times n}, \boldsymbol{X} \in \mathbb{R}^{n \times d}$ with a bijective mapping $f(b, \boldsymbol{A}, \boldsymbol{X})$. To lighten notations we denote $(b(\mathbf{z}_k), \boldsymbol{A}(\mathbf{z}_k), \boldsymbol{X}(\mathbf{z}_k)) = f^{-1}(\mathbf{z}_k)$ to be the inverse mapping of $f$. Given $(\boldsymbol{A}, \boldsymbol{X})$, we design the potential functions as follows:

$$\Phi_1(\mathbf{z}_i, \mathbf{x}_i) = \mathbf{1}\{\boldsymbol{X}(\mathbf{z}_i)_{b(\mathbf{z}_i)} = \mathbf{x}_i\} \cdot g(b(\mathbf{z}_i)),$$

$$\Phi_2(\mathbf{z}_i, \mathbf{z}_j) = \mathbf{1}\left\{ \begin{array}{l} \boldsymbol{A}(\mathbf{z}_i) = \boldsymbol{A}(\mathbf{z}_j), \text{ and} \\ \boldsymbol{X}(\mathbf{z}_i) = \boldsymbol{X}(\mathbf{z}_j), \text{ and} \\ \boldsymbol{A}(\mathbf{z}_i)_{b(\mathbf{z}_i), b(\mathbf{z}_j)} = 1 \end{array} \right\},$$

where we define $g(b) = (n! + 1)^{n^b}$. Denote $g_{i'}(\boldsymbol{z}) = Z_{G'} p_{G'}(\mathbf{z}_{i'} = \boldsymbol{z} \mid \mathbf{X})$, we now prove

$$g_{i'}(f(i', \boldsymbol{A}, \boldsymbol{X})) \neq g_{j'}(f(i', \boldsymbol{A}, \boldsymbol{X})),$$

thus proving the theorem. We do this by manually summing all non-zero assignments of $\mathbf{Z}$ and show that the special bit of the base-$(n! + 1)$ numbers of $g_{i'}(f(i', \boldsymbol{A}, \boldsymbol{X}))$ and $g_{j'}(f(i', \boldsymbol{A}, \boldsymbol{X}))$ are different. First, it's obvious that both $g_{i'}(f(i', \boldsymbol{A}, \boldsymbol{X}))$ and $g_{j'}(f(i', \boldsymbol{A}, \boldsymbol{X}))$ are integers. Now let's consider the possible values of $\mathbf{Z}$ given $\mathbf{z}_{i'} = f(i', \boldsymbol{A}, \boldsymbol{X})$. Obviously all $\mathbf{z}_k$ for $k \in \mathcal{V}$ must share the same $\boldsymbol{X}$ and $\boldsymbol{A}$, thus the only difference is the assignments of $b$. Note that the non-zero probability assignments of $b$ satisfying $\{\{b(\mathbf{z}_1), ..., b(\mathbf{z}_n)\}\} = \{\{b_1, ..., b_n\}\}$ is exactly the $\left(\sum_{i \in [n]} n^{b_i}\right)$-bit of the base-$(n! + 1)$ form of $g_{i'}(f(i', \boldsymbol{A}, \boldsymbol{X}))$ (Since there are no more than $n!$ permutations of $\{\{b_1, ..., b_n\}\}$, no carry-in are produced). Thus the $\left(\sum_{i \in [n]} n^i\right)$ bit of the base-$(n! + 1)$ form of $g_{i'}(f(i', \boldsymbol{A}, \boldsymbol{X}))$ corresponds to situations where $\{\{b(\mathbf{z}_1), ..., b(\mathbf{z}_n)\}\} = \{\{1, 2, ..., n\}\}$. Clearly, $b(\mathbf{z}_k) = k$ for $k \in [n]$ is a valid situation for calculating $g_{i'}(f(i', \boldsymbol{A}, \boldsymbol{X}))$, therefore the $\left(\sum_{i \in [n]} n^i\right)$ bit is non-zero. In contrast, because $G$ and $H$ are non-isomorphic, when calculating $g_{j'}(f(i', \boldsymbol{A}, \boldsymbol{X}))$ it's impossible to find a valid assignment of $b$ that both satisfies $\{\{b(\mathbf{z}_1), ..., b(\mathbf{z}_n)\}\} = \{\{1, 2, ..., n\}\}$ and makes the potentials non-zero, therefore the $\left(\sum_{i \in [n]} n^i\right)$ bit is zero. Thus, we have

$$g_{i'}(f(i', \boldsymbol{A}, \boldsymbol{X})) \neq g_{j'}(f(i', \boldsymbol{A}, \boldsymbol{X})).$$

$\square$

# G  PROOF W.R.T. MPNNS

## G.1  PROOF OF THEOREM 3

**Theorem 3.** *Given any graphs $G = (\boldsymbol{A}_G, \boldsymbol{X}_G), H = (\boldsymbol{A}_H, \boldsymbol{X}_H)$ and any 1-WL distinguishable 2-order $\mathcal{A}$. If at some iteration, the messages of parallel belief propagation on $G$ and $H$ are different, then 1-WL also distinguishes $G, H$.*

*Proof.* **1→2.** We suppose after iteration $t$, 1-WL assigns different colors for nodes $i, j$ in $G, H$, and denote the node colors as $\mathrm{Col}(i, G)$ and $\mathrm{Col}(j, H)$. We define $\mathbf{z}_i \in \mathcal{Z} = \{0, 1\}$. Denoting

$\Psi_{k,G'} = \mathcal{A}(k, G')$ and $\Psi_{\{k,l\},G'} = \mathcal{A}(\{k, l\}, G')$, we let

$$\Psi_{k,G'}(0) = \begin{cases} 1, & \mathrm{Col}(\mathrm{G}', \mathrm{k}) = \mathrm{Col}(\mathrm{G}, \mathrm{i}), \\ 0, & else. \end{cases}$$

And

$$\Psi_{k,G'}(1) \equiv 0,$$
$$\Psi_{\{k,l\},G'}(\cdot) \equiv 1.$$

Obviously, we have $q_G(\mathbf{z}_i = 0 \mid \boldsymbol{X}_G) > 0$ while $q_H(\mathbf{z}_j = 0 \mid \boldsymbol{X}_H) = 0$ for both methods.

**2→1.** The standard belief propagation is summarized as follows.

**Probabilistic formulation.** Given any graph, the conditional distribution $p(\mathbf{Z} \mid \mathbf{X})$ is given as

$$p(\mathbf{Z} \mid \mathbf{X}) = \frac{1}{Z} \prod_{i \in \mathcal{V}} \Psi_i(\mathbf{z}_i, \mathbf{x}_i) \prod_{(i,j) \in \mathcal{E}} \Psi_{ij}(\mathbf{z}_i, \mathbf{z}_j).$$

**Initialization.** For $(i, j) \in \mathcal{E}$ and $\mathbf{z}_i, \mathbf{z}_j \in \mathcal{Z}$,

$$m^0_{i \to j}(\mathbf{z}_j) \propto 1 \propto m^0_{j \to i}(\mathbf{z}_i).$$

**Message passing.** $(t = 0, 1, 2, ...)$ For $i \in \mathcal{V}$ and $j \in \mathcal{N}(i)$,

$$m^{t+1}_{i \to j}(\mathbf{z}_j) \propto \sum_{\mathbf{z}_i \in \mathcal{Z}} \Psi_i(\mathbf{z}_i, \mathbf{x}_i) \Psi_{ij}(\mathbf{z}_i, \mathbf{z}_j) \prod_{k \in \mathcal{N}(i) \backslash j} m^t_{k \to i}(\mathbf{z}_i),$$

where all messages are normalized: $\sum_{\mathbf{z}_j \in \mathcal{Z}} m^t_{i \to j}(\mathbf{z}_j) = 1$.

We denote $\boldsymbol{m}^t_{i \to j}$ to be all messages computed in iteration $t$ w.r.t. message functions $m^t_{i \to j}$ stacked up into a giant vector. Then we have

$$\boldsymbol{m}^0_{i \to j} \propto \mathbf{1}$$

and

$$\begin{aligned} \boldsymbol{m}^{t+1}_{i \to j} &= f \left( \mathbf{x}_i, \{\{\boldsymbol{m}^t_{k \to i} \mid k \in \mathcal{N}(i) \backslash j\}\} \right) \\ &= f' \left( \mathbf{x}_i, \boldsymbol{m}^t_{j \to i}, \{\{\boldsymbol{m}^t_{k \to i} \mid k \in \mathcal{N}(i)\}\} \right). \end{aligned} \tag{5}$$

Next, we show that all messages can be written as $\boldsymbol{m}^t_{i \to j} = \phi^t(\boldsymbol{c}_i, \boldsymbol{c}_j)$ where $\boldsymbol{c}^t_i$ is the color of node $i$ computed by 1-WL test. At iteration 0, clearly we have $\boldsymbol{m}^0_{i \to j} \propto \mathbf{1} = \phi^0(\boldsymbol{c}_i, \boldsymbol{c}_j)$ where for all $i \in \mathcal{V}$ (recall that $\mathcal{A}$ is within 1-WL's expressiveness). At iteration $t$, we have

$$\begin{aligned} \boldsymbol{m}^{t+1}_{i \to j} &= f' \left( \mathbf{x}_i, \boldsymbol{m}^t_{j \to i}, \{\{\boldsymbol{m}^t_{k \to i} \mid k \in \mathcal{N}(i)\}\} \right) \\ &= f' \left( \varphi(\boldsymbol{c}_i), \phi^t(\boldsymbol{c}_j, \boldsymbol{c}_i), \{\{\phi^t(\boldsymbol{c}_k, \boldsymbol{c}_i) \mid k \in \mathcal{N}(i)\}\} \right) \\ &= f'' \left( \boldsymbol{c}_i, \boldsymbol{c}_j, \{\{\boldsymbol{c}_k \mid k \in \mathcal{N}(i)\}\} \right). \end{aligned}$$

Substituting $\boldsymbol{c}^{t+1}_i = \mathrm{Hash}(\boldsymbol{c}^t_i, \{\{\boldsymbol{c}_j \mid j \in \mathcal{N}(i)\}\})$, we have

$$\boldsymbol{m}^{t+1}_{i \to j} = \phi^{t+1}(\boldsymbol{c}_i, \boldsymbol{c}_j).$$

The marginal distribution $q(\mathbf{z}_i \mid \mathbf{X})$ is then given by

$$q(\mathbf{z}_i \mid \mathbf{X}) \propto \Psi_1(\mathbf{z}_i, \mathbf{x}_i) \prod_{j \in \mathcal{N}(i)} m^t_{j \to i}(\mathbf{z}_i).$$

Similarly, we represent $q(\mathbf{z}_i \mid \mathbf{X})$ with a embedding $\boldsymbol{q}_i$, and

$$\begin{aligned} \boldsymbol{q}_i &= h(\{\{\boldsymbol{m}^t_{j \to i}\}\} \mid j \in \mathcal{N}(i)) \\ &= h(\{\{\phi^t(\boldsymbol{c}^t_j, \boldsymbol{c}^t_i) \mid j \in \mathcal{N}(i)\}\}) \\ &= h'(\boldsymbol{c}^{t+1}_i). \end{aligned}$$

As a conclusion, we have proved that the belief propagation update steps can be described by the 1-WL (color refinement), therefore any nodes that are distinguished by belief propagation can also be distinguished by a MPNN. □

## G.2 PROOF OF THEOREM 4

**Theorem 4.** *MPNNs can learn marginals that are at least as accurate as Bethe approximation. Formally, there exists decoding functions $f, g$ such that given any 1-WL distinguishable 2-order $\mathcal{A}$, for arbitrary $\epsilon > 0$ and $n \in \mathbb{N}$, there exists a MPNN such that for any graphs with no more than $n$ nodes, such that:*

- *$D_{\mathrm{KL}}(f(\boldsymbol{h}_i) \mid q_i) \leq \epsilon$ for $i \in \mathcal{V}$ and $D_{\mathrm{KL}}(g(\boldsymbol{h}_i, \boldsymbol{h}_j) \mid q_{ij}) \leq \epsilon$ for $(i,j) \in \mathcal{E}$, where $q_i, q_{ij}$ are node and edge marginals specified by a local extrema of Bethe approximation.*

*where $\boldsymbol{h}_i$ is the representation of node $i$ obtained by the MPNN, $D_{\mathrm{KL}}$ is the Kullback-Leibler divergence.*

*Proof.* Given a graph $G$, the conditional distribution is given by

$$p(\mathbf{Z} \mid \mathbf{X}) = \frac{1}{Z} \prod_{i \in \mathcal{V}} \Psi_i(\mathbf{z}_i, \mathbf{x}_i) \prod_{(i,j) \in \mathcal{E}} \Psi_{ij}(\mathbf{z}_i, \mathbf{z}_j).$$

Bethe approximation is defined by minimizing the corresponding free energy $D_{\mathrm{KL}}(q\|p)$, where we assume $q(\mathbf{Z}) = \prod_{i \in \mathcal{V}} q_i(\mathbf{z}_i) \prod_{(i,j) \in \mathcal{V}} \frac{q_{ij}(\mathbf{z}_i, \mathbf{z}_j)}{q_i(\mathbf{z}_i) q_j(\mathbf{z}_j)}$ and the Bethe free energy is

$$E_{Bethe} = \sum_{(i,j) \in \mathcal{E}} \sum_{\mathbf{z}_i, \mathbf{z}_j \in \mathcal{Z}} q_{ij}(\mathbf{z}_i, \mathbf{z}_j) \log \frac{q_{ij}(\mathbf{z}_i, \mathbf{z}_j)}{\Psi_{ij}(\mathbf{z}_i, \mathbf{z}_j)} - \sum_{i \in \mathcal{V}} (d_i - 1) \sum_{\mathbf{z}_i} q_i(\mathbf{z}_i) \log \frac{q_i(\mathbf{z}_i)}{\Psi_i(\mathbf{z}_i)},$$

where $q_i(\mathbf{z}_i) \geq 0, \sum_{\mathbf{z}_i} q_i(\mathbf{z}_i) = 1, q_{ij}(\mathbf{z}_i, \mathbf{z}_j) \geq 0, \sum_{\mathbf{z}_i} q_{ij}(\mathbf{z}_i, \mathbf{z}_j) = q_j(\mathbf{z}_j), \sum_{\mathbf{z}_j} q_{ij}(\mathbf{z}_i, \mathbf{z}_j) = q_i(\mathbf{z}_i)$, and $d_i$ is the degree of node $i$. Unluckily, the variational free energy is not convex, therefore finding the global extrema is hard.

We now describe a procedure to find local extrema of the Bethe free energy $E_{Bethe}$, which is further captured by MPNNs. The procedure is based on the concave-convex procedure (Yuille & Rangarajan, 2001), which states that by decomposing an energy function into a convex function and a concave function, we can use an discrete iteration procedure to monotonically decrease the energy and hence converge to a minimum or a saddle point.

We can decompose the Bethe free energy into convex and concave parts (Yuille, 2001):

$$E_{Bethe}^{vex} = \sum_{(i,j) \in \mathcal{E}} \sum_{\mathbf{z}_i, \mathbf{z}_j \in \mathcal{Z}} q_{ij}(\mathbf{z}_i, \mathbf{z}_j) \log \frac{q_{ij}(\mathbf{z}_i, \mathbf{z}_j)}{\Psi_{ij}(\mathbf{z}_i, \mathbf{z}_j)} + \sum_{i \in \mathcal{V}} \sum_{\mathbf{z}_i \in \mathcal{Z}} q_i(\mathbf{z}_i) \log \frac{q_i(\mathbf{z}_i)}{\Psi_i(\mathbf{z}_i)},$$

$$E_{Bethe}^{cave} = -\sum_{i \in \mathcal{V}} d_i \sum_{\mathbf{z}_i \in \mathcal{Z}} q_i(\mathbf{z}_i) \log \frac{q_i(\mathbf{z}_i)}{\Psi_i(\mathbf{z}_i, \boldsymbol{x}_i)}.$$

The double-loop algorithm for optimization is proposed by Yuille (2001), which is summarized as follows. The outer loop which has discrete time parameter $t$ is given by:

$$q_{ij}^{t+1}(\mathbf{z}_i, \mathbf{z}_j) = \Psi_{ij}(\mathbf{z}_i, \mathbf{z}_j) e^{-1} e^{\lambda_{ij}(\mathbf{z}_j)} e^{-\lambda_{ji}(\mathbf{z}_i)} e^{-\gamma_{ij}},$$

$$q_i^{t+1}(\mathbf{z}_i) = \Psi_i(\mathbf{z}_i, \boldsymbol{x}_i) e^{-1} e^{d_i} \left( \frac{q_i^t(\mathbf{z}_i)}{\Psi_i(\mathbf{z}_i, \boldsymbol{x}_i)} \right)^{d_i} e^{\sum_{k \in \mathcal{N}(i)} \lambda_{ki}(\mathbf{z}_i)}.$$

The $\{\gamma_{ij}\}, \{\lambda_{ij}\}, \{\lambda_{ji}\}$ are specified by the inner loop which has discrete time parameter $\tau$:

$$e^{\gamma_{ij}^{\tau+1}} = \sum_{\mathbf{z}_i, \mathbf{z}_j \in \mathcal{Z}} \Psi_{ij}(\mathbf{z}_i, \mathbf{z}_j) e^{-1} e^{-\lambda_{ij}^{\tau}(\mathbf{z}_j)} e^{-\lambda_{ji}^{\tau}(\mathbf{z}_i)},$$

$$e^{2\lambda_{ij}^{\tau+1}(\mathbf{z}_j)} = \frac{\sum_{\mathbf{z}_i} \Psi_{ij}(\mathbf{z}_i, \mathbf{z}_j) e^{-\lambda_{ji}^{\tau}(\mathbf{z}_i)} e^{-\gamma_{ij}^{\tau}}}{\Psi_j(\mathbf{z}_j, \boldsymbol{x}_j) e^{d_j \left( \frac{q_j^t(\mathbf{z}_j)}{\Psi_j(\mathbf{z}_j, \boldsymbol{x}_j)} \right)^{d_j}} e^{\sum_{k \in \mathcal{N}(j) \setminus i} \lambda_{kj}^{\tau}(\mathbf{z}_j)}}.$$

We now show that these iterations can be captured by MPNNs. At the beginning, we assume that we initialize $\{q_{ij}^0\}, \{q_i^0\}, \{\gamma_{ij}^0\}, \{\lambda_{ij}^0\}$ uniformly:

$$q_{ij}^0(\cdot, \cdot) \propto q_i^0(\cdot) \propto \gamma_{ij}^0 \propto \lambda_{ij}^0(\cdot, \cdot) \propto 1.$$

Since the random variables $\mathbf{z}_i \in \mathcal{Z}$ have finite states, the functions $q_{ij}, q_i, \lambda_{ij}$ over $\mathcal{Z}$ or $\mathcal{Z} \times \mathcal{Z}$ can be fully described by vectors. For example, if we assume $\mathcal{Z}$ has $K$ states, we can use a $K^2$-dimensional vector to represent the function $q_{ij}$ where $q_{ij}(p, q)$ corresponds to the $(pK + q)$-th entrance of the vector. We then design the MPNN as follows. The first layers of the MPNN are corresponded to the 1-WL coloring procedure. We add sufficient layers for all graphs of no more than $n$ nodes to obtain a stable node color. For ease of notation we denote the colors by $\boldsymbol{h}_i^0$ for all $i \in \mathcal{V}$. The parameters of the double-loop algorithm (at iteration 0) can be expressed as

$$q_{ij}^0 = g_2^0(\boldsymbol{h}_i^0, \boldsymbol{h}_j^0),$$
$$q_i^0 = g_1^0(\boldsymbol{h}_i^0),$$
$$\gamma_{ij}^0 = g_\gamma^0(\boldsymbol{h}_i^0, \boldsymbol{h}_j^0),$$
$$\lambda_{ij}^0 = g_\lambda^0(\boldsymbol{h}_i^0, \boldsymbol{h}_j^0),$$

where $g_2^0, g_1^0, g_\gamma^0, g_\lambda^0$ are some functions that maps the node representations $\boldsymbol{h}_i^0$ to the parameters in the double-loop algorithm.

For ease of notation we use a binary tuple $(t, \tau)$ to sequentially and uniquely represent the number of the iterations, where $t$ corresponds the current iteration of the outer loop and $\tau$ corresponds to the current iteration of the inner loop. We now prove that for both the inner loop and the outer loop, if at iteration $(t, \tau)$ we can express the parameters as follows:

$$q_{ij}^{t,\tau} = g_2^t(\boldsymbol{h}_i^{t,\tau}, \boldsymbol{h}_j^{t,\tau}),$$
$$q_i^{t,\tau} = g_1^t(\boldsymbol{h}_i^{t,\tau}),$$
$$\gamma_{ij}^{t,\tau} = g_\gamma^{t,\tau}(\boldsymbol{h}_i^{t,\tau}, \boldsymbol{h}_j^{t,\tau}),$$
$$\lambda_{ij}^{t,\tau} = g_\lambda^{t,\tau}(\boldsymbol{h}_i^{t,\tau}, \boldsymbol{h}_j^{t,\tau}),$$

then at iteration $(t+1, \tau)$ or $(t, \tau+1)$, we can still express the parameters in the same forms, where node representations are computed by a MPNN. Since the MRF is within 1-WL's discriminative power, the potential functions $\Psi_i, \Psi_{ij}$ can be computed by $\Psi_i = \Phi(\boldsymbol{h}_i^0)$ and $\Psi_{ij} = \Phi(\boldsymbol{h}_{ij}^0)$. For $q_{ij}^t$ in the outer loop, we thus have

$$q_{ij}^{t+1,\tau}(\mathbf{z}_i, \mathbf{z}_j) = \Psi_{ij}(\mathbf{z}_i, \mathbf{z}_j) e^{-1} e^{\lambda_{ij}^{t,\tau}(\mathbf{z}_j)} e^{-\lambda_{ji}^{t,\tau}(\mathbf{z}_i)} e^{-\gamma_{ij}}$$
$$= f(\boldsymbol{h}_i^{t,\tau}, \boldsymbol{h}_j^{t,\tau}),$$

where we only use $f$ to refer to node-irrelevant functions. For $q_i^t$ in the outer loop, we have

$$q_i^{t+1,\tau}(\mathbf{z}_i) = \Psi_i(\mathbf{z}_i, \boldsymbol{x}_i) e^{-1} e^{d_i} \left( \frac{q_i^{t,\tau}(\mathbf{z}_i)}{\Psi_i(\mathbf{z}_i, \boldsymbol{x}_i)} \right)^{d_i} e^{\sum_{k \in \mathcal{N}(i)} \lambda_{ki}^{t,\tau}(\mathbf{z}_i)}$$
$$= f(\boldsymbol{h}_i^{t,\tau}, \{\!\{ (\boldsymbol{h}_k^{t,\tau}, \boldsymbol{h}_i^{t,\tau}) \mid k \in \mathcal{N}(i) \}\!\})$$
$$= f(\boldsymbol{h}_i^{t+1,\tau}),$$

where $\boldsymbol{h}_i^{t+1,\tau} = \text{hash}(\boldsymbol{h}_i^{t,\tau}, \{\!\{ \boldsymbol{h}_j^{t,\tau} \mid j \in \mathcal{N}(i) \}\!\})$. For $\gamma_{ij}^{t,\tau+1}$ in the inner loop, we have

$$e^{\gamma_{ij}^{t,\tau+1}} = \sum_{\mathbf{z}_i, \mathbf{z}_j \in \mathcal{Z}} \Psi_{ij}(\mathbf{z}_i, \mathbf{z}_j) e^{-1} e^{-\lambda_{ij}^{t,\tau}(\mathbf{z}_j)} e^{-\lambda_{ji}^{t,\tau}(\mathbf{z}_i)},$$
$$= f(\boldsymbol{h}_i^{t,\tau}, \boldsymbol{h}_j^{t,\tau}).$$

For $\lambda_{ij}^{t,\tau+1}$ in the inner loop, we have

$$e^{2\lambda_{ij}^{t,\tau+1}(\mathbf{z}_j)} = \frac{\sum_{\mathbf{z}_i} \Psi_{ij}(\mathbf{z}_i, \mathbf{z}_j) e^{-\lambda_{ji}^{t,\tau}(\mathbf{z}_i)} e^{-\gamma_{ij}^{t,\tau}}}{\Psi_j(\mathbf{z}_j, \boldsymbol{x}_j) e^{d_j} \left( \frac{q_j^{t,\tau}(\mathbf{z}_j)}{\Psi_j(\mathbf{z}_j, \boldsymbol{x}_j)} \right)^{d_j} e^{\sum_{k \in \mathcal{N}(j) \setminus i} \lambda_{kj}^{t,\tau}(\mathbf{z}_j)}}$$
$$= f(\boldsymbol{h}_i^{t,\tau}, \boldsymbol{h}_j^{t,\tau}, \{\!\{ (\boldsymbol{h}_k^{t,\tau}, \boldsymbol{h}_j^{t,\tau}) \mid k \in \mathcal{N}(i) \setminus j \}\!\})$$
$$= f(\boldsymbol{h}_i^{t,\tau}, \boldsymbol{h}_j^{t,\tau}, \{\!\{ (\boldsymbol{h}_k^{t,\tau}, \boldsymbol{h}_j^{t,\tau}) \mid k \in \mathcal{N}(i) \}\!\}))$$
$$= f(\boldsymbol{h}_i^{t,\tau+1}),$$

where $\boldsymbol{h}_i^{t,\tau+1} = \text{hash}(\boldsymbol{h}_i^{t,\tau}, \{\{\boldsymbol{h}_j^{t,\tau} \mid j \in \mathcal{N}(i)\}\})$. Therefore, all the iterations in the double-loop algorithm can be described by iterations of MPNNs. Since the double-loop algorithm is proved to converge (Yuille, 2001), given any positive integer $n$, we can implement the algorithm with sufficient MPNN layers to guarantee that for all graphs of size less or equal than $n$, the error between the MPNN outputs and the true local extrema is arbitrary small. $\qquad\square$

### G.3 PROOF OF THEOREM 20

**Theorem 20.** *Given a 1-WL distinguishable 2-order $\mathcal{A}$, there is a MPNN layer $\mathcal{L}$ such that all fixed points of $\boldsymbol{H} = \mathcal{L}(\text{Concat}(\boldsymbol{H}, \boldsymbol{X}), \boldsymbol{A})$ implies fixed points of Bethe approximation of the MRF given any graph G, and all fixed points of Bethe approximation are implied in $\mathcal{L}$. That is, there exists a function $f$ such that given any graph G (of the size n), let $\mathcal{H}$ be the set of fixed points of $\mathcal{L}$ and $\mathcal{S}$ be the set of local extrema of Bethe approximation, then $f$ is always surjective from $\mathcal{H}$ to $\mathcal{S}$.*

*Proof.* The layer is implemented as

$$\boldsymbol{h}_i = \phi\left(\boldsymbol{h}_i, \boldsymbol{x}_i, \{\{\boldsymbol{h}_j \mid j \in \mathcal{N}(i)\}\}\right).$$

We need to add the node features $\boldsymbol{X}$ into the layer because otherwise the fixed points would be irrelevant w.r.t. $\boldsymbol{X}$. Without loss of generality we assume $\mathbf{Z} \in \mathcal{Z}^n$ comes from a discrete space, but the proof is equivalently applicable on continuous situations.

We begin by summarizing the Bethe approximation of the original distribution $p(\mathbf{Z} \mid \mathbf{X})$ given some fixed $\mathbf{X} \in \mathbb{R}^{n \times d}$. Recall that

$$p(\mathbf{Z} \mid \mathbf{X}) = \frac{1}{Z} \prod_{i \in \mathcal{V}} \Psi_i(\mathbf{z}_i, \mathbf{x}_i) \prod_{(i,j) \in \mathcal{E}} \Psi_{ij}(\mathbf{z}_i, \mathbf{z}_j) = \frac{1}{Z} \exp E(\mathbf{Z}).$$

By introducing

$$\mu(\mathbf{Z}) = \prod_{i \in \mathcal{V}} \mu_i(\mathbf{x}_i) \prod_{(i,j) \in \mathcal{E}} \frac{\mu_{ij}(\mathbf{z}_i, \mathbf{z}_j)}{\mu_i(\mathbf{z}_i)\mu_j(\mathbf{z}_j)},$$

where $\mu_i, \mu_{ij}$ are pseudo node marginals and edge marginals, we aim to minimize the Kullback-Leibler divergence $D_{\text{KL}}(\mu \| p)$ between $\mu$ and $p$. By further denoting

$$\mathcal{F}(\mu) = - \sum_{\mathbf{Z} \in \mathcal{Z}^n} \mu(\mathbf{Z})E(\mathbf{Z}) - \sum_{\mathbf{Z} \in \mathcal{Z}^n} \mu(\mathbf{Z}) \log \mu(\mathbf{Z})$$
$$= -\mathbb{E}_\mu[E(\mathbf{Z})] + \mathbb{E}_\mu[-\log \mu(\mathbf{Z})],$$

the Bethe variational problem for the log partition function then given as

$$\log Z_{bethe} = \max_\mu \mathcal{F}(\mu), \tag{6}$$

subject to:

$$\mu(\mathbf{Z}) = \prod_{i \in \mathcal{V}} \mu_i(\mathbf{x}_i) \prod_{(i,j) \in \mathcal{E}} \frac{\mu_{ij}(\mathbf{z}_i, \mathbf{z}_j)}{\mu_i(\mathbf{z}_i)\mu_j(\mathbf{z}_j)} \qquad \text{for all } \mathbf{Z} \in \mathcal{Z}^n$$

$$\mu_i(\mathbf{z}_i) \geq 0 \qquad \text{for all } i \in \mathcal{V}, \mathbf{z}_i \in \mathcal{Z}$$

$$\sum_{\mathbf{z}_i \in \mathcal{Z}} \mu_i(\mathbf{z}_i) = 1 \qquad \text{for all } i \in \mathcal{V}$$

$$\mu_i j(\mathbf{z}_i, \mathbf{z}_j) \geq 0 \qquad \text{for all } (i,j) \in \mathcal{E}, \mathbf{z}_i, \mathbf{z}_j \in \mathcal{Z}$$

$$\sum_{\mathbf{z}_j \in \mathcal{Z}} \mu_{ij}(\mathbf{z}_i, \mathbf{z}_j) = \mu_i(\mathbf{z}_i) \qquad \text{for all } (i,j) \in \mathcal{E}, \mathbf{z}_i \in \mathcal{Z}$$

$$\sum_{\mathbf{z}_i \in \mathcal{Z}} \mu_{ij}(\mathbf{z}_i, \mathbf{z}_j) = \mu_j(\mathbf{z}_j) \qquad \text{for all } (i,j) \in \mathcal{E}, \mathbf{z}_j \in \mathcal{Z}$$

To solve Eq. 7 we first unroll $\mathcal{F}(\mu)$ to be

$$
\begin{aligned}
\mathcal{F}(\mu) =& -\mathbb{E}_\mu[E(\mathbf{Z})] + \mathbb{E}_\mu[-\log\mu(\mathbf{Z})] \\
=& \mathbb{E}_\mu[\log Z + \log p(\mathbf{Z}\mid\mathbf{X})] + \mathbb{E}_\mu[-\log\mu(\mathbf{Z})] \\
=& \mathbb{E}_\mu\left[\sum_{i\in\mathcal{V}}(1-d_i)\log\Psi_i(\mathbf{z}_i,\mathbf{x}_i)\right] \\
& + \mathbb{E}_\mu\left[\sum_{(i,j)\in\mathcal{E}}(\log\Psi_{ij}(\mathbf{z}_i,\mathbf{z}_j) + \log\Psi_i(\mathbf{z}_i,\mathbf{x}_i) + \log\Psi_j(\mathbf{z}_j,\mathbf{x}_j))\right] \\
& - \mathbb{E}_\mu\left[\sum_{i\in\mathcal{V}}(1-d_i)\log\mu_i(\mathbf{z}_i) + \sum_{(i,j)\in\mathcal{E}}\log\mu_{ij}(\mathbf{z}_i,\mathbf{z}_j)\right] \\
=& \sum_{i\in\mathcal{V}}(1-d_i)\mathbb{E}_{\mu_i}[\log\Psi_i(\mathbf{z}_i,\mathbf{x}_i)] \\
& + \sum_{(i,j)\in\mathcal{E}}\mathbb{E}_{\mu_{ij}}[\log\Psi_{ij}(\mathbf{z}_i,\mathbf{z}_j)\log\Psi_i(\mathbf{z}_i,\mathbf{x}_i) + \log\Psi_j(\mathbf{z}_j,\mathbf{x}_j)] \\
& + \sum_{i\in\mathcal{V}}(1-d_i)\mathbb{E}_{\mu_i}[-\log\mu_i(\mathbf{z}_i)] + \sum_{(i,j)\in\mathcal{E}}\mathbb{E}_{\mu_{ij}}[-\log\mu_{ij}(\mathbf{z}_i,\mathbf{z}_j)] \\
=& \mathcal{F}_{bethe}(\mu) \\
=& \sum_{i\in\mathcal{V}}(1-d_i)\sum_{\mathbf{z}_i\in\mathcal{Z}}\mu_i(\mathbf{z}_i)[\log\Psi_i(\mathbf{z}_i,\mathbf{x}_i) - \log\mu_i(\mathbf{z}_i)] \\
& + \sum_{(i,j)\in\mathcal{E}}\sum_{\mathbf{z}_i,\mathbf{z}_j\in\mathcal{Z}}\mu_{ij}(\mathbf{z}_i,\mathbf{z}_j) \\
& \quad [\log\Psi_{ij}(\mathbf{z}_i,\mathbf{z}_j) + \log\Psi_i(\mathbf{z}_i,\mathbf{x}_i) + \log\Psi_j(\mathbf{z}_j,\mathbf{x}_j) - \log\mu_{ij}(\mathbf{z}_i,\mathbf{z}_j)],
\end{aligned}
$$

where $d_i$ is the degree of node $i$. Then, the problem 6 is equivalent to:

$$
\max_\mu \mathcal{F}_{bethe}(\mu) \tag{7}
$$

subject to:

$$
\begin{aligned}
\mu_i(\mathbf{z}_i) &\geq 0 && \text{for all } i\in\mathcal{V}, \mathbf{z}_i\in\mathcal{Z} \\
\sum_{\mathbf{z}_i\in\mathcal{Z}}\mu_i(\mathbf{z}_i) &= 1 && \text{for all } i\in\mathcal{V} \\
\mu_{ij}(\mathbf{z}_i,\mathbf{z}_j) &\geq 0 && \text{for all } (i,j)\in\mathcal{E}, \mathbf{z}_i,\mathbf{z}_j\in\mathcal{Z} \\
\sum_{\mathbf{z}_j\in\mathcal{Z}}\mu_{ij}(\mathbf{z}_i,\mathbf{z}_j) &= \mu_i(\mathbf{z}_i) && \text{for all } (i,j)\in\mathcal{E}, \mathbf{z}_i\in\mathcal{Z} \\
\sum_{\mathbf{z}_i\in\mathcal{Z}}\mu_{ij}(\mathbf{z}_i,\mathbf{z}_j) &= \mu_j(\mathbf{z}_j) && \text{for all } (i,j)\in\mathcal{E}, \mathbf{z}_j\in\mathcal{Z}
\end{aligned}
$$

We now introduce Lagrange multipliers to solve Eq. 7: $\lambda_i$ for constraints $\sum_{\mathbf{z}_i\in\mathcal{Z}}\mu_i(\mathbf{z}_i) = 1$, $\lambda_{ji}(\mathbf{z}_i)$ for constraints $\sum_{\mathbf{z}_j\in\mathcal{Z}}\mu_{ij}(\mathbf{z}_i,\mathbf{z}_j) = \mu_i(\mathbf{z}_i)$, and $\lambda_{ij}(\mathbf{z}_j)$ for constraints $\sum_{\mathbf{z}_i\in\mathcal{Z}}\mu_{ij}(\mathbf{z}_i,\mathbf{z}_j) = \mu_j(\mathbf{z}_j)$. The non-negative constraints are not active, therefore we do not introduce multipliers for these constraints. The Lagrangian is defined as

$$
\begin{aligned}
\mathcal{L}(\mu,\lambda) =& \mathcal{F}_{bethe}(\mu) + \sum_{i\in\mathcal{V}}\lambda_i\left(\sum_{\mathbf{z}_i\in\mathcal{Z}}\mu_i(\mathbf{z}_i) - 1\right) \\
& + \sum_{(i,j)\in\mathcal{E}}\sum_{\mathbf{z}_i\in\mathcal{Z}}\lambda_{ji}(\mathbf{z}_i)\left(\sum_{\mathbf{z}_j\in\mathcal{Z}}\mu_{ij}(\mathbf{z}_i,\mathbf{z}_j) - \mu_i(\mathbf{z}_i)\right) \\
& + \sum_{(i,j)\in\mathcal{E}}\sum_{\mathbf{z}_j\in\mathcal{Z}}\lambda_{ij}(\mathbf{z}_j)\left(\sum_{\mathbf{z}_i\in\mathcal{Z}}\mu_{ij}(\mathbf{z}_i,\mathbf{z}_j) - \mu_j(\mathbf{z}_j)\right).
\end{aligned}
$$

Since the local extrema of $\mathcal{L}$ corresponds to extrema of the problem 7, we now only need to prove that there exists a MPNN layer such that all the fixed points satisfying

$$\boldsymbol{h}_i = \phi\left(\boldsymbol{h}_i, \boldsymbol{x}_i, \{\!\{\boldsymbol{h}_j \mid j \in \mathcal{N}(i)\}\!\}\right)$$

corresponds to local extrema of the Lagrangian $\mathcal{L}$. This is to show that $\boldsymbol{h}_i$ contains sufficient statistic for $\mu_i$ for all $i \in \mathcal{V}$ and $(\boldsymbol{h}_i, \boldsymbol{h}_j)$ contains sufficient statistic for $\mu_{ij}$ for all $(i, j) \in \mathcal{E}$.

We begin by setting the derivatives of $\mathcal{L}(\mu, \lambda)$ to be 0. First we differentiate w.r.t. $\mu_i$:

$$\frac{\partial \mathcal{L}(\mu, \lambda)}{\partial \mu_i(\mathbf{z}_i)} = (1 - d_i)(\log \Psi_i(\mathbf{z}_i, \mathbf{x}_i) - \log \mu_i(\mathbf{z}_i) - 1) + \lambda_i - \sum_{j \in \mathcal{N}(i)} \lambda_{ji}(\mathbf{z}_i)$$

$$= -(d_i - 1) \log \Psi_i(\mathbf{z}_i, \mathbf{x}_i) + (d_i - 1)(\log \mu_i(\mathbf{z}_i) + 1) + \lambda_i - \sum_{j \in \mathcal{N}(i)} \lambda_{ji}(\mathbf{z}_i) = 0.$$

Therefore

$$\mu_i(\mathbf{z}_i) = \exp\left\{\log \Psi_i(\mathbf{z}_i, \mathbf{x}_i) + \frac{1}{d_i - 1}\left(\sum_{j \in \mathcal{N}(i)} \lambda_{ji}(\mathbf{z}_i)\lambda_i\right) - 1\right\}$$

$$= f_1\left(\mathbf{z}_i, \mathbf{x}_i, \lambda_i, \{\!\{\lambda_{ji} \mid j \in \mathcal{N}(i)\}\!\}, \mathcal{A}(i, G)\right),$$

where $d_i$ is implied in $\{\!\{\lambda_{ji} \mid j \in \mathcal{N}(i)\}\!\}$. Since $\mathcal{A}$ is within 1-WL expressiveness, by denoting $\boldsymbol{c}_i$ to be the 1-WL color of node $i$, we have

$$\mu_i = \mathcal{F}_1\left(\mathbf{x}_i, \lambda_i, \{\!\{\lambda_{ji} \mid j \in \mathcal{N}(i)\}\!\}, \boldsymbol{c}_i\right).$$

Next, we differentiate w.r.t. $\mu_{ij}$:

$$\frac{\partial \mathcal{L}(\mu, \lambda)}{\partial \mu_{ij}(\mathbf{z}_i, \mathbf{z}_j)} = \log \Psi_{ij}(\mathbf{z}_i, \mathbf{z}_j) + \log \Psi_i(\mathbf{z}_i, \mathbf{x}_i) + \log \Psi_j(\mathbf{z}_j, \mathbf{x}_j) - 1$$

$$- \log \mu_{ij}(\mathbf{z}_i, \mathbf{z}_j) + \lambda_{ji}(\mathbf{z}_i) + \lambda_{ij}(\mathbf{z}_j) = 0.$$

Therefore

$$\mu_{ij}(\mathbf{z}_i, \mathbf{z}_j) = \exp\left\{\log \Psi_{ij}(\mathbf{z}_i, \mathbf{z}_j) + \log \Psi_i(\mathbf{z}_i, \mathbf{x}_i) + \log \Psi_j(\mathbf{z}_j, \mathbf{x}_j) - 1 + \lambda_{ji}(\mathbf{z}_i) + \lambda_{ij}(\mathbf{z}_j)\right\}$$

$$= f_2\left(\mathbf{z}_i, \mathbf{z}_j, \mathbf{x}_i, \mathbf{x}_j, \lambda_{ij}, \lambda_{ji}, \mathcal{A}(\{i, j\}, G)\right),$$

and similarly

$$\mu_{ij} = \mathcal{F}_2(\mathbf{x}_i, \mathbf{x}_j, \lambda_{ij}, \lambda_{ji}, \boldsymbol{c}_i, \boldsymbol{c}_j).$$

Setting the derivatives w.r.t. $\lambda$ to 0 results in

$$1 = \mathcal{F}_3(\mu_i, \boldsymbol{c}_i),$$
$$\mu_i = \mathcal{F}_4(\mu_{ij}, \boldsymbol{c}_i, \boldsymbol{c}_j),$$
$$\mu_i = \mathcal{F}_5(\mu_{ji}, \boldsymbol{c}_j, \boldsymbol{c}_i),$$

We now parameterize $\mu_i, \mu_{ij}, \lambda_i, \lambda_{ij}$ as

$$\mu_i = \psi_{\mu_1}(\boldsymbol{h}_i),$$
$$\mu_{ij} = \psi_{\mu_2}(\boldsymbol{h}_i, \boldsymbol{h}_j),$$
$$\lambda_i = \psi_{\lambda_1}(\boldsymbol{h}_i),$$
$$\lambda_{ij} = \psi_{\lambda_2}(\boldsymbol{h}_i, \boldsymbol{h}_j),$$

where $\boldsymbol{h}_i$ is the node representation for node $i$. Note that for arbitrary choices of $\{\mu_i\}, \{\mu_{ij}\}, \{\lambda_i\}, \{\lambda_{ij}\}$ there must be a set of node representations $\{\boldsymbol{h}_i\}$ that express them. For example, we can set $\boldsymbol{h}_i[0] = i, \boldsymbol{h}_i[1] = \mu_i, \boldsymbol{h}_i[2] = \lambda_i, \boldsymbol{h}_i[2+j] = \mu_{ij}$ if $(i, j) \in \mathcal{E}, \boldsymbol{h}_i[2+n+j] = \lambda_{ij}$ if $(i, j) \in \mathcal{E}$, and all other dimensions of $\boldsymbol{h}_i$ being 0. Then we can express those terms via node representations. The above results are then written as

$$\psi_{\mu_1}(\boldsymbol{h}_i) = \mathcal{F}_1(\mathbf{x}_i, \psi_{\lambda_1}(\boldsymbol{h}_i), \{\!\{\psi_{\lambda_2}(\boldsymbol{h}_j, \boldsymbol{h}_i) \mid j \in \mathcal{N}(i)\}\!\}),$$
$$\psi_{\mu_2}(\boldsymbol{h}_i, \boldsymbol{h}_j) = \mathcal{F}_2(\mathbf{x}_i, \mathbf{x}_j, \psi_{\lambda_2}(\boldsymbol{h}_i, \boldsymbol{h}_j), \psi_{\lambda_2}(\boldsymbol{h}_j, \boldsymbol{h}_i)),$$
$$1 = \mathcal{F}_3(\psi_{\mu_1}(\boldsymbol{h}_i)),$$
$$\psi_{\mu_1}(\boldsymbol{h}_i) = \mathcal{F}_4(\psi_{\mu_2}(\boldsymbol{h}_i, \boldsymbol{h}_j)),$$
$$\psi_{\mu_1}(\boldsymbol{h}_i) = \mathcal{F}_5(\psi_{\mu_2}(\boldsymbol{h}_j, \boldsymbol{h}_i)),$$

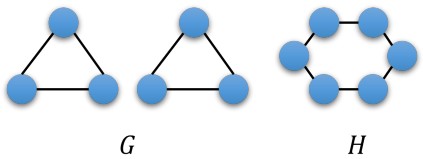

Figure 2: 1-WL undistinguished graphs

To summarize, these equations correspond to two properties: a node property where for all $i \in \mathcal{V}$,

$$\mathcal{M}_{node}(\boldsymbol{h}_i, \mathbf{x}_i, \{\!\{\boldsymbol{h}_j \mid j \in \mathcal{N}(i)\}\!\}) = 0,$$

and an edge property where for all $(i, j) \in \mathcal{E}$,

$$\mathcal{M}_{edge}(\boldsymbol{h}_i, \boldsymbol{h}_j, \mathbf{x}_i, \mathbf{x}_j) = 0$$

which can be equivalently described as for all $i \in \mathcal{V}$,

$$\mathcal{M}'_{edge}(\{\!\{(\boldsymbol{h}_i, \boldsymbol{h}_j, \mathbf{x}_i, \mathbf{x}_j) \mid j \in \mathcal{N}(i)\}\!\}) = 0.$$

By introducing constraints for all $i \in \mathcal{V}$, $\psi_x(\boldsymbol{h}_i) = \mathbf{x}_i$, we have

$$\mathcal{M}''_{edge}(\boldsymbol{h}_i, \{\!\{\boldsymbol{h}_j \mid j \in \mathcal{N}(i)\}\!\}) = 0.$$

Therefore, all constraints are specified by a MPNN layer. □

### G.4   PROOF OF COROLLOARY 5

**Corolloary 5.** *Given any graph $G$ and corresponding pairwise MRF, if $G$ is a tree, there is a MPNN that outputs true node and edge marginals.*

*Proof.* From Theorem 4 we know that MPNNs capture belief propagation. Since belief propagation can compute exact marginals on trees, so do MPNNs. □

### G.5   PROOF OF THEOREM 7

**Theorem 7.** *MPNNs can at most 1-2 approximate $p(\mathbf{z}_i \mid \mathbf{X})$ for arbitrary $G$ and $i \in \mathcal{V}_G$.*

*Proof.* We only need to show that MPNNs cannot 2-2 or 1-3 approximate $p(\mathbf{z}_i \mid \mathbf{X})$. For 2-2 approximations, recall that although 2-WL and 1-WL are equivalent in distinguishing graphs, they differ in distinguishing nodes. We set a graph $G$ to only contain one nodes, and set $H$ to only contain two isolated nodes. 1-WL always assigns all nodes the same color, but 2-WL assigns different color for nodes in $G$ and $H$. In this manner we can set a 2-WL distinguishable $\mathcal{A}$ to assign different potentials for nodes in $G$ and $H$, thus the posteriori are different.

Now we consider the 1-3 approximations. Consider the graphs in Figure 2. $G$ and $H$ are undistinguished by 1-WL, but we can set potential functions to be

$$\Psi_i(\mathbf{z}_i) \equiv \Psi_{ij}(\mathbf{z}_i, \mathbf{z}_j) \equiv 1,$$
$$\Psi_{ijk}(\mathbf{z}_i, \mathbf{z}_j, \mathbf{z}_k) = \begin{cases} 1, & \mathbf{z}_i = \mathbf{z}_j = \mathbf{z}_k = 0, \\ 0, & else. \end{cases}$$

for all $i, j, k$, where we let $\mathcal{Z} = \{0, 1\}$. Obviously due to the application of the 3-order potential function in $G$, the posteriori of nodes in $G$ is different from that in $H$. □

## H    PROOF W.R.T. APPROXIMATING COMPLEX DISTRIBUTIONS

### H.1    $k$-$l$-CWL: A TOOL FOR THE PROOF

In this section we will prove results for many GNN variants. For simplicity we first design a family of WL variants that provably approximate complex MRFs, then show that the GNN variants are expressive enough for capturing these WL variants.

We name our WL variants $k$-$l$ Clique WL (CWL). At the beginning, the $k$-$l$ CWL first apply classic $k$-WL on the graph and assign colors for $k$-tuples. Recall that for any subset of nodes $C = \{i_1, ... i_m\}$, $k$-WL computes the color of $C$ as $\mathrm{Col}^k(C) = \{\{c_v \mid v \in C^k\}\}$. At iteration 0, we initialize the colors of CWL as $c^0(i) = \mathrm{Hash}(x_i)$ for all $i \in \mathcal{V}$. At iteration $t + 1$, we have

$$c_i^{t+1} = \mathrm{Hash}\left(c_i^t, \left\{\left\{(c_j^t, \mathrm{Col}^k(\{i, j\})) \mid j \in \mathcal{N}_1(i)\right\}\right\}, ..., \left\{\left\{(c_{\mathbf{j}}^t, \mathrm{Col}^k(\mathbf{j} \cup \{i\})) \mid \mathbf{j} \in \mathcal{N}_l(i)\right\}\right\}\right),$$

where $\mathcal{N}_k(i) = \{\{j_1, ..., j_k\} \mid j_1, ..., j_l \in \mathcal{N}(i), (j_1, ..., j_l) \in \mathcal{C}\}$,
$c_{\{j_1, ..., j_l\}}^t = \{\{c_{j_1}^t, ..., c_{j_l}^t\}\}$, $\mathcal{C}$ is the set of cliques, $\mathrm{Col}^k$ is the colors specified by k-WL.

By parameterizing the hashing functions in $k$-$l$-CWL, we obtain the $k$-$l$-CWL-GNNs. Our next result confirms the expressiveness of $k$-$l$-CWL-GNNs.

**Theorem 21.** $k$-$l$-CWL-GNNs can $k$-$(l + 1)$ approximate $p(\mathbf{z}_i \mid \mathbf{X})$.

*Proof.* We first show that $k$-$l$-CWL is expressive enough for distinguishing non-isomorphic graphs compared with belief propagation. Belief propagation on a $(l + 1)$-order MRF is summarized as follows.

**Initialization.** We first transform the original graph $G$ into a factor graph $G_f$. For each term in the MRF formulation, we create a corresponding factor. That is, each clique $C$ of order no more than $l + 1$ is corresponded to a unique factor $c$, corresponding to the term $\Psi_C$. Further more, each node $i$ is connected with its own factor, corresponding to the term $\Psi_i$. In the factor graph, each node is connected and only connected factor nodes, thus composing a bipartite graph. We use $i, j, ...$ to refer to nodes and $a, b, c, ...$ to refer to factors. For all $(i, a) \in \mathcal{E}_{G_f}$ and $\mathbf{z}_i \in \mathcal{Z}$,

$$m_{a \to i}(\mathbf{z}_i) \propto 1 \propto m_{i \to a}(\mathbf{z}_i).$$

**Message passing.** $(t = 0, 1, 2, ...)$ For $(i, a) \in \mathcal{E}_{G_f}$,

$$m_{a \to i}^{t+1}(\mathbf{z}_i) = \sum_{\mathbf{z}_{\mathcal{N}(a)\backslash i}} \Psi_a(\mathbf{z}_i, \mathbf{z}_{\mathcal{N}(a)\backslash i}) \prod_{j \in \mathcal{N}(a)\backslash i} m_{j \to a}^t(\mathbf{z}_j),$$

$$m_{i \to a}^{t+1}(\mathbf{z}_i) = \prod_{b \in \mathcal{N}(i)\backslash a} m_{b \to i}^t(\mathbf{z}_i),$$

where $\Psi_a$ denotes the potential function w.r.t. the factor $a$. Specifically, when $a$ only contains a single node $i$, we let $m_{a \to i}^{t+1}(\mathbf{z}_i) = \Psi_i(\mathbf{z}_i, \mathbf{x}_i)$.

Similarly as before, if we use $\boldsymbol{m}_{a \to i}^t, \boldsymbol{m}_{i \to a}^t$ to represent sufficient statistics of the pseudo distributions $m_{a \to i}^t, m_{i \to a}^t$, the above iterations can be written as follows in the factor graph:

$$\boldsymbol{m}_{a \to i}^{t+1} = f(|\mathcal{N}(a)|, \mathbf{x}_i, \mathrm{Col}^k(\mathcal{N}(a)), \{\{\boldsymbol{m}_{j \to a}^t \mid j \in \mathcal{N}(a)\backslash j\}\})$$
$$= f'(\mathbf{x}_i, \boldsymbol{m}_{i \to a}^t, \mathrm{Col}^k(\mathcal{N}(a)), \{\{\boldsymbol{m}_{j \to a}^t \mid j \in \mathcal{N}(a)\}\}),$$
$$\boldsymbol{m}_{i \to a}^{t+1} = g(\{\{\boldsymbol{m}_{b \to i}^t \mid b \in \mathcal{N}(i)\backslash a\}\})$$
$$= g'(\boldsymbol{m}_{a \to i}^t, \{\{\boldsymbol{m}_{b \to i}^t \mid b \in \mathcal{N}(i)\}\}).$$

We first prove that this procedure can be captured by 1-WL on the factor graph, where we set the initial color of factors to be equivalent but different from the nodes. Next we show that all messages can be written as $\boldsymbol{m}_{a \to i}^t = \phi^t(c_a^t, c_i^t)$ and $\boldsymbol{m}_{i \to a}^t = \psi^t(c_i^t, c_a^t)$, where $c_i^t$ is the color of $k$-$l$-CWL for node $i$ in the original graph $G$ at iteration $t$ and $c_a^t$ for $a$ in the factor graph $G_f$ stands for

$\boldsymbol{c}_a^t = \{\{\boldsymbol{c}_j^t \mid j \in \mathcal{N}(a)\}\}$. At iteration $0$ clearly the statement is true. At iteration $t$ we assume it is still true. At iteration $t+1$, we have

$$
\begin{aligned}
\boldsymbol{m}_{a \to i}^{t+1} &= f'(\mathbf{x}_i, \mathrm{Col}^k(\mathcal{N}(a)), \boldsymbol{m}_{i \to a}^t, \{\{\boldsymbol{m}_{j \to a}^t \mid j \in \mathcal{N}(a)\}\}) \\
&= f''(\mathbf{x}_i, \mathrm{Col}^k(\mathcal{N}(a)), \phi^t(\boldsymbol{c}_i^t, \boldsymbol{c}_a^t), \{\{\phi^t(\boldsymbol{c}_j^t, \boldsymbol{c}_a^t) \mid j \in \mathcal{N}(a)\}\}) \\
&= f'''(\boldsymbol{c}_i^t, \mathrm{Col}^k(\mathcal{N}(a)), \boldsymbol{c}_a^t, \{\{\boldsymbol{c}_j^t \mid j \in \mathcal{N}(a)\}\}) \\
&= \phi^{t+1}(\boldsymbol{c}_a^{t+1}, \boldsymbol{c}_i^{t+1}). \\
\boldsymbol{m}_{i \to a}^{t+1} &= g'(\boldsymbol{m}_{a \to i}^t, \{\{\boldsymbol{m}_{b \to i}^t \mid b \in \mathcal{N}(i)\}\}) \\
&= g''(\psi^t(\boldsymbol{c}_a^t, \boldsymbol{c}_i^t), \{\{\psi^t(\boldsymbol{c}_b^t, \boldsymbol{c}_i^t) \mid b \in \mathcal{N}(i)\}\}) \\
&= \psi^{t+1}(\boldsymbol{c}_i^{t+1}, \boldsymbol{c}_a^{t+1}).
\end{aligned}
$$

Thus the statement holds. The last steps hold because

$$
\begin{aligned}
&\mathrm{Hash}(\boldsymbol{c}_i^t, \{\{\boldsymbol{c}_a^t \mid a \in \mathcal{N}(i)\}\}) \\
=&\mathrm{Hash}(\boldsymbol{c}_i^t, \{\{(\boldsymbol{c}_a^{t-1}, \{\{\boldsymbol{c}_j^{t-1} \mid j \in \mathcal{N}(a)\}\}) \mid a \in \mathcal{N}(i)\}\}) \\
=&\mathrm{Hash}(\boldsymbol{c}_i^t, \{\{(\boldsymbol{c}_a^{t-2}, \{\{\boldsymbol{c}_j^{t-2} \mid j \in \mathcal{N}(a)\}\}, \{\{\boldsymbol{c}_j^{t-1} \mid j \in \mathcal{N}(a)\}\}) \mid a \in \mathcal{N}(i)\}\}) \\
=& \cdots \\
=&\mathrm{Hash}\left(\boldsymbol{c}_i^t, \{\{(\{\{\boldsymbol{c}_j^0 \mid j \in \mathcal{N}(a)\}\}, ..., \{\{\boldsymbol{c}_j^{t-1} \mid j \in \mathcal{N}(a)\}\}) \mid a \in \mathcal{N}(i)\}\}\right) \\
=&\mathrm{Hash}\left(\boldsymbol{c}_i^t, \{\{\{\{\boldsymbol{c}_j^{t-1} \mid j \in \mathcal{N}(a)\}\} \mid a \in \mathcal{N}(i)\}\}\right) \\
=&\mathrm{Hash}\left(\boldsymbol{c}_i^t, \{\{\boldsymbol{c}_j^{t-1} \mid j \in \mathcal{N}_1(i)\}\}, ..., \{\{\boldsymbol{c}_j^{t-1} \mid \mathbf{j} \in \mathcal{N}_k(i)\}\}\right) \\
=&\mathrm{Hash}\left(\boldsymbol{c}_i^{t-1}, \{\{\boldsymbol{c}_j^{t-2} \mid j \in \mathcal{N}_1(i)\}\}, ..., \{\{\boldsymbol{c}_j^{t-2} \mid \mathbf{j} \in \mathcal{N}_k(i)\}\}, \right. \\
&\left. \{\{\boldsymbol{c}_j^{t-1} \mid j \in \mathcal{N}_1(i)\}\}, ..., \{\{\boldsymbol{c}_j^{t-1} \mid \mathbf{j} \in \mathcal{N}_k(i)\}\}\right) \\
=&\mathrm{Hash}\left(\boldsymbol{c}_i^{t-1}, \{\{\boldsymbol{c}_j^{t-1} \mid j \in \mathcal{N}_1(i)\}\}, ..., \{\{\boldsymbol{c}_j^{t-1} \mid \mathbf{j} \in \mathcal{N}_k(i)\}\}\right),
\end{aligned}
$$

where $\mathcal{N}_1, ..., \mathcal{N}_k$ follow the definitions given by $k$-CWL in the original graph. Therefore, the updates follow the $k$-CWL procedure. The marginal distribution $\boldsymbol{q}_i$ can be described as

$$
\begin{aligned}
\boldsymbol{q}_i &= \psi(\{\{\boldsymbol{m}_{a \to i}^t \mid a \in \mathcal{N}(i)\}\}) \\
&= \psi(\{\{\phi^t(\boldsymbol{c}_a^t, \boldsymbol{c}_i^t) \mid a \in \mathcal{N}(i)\}\}) \\
&= \psi'(\boldsymbol{c}_i^{t+1}).
\end{aligned}
$$

Thus the statement holds.

Now we show that $k$-$l$-CWL-GNNs can approximate a local extrema of the Bethe approximation to arbitrary precision. We will adopt the general Bethe free energy on factor graphs, which have been shown its connection with belief propagation on factor graphs (Heskes, 2002a; Yedidia et al., 2005), which is defined as follows:

$$
E_{Bethe} = \sum_{C \in \mathcal{C}} \sum_{\mathbf{z}_C} q_C(\mathbf{z}_C) \log \frac{q_C(\mathbf{z}_C)}{\Psi_C(\mathbf{z}_C)} - \sum_{i \in \mathcal{V}} (n_i - 1) \sum_{\mathbf{z}_i} q_i(\mathbf{z}_i) \log q_i(\mathbf{z}_i),
$$

where for simplicity we assume a single node $i$ also composes a clique $C$ and in that case $\Psi_C(\mathbf{z}_C)$ is shorthand for $\Psi_i(\mathbf{z}_i, \boldsymbol{x}_i)$. Again, there is a convex-concave algorithm for finding the local extrema of the Bethe approximation (Heskes, 2002a), where for the outer loop at iteration $(t+1, \tau)$ we have

$$
\log \Psi_a^{t+1, \tau}(\mathbf{z}_a) = \log \Psi_a^{t, \tau}(\mathbf{z}_a) + \sum_{i \in \mathcal{N}(a)} \frac{n_i - 1}{n_i} \log q_a^{t, \tau}(\mathbf{z}_a),
$$

$$
q_a^{t+1, \tau}(\mathbf{z}_a) = \frac{1}{Z_a} \Psi^{t+1, \tau}(\mathbf{z}_a) \prod_{i \in \mathcal{N}(a)} m_{i \to a}^{t, \tau}(\mathbf{z}_i),
$$

and for the inner loop at iteration $t, \tau + 1$ the messages $m$ are computed by regular belief propagation and

$$q_i^{t,\tau+1}(\mathbf{z}_i) = \frac{1}{Z_i} \left( \prod_{a \in \mathcal{N}(i)} m_{a \to i}^{t,\tau}(\mathbf{z}_i) \right)^{\frac{1}{n_i}},$$

$$q_a^{t,\tau+1}(\mathbf{z}_a) = \frac{1}{Z_a} \Psi_a^{t+1,\tau}(\mathbf{z}_a) \prod_{i \in \mathcal{N}(a)} m_{i \to a}^{t,\tau}(\mathbf{z}_i).$$

We now show that both the inner and outer loops are captured by $k$-$l$-CWL-GNNs. Previously we have shown that the messages of belief propagation can be captured by $k$-$l$-CWL-GNNs. By assuming

$$\Psi_a^{t,\tau} = g_\Psi^{t,\tau}(\{\{ c_i^{t,\tau} \mid i \in \mathcal{N}(a) \}\}),$$
$$q_a^{t,\tau} = g_C^{t,\tau}(\{\{ c_i^{t,\tau} \mid i \in \mathcal{N}(a) \}\}),$$
$$q_i^{t,\tau} = g_1^{t,\tau}(c_i^{t,\tau}),$$

we have

$$\log \Psi_a^{t+1,\tau}(\mathbf{z}_a) = \log \Psi_a^{t,\tau}(\mathbf{z}_a) + \sum_{i \in \mathcal{N}(a)} \frac{n_i - 1}{n_i} \log q_a^{t,\tau}(\mathbf{z}_a)$$

$$= f(\mathrm{Col}^k(\mathcal{N}(a)), \{\{ c_i^{t,\tau} \mid i \in \mathcal{N}(a) \}\})$$

$$= f(\{\{ c_i^{t,\tau+1} \mid i \in \mathcal{N}(a) \}\}),$$

$$q_a^{t,\tau+1}(\mathbf{z}_a) = \frac{1}{Z_a} \Psi_a^{t+1,\tau}(\mathbf{z}_a) \prod_{i \in \mathcal{N}(a)} m_{i \to a}^{t,\tau}(\mathbf{z}_i)$$

$$= f(\mathrm{Col}^k(\mathcal{N}(a)), \{\{ c_i^{t,\tau} \mid i \in \mathcal{N}(a) \}\})$$

$$= f(\{\{ c_i^{t,\tau+1} \mid i \in \mathcal{N}(a) \}\}),$$

$$q_i^{t,\tau+1}(\mathbf{z}_i) = \frac{1}{Z_i} \left( \prod_{a \in \mathcal{N}(i)} m_{a \to i}^{t,\tau}(\mathbf{z}_i) \right)^{\frac{1}{n_i}}$$

$$= f(\{\{ c_a^{t,\tau} \mid a \in \mathcal{N}(i) \}\}),$$

thus the iterations are also within $k$-$l$-CWL-GNNs' expressive power. $\square$

## H.2 Proof. w.r.t. $k$-GNNs

**Proposition 8.** *$k$-GNNs can $k$-$k$ approximate $p(\mathbf{z}_i \mid \mathbf{X})$ for arbitrary $G$.*

*Proof.* We prove this by showing $k$-GNNs can implement $k$-$(k-1)$-CWL-GNNs. We use $\mathbf{h}$ to stand for the embeddings of $k$-$(k-1)$-CWL-GNNs and $h$ for $k$-GNNs. We will reuse the symbol $f$ to represent arbitrary functions. We have

$$\mathbf{h}_i^{t+1} = f\left( \mathbf{h}_i^t, \left\{\left\{ (\mathbf{h}_j^t, \mathrm{Col}^k(\{i,j\})) \mid j \in \mathcal{N}_1(i) \right\}\right\}, ..., \left\{\left\{ (\mathbf{h}_\mathbf{j}^t, \mathrm{Col}^k(\mathbf{j} \cup \{i\})) \mid \mathbf{j} \in \mathcal{N}_{k-1}(i) \right\}\right\} \right).$$

We first show that $k$-GNNs can implement the following procedure

$$\mathbf{h}_i^{t+1} = f(\mathbf{h}_i^t, \left\{\left\{ (\mathbf{h}_\mathbf{j}^t, \mathrm{Col}^k(\mathbf{j} \cup \{i\}) \mid j \in \mathcal{N}_l(i)) \right\}\right\})$$

for any $l \le k-1$. By letting the corresponding $k$-GNN layers to be

$$h_{i,i,...,i}^\tau = f(h_{i,i,...,i}^{\tau-1}, \{\{ h_{j,i,...,i}^{\tau-1} \mid j \in \mathcal{V} \}\}, ..., \{\{ h_{i,i,...,j}^{\tau-1} \mid j \in \mathcal{V} \}\})$$

$$= f_0(h_{i,i,...,i}^{\tau-1}, \{\{ h_{i,j,...,i}^{\tau-1} \mid j \in \mathcal{V} \}\})$$

$$= f_1(h_{i,i,...,i}^{\tau-1}, \{\{ \{\{ h_{i,j_1,j_2,...,i}^{\tau-2} \mid j_2 \in \mathcal{V} \}\} \mid j_1 \in \mathcal{V} \}\})$$

$$= f_2(h_{i,i,...,i}^{\tau 1}, \{\{ h_{i,j_1,j_2,...,i}^{\tau-2} \mid j_1, j_2 \in \mathcal{V} \}\})$$

$$= ...$$

$$= f_\tau(h_{i,i,...,i}^{\tau-1}, \{\{ h_{i,j_1,j_2,...j_l,i}^{\tau-l} \mid j_1, j_2, ..., j_l \in \mathcal{V} \}\})$$

Note that since the embeddings of $k$-GNNs are initialized with internal structural information, we can define

$$\boldsymbol{h}_{i,i,...,i}^{\tau} = f_\tau(\boldsymbol{h}_{i,i,...,i}^{\tau-1}, \{\{\boldsymbol{h}_{i,j_1,j_2,...j_l,i}^{\tau-l} \mid j_1, j_2, ..., j_l \in \mathcal{V}\}\})$$
$$= f'(\boldsymbol{h}_{i,i,...,i}^{\tau-1}, \{\{\boldsymbol{h}_{i,j_1,j_2,...j_l,i}^{\tau-l} \mid \{j_1, ..., j_l\} \in \mathcal{N}_l(i)\}\}).$$

Thus any function $f'$ can be implemented by $k$-GNNs via backtrace of the above procedure. Since $\mathbf{h}_{i,j_1,...}^{\tau}$ are computed by $k$-GNNs it can also capture the $k$-WL colors with sufficient layers. Thus by using multiple layers together the $k$-GNN layers can implement the above procedure with

$$\mathbf{h}_i^t = \boldsymbol{h}_{i,i,...,i}^{\tau}$$

for some $\tau$. Obviously, the $k$-$(k-1)$-CWL-GNN layers are implemented by this procedure, thus $k$-GNNs implement $k$-$(k-1)$-CWL-GNNs. $\qquad\square$

### H.3 PROOF W.R.T. SUBGRAPH GNNS

**Proposition 9.** *ESANs with node marking policy can 1-3 approximate $p(\mathbf{z}_i \mid \mathbf{X})$ for arbitrary $G$.*

*Proof.* Recall that the update iterations in ESANs is called DSS-WL. Its node marking variant is implemented as:

$$\boldsymbol{c}_i^{t+1} = \text{Hash}(\boldsymbol{c}_i^{t+1}(j) \mid j \in \mathcal{V}),$$
$$\boldsymbol{c}_i^{t+1}(j) = \text{Hash}(\boldsymbol{c}_i^t(j), \{\{\boldsymbol{c}_k^t(j) \mid k \in \mathcal{N}(i)\}\}, \boldsymbol{c}_i^t, \{\{\boldsymbol{c}_k^t \mid k \in \mathcal{N}(i)\}\}),$$

with initialization

$$\boldsymbol{c}_i^0(j) = \begin{cases} (0, \boldsymbol{x}_i), & i \neq j, \\ (1, \boldsymbol{x}_i), & i = j. \end{cases}$$

We will focus on the 1-2-CWL-GNN variant, which is given as

$$\mathbf{c}_i^{t+1} = f(\mathbf{c}_i^t, \{\{\mathbf{c}_j^t \mid j \in \mathcal{N}_1(i)\}\}, \{\{\mathbf{c}_{j_1,j_2}^t \mid (j_1, j_2) \in \mathcal{N}_2(i)\}\}).$$

Note that we exclude the $\text{Col}^1$ terms in the original definition because since the 1-2-CWL already implies 1-WL, the colors of 1-WL can also be obtained by applying 1-2-CWL solely. We next show how we can implement 1-2-CWL-GNNs with ESANs. In the first iteration we can compute the colors of ESANs to be

$$\boldsymbol{c}_i^1(j) = (\mathbf{1}_{(i,j) \in \mathcal{E}}, \mathbf{1}_{i=j}, \boldsymbol{x}_i),$$

thus identifying the neighbors of each marked nodes. Then we can implement 1-2-CWL as

$$\begin{aligned} \boldsymbol{c}_i^{t+1} &= f(\boldsymbol{c}_i^t, \{\{\boldsymbol{c}_j^t \mid j \in \mathcal{N}_1(i)\}\}, \{\{\boldsymbol{c}_{j_1,j_2}^t \mid (j_1, j_2) \in \mathcal{N}_2(i)\}\}) \\ &= f_0(\boldsymbol{c}_i^t, \{\{\boldsymbol{c}_j^t \mid j \in \mathcal{N}(i)\}\}, \{\{\boldsymbol{c}_{j_1}^t(j_2) \mid (j_1, j_2), (i, j_1), (i, j_2) \in \mathcal{E}\}\}) \\ &= f_1(\boldsymbol{c}_i^t, \{\{\boldsymbol{c}_j^t \mid j \in \mathcal{N}(i)\}\}, \{\{\boldsymbol{c}_{j_1}^t(j_2) \mid j_1 \in \mathcal{N}(i), j_2 \in \mathcal{N}(i)\}\}) \\ &= f_2(\boldsymbol{c}_i^t, \{\{\boldsymbol{c}_j^t \mid j \in \mathcal{N}(i)\}\}, \{\{\{\{\boldsymbol{c}_{j_1}^t(j_2) \mid j_1 \in \mathcal{N}(i)\}\} \mid j_2 \in \mathcal{N}(i)\}\}) \\ &= f_3(\boldsymbol{c}_i^t, \{\{\boldsymbol{c}_j^t \mid j \in \mathcal{N}(i)\}\}, \{\{\boldsymbol{c}_i^{t+1}(j_2) \mid j_2 \in \mathcal{N}(i)\}\}) \\ &= f_4(\boldsymbol{c}_i^t, \{\{\boldsymbol{c}_j^t \mid j \in \mathcal{N}(i)\}\}, \{\{\boldsymbol{c}_i^{t+1}(j_2) \mid j_2 \in \mathcal{V}\}\}) \\ &= f_5(\{\{\boldsymbol{c}_i^{t+1}(j_2) \mid j_2 \in \mathcal{V}\}\}), \end{aligned}$$

where $\boldsymbol{c}_i^{t+1}(j) = \text{Hash}(\boldsymbol{c}_i^t(j), \{\{\boldsymbol{c}_k^t(j) \mid k \in \mathcal{N}(i)\}\}, \boldsymbol{c}_i^t, \{\{\boldsymbol{c}_k^t \mid k \in \mathcal{N}(i)\}\})$. Thus we can implement 1-2-CWL-GNN layers with ESANs and DSS-WL. $\qquad\square$

### H.4 PROOF W.R.T. SWL / CWL

**Proposition 10.** *The SWL / CWL and the corresponding GNN variants with $k$-clique simplex can 1-$k$ approximate $p(\mathbf{z}_i \mid \mathbf{X})$.*

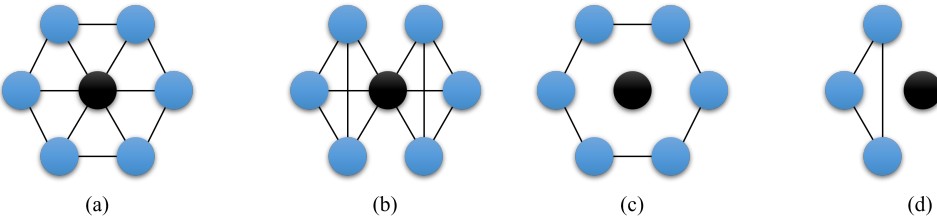

Figure 3: Counter example for ID-GNNs and Nested GNNs.

*Proof.* Considering the expressiveness of the two approaches (Bodnar et al., 2021a), we will focus on the less expressive SWL variant, together with its GNN variant, MPSN, which is given by

$$\boldsymbol{h}_\sigma^{t+1} = f(\boldsymbol{h}_\sigma^{t+1}, m_\mathcal{B}^{t+1}(\sigma), m_\mathcal{C}^{t+1}(\sigma), m_\downarrow^{t+1}(\sigma), m_\uparrow^{t+1}(\sigma)),$$

$$m_\mathcal{B}^{t+1}(\sigma) = \left\{\left\{ M_\mathcal{B}(\boldsymbol{h}_\sigma^t, \boldsymbol{h}_\tau^t) \mid \tau \in \mathcal{B}(\sigma) \right\}\right\},$$

$$m_\mathcal{C}^{t+1}(\sigma) = \left\{\left\{ M_\mathcal{C}(\boldsymbol{h}_\sigma^t, \boldsymbol{h}_\tau^t) \mid \tau \in \mathcal{C}(\sigma) \right\}\right\},$$

$$m_\uparrow^{t+1}(\sigma) = \left\{\left\{ M_\uparrow(\boldsymbol{h}_\sigma^t, \boldsymbol{h}_\tau^t, \boldsymbol{h}_{\sigma\cup\tau}^t) \mid \tau \in \mathcal{N}_\uparrow(\sigma) \right\}\right\},$$

$$m_\downarrow^{t+1}(\sigma) = \left\{\left\{ M_\downarrow(\boldsymbol{h}_\sigma^t, \boldsymbol{h}_\tau^t, \boldsymbol{h}_{\sigma\cap\tau}^t) \mid \tau \in \mathcal{N}_\downarrow(\sigma) \right\}\right\}.$$

For details of the definitions please refer to Bodnar et al. (2021b). The $k$-clique complex lifting is described as follows. Given a graph $G$, its corresponding simplicial complex $\mathcal{K}$ satisfies: if nodes $\{i_1, ..., i_l\}$ where $l \le k$ forms a clique in $G$, then $\{i_1, ..., i_l\} \in \mathcal{K}$.

We first show that in this case, nodes and cliques can interchange messages via finite of MPSN layers. Suppose $\sigma$ corresponds to a single node. Then by repeatedly applying the co-boundary adjacency $\mathcal{C}(\sigma)$, we obtain the edges, 3-cliques, ... that contains $\sigma$. For simplicity we refer to the set of edges, 3-cliques, ... as $\mathcal{C}(\sigma), \mathcal{C}^3(\sigma), ....$ Similarly, if $\sigma$ corresponds to a clique, then by repeatedly applying the boundary adjacency $\mathcal{B}(\sigma)$, we eventually obtain the nodes that it contains. We also refer to the set of nodes as $\mathcal{B}^0(\sigma)$ Thus we can implement 1-$(k-1)$-CWL with MPSNs as

$$\boldsymbol{h}_i^{t+1} = f(\boldsymbol{h}_i^t, \left\{\left\{ \boldsymbol{h}_j^t \mid j \in \mathcal{N}_1(i) \right\}\right\}, ..., \left\{\left\{ \boldsymbol{h}_\mathbf{j}^t \mid \mathbf{j} \in \mathcal{N}_{k-1}(i) \right\}\right\})$$

$$= f(\boldsymbol{h}_i^t, \left\{\left\{ (\boldsymbol{h}_i^t, \boldsymbol{h}_j^t) \mid j \in \mathcal{N}_1(i) \right\}\right\}, ..., \left\{\left\{ \boldsymbol{h}_{\mathbf{j}\cup\{i\}}^t \mid \mathbf{j} \in \mathcal{N}_{k-1}(i) \right\}\right\})$$

$$= f(\boldsymbol{h}_i^t, \left\{\left\{ \boldsymbol{h}_\sigma^t \mid \sigma \in \mathcal{C}(i) \right\}\right\}, ..., \left\{\left\{ \boldsymbol{h}_\sigma^t \mid \sigma \in \mathcal{C}^k(i) \right\}\right\}),$$

where $\boldsymbol{h}_\sigma^t = \mathcal{B}^0(\sigma)$. Thus the above procedure can be realized by finite iterations of MPSN layers. □

### H.5 PROOF W.R.T. OTHER VARIANTS

We now construct counter examples for these GNN variants.

**ID-GNNs.** Consider the graphs in Figure 3, where we use ID-GNNs to learn the black colored nodes. Obviously, ID-GNNs cannot distinguish the node in (a) with the node in (b), but clearly (a) has 6 3-cliques while (b) has 8 3-cliques, thus (a) and (b) are distinguishable by 3-order MRFs. To show that ID-GNNs is weaker than 2-WL, consider the graphs (c) and (d) in Figure 3. Since (c) has 7 nodes and (d) has 4 nodes, the black nodes are distinguished by 2-WL. However, since 1-WL only aggregates local neighbors, it cannot distinguish the black nodes in (c) and (d).

**Nested GNNs** We still consider the graphs in Figure 3, where we use Nested GNNs to learn the black colored nodes. For graphs (a) and (b), no matter what the number of hops we choose, NGNNs cannot distinguish the black nodes (a) between the black node in (b). Similarly, it also cannot distinguish the black nodes in (c) and (d).

## I  PROOF W.R.T. APPROXIMATING JOINT POSTERIORI

### I.1  $k$-NODE LABELING: A TOOL FOR PROOF

We first introduce the $k$-node labeling framework. Suppose we are to learn the posteriori $p(\mathbf{z}_{i_1}, ..., \mathbf{z}_{i_k} \mid \mathbf{X})$. We set augmented node features as

$$\tilde{\boldsymbol{x}}_i = (\mathbf{1}_{i=i_2}, ..., \mathbf{1}_{i=i_k}, \boldsymbol{x}_i),$$

Then we apply a MPNN on the augmented graph and output the node representation $\boldsymbol{h}_1$ as the final representation of the posteriori $p(\mathbf{z}_{i_1}, ..., \mathbf{z}_{i_k} \mid \mathbf{X})$. Then, we have

**Theorem 22.** *The $k$-node labeling with MPNNs 1-2 approximates $p(\mathbf{z}_{i_1}, ..., \mathbf{z}_{i_k} \mid \mathbf{X})$.*

*Proof.* As discussed above, we factorize $p(\mathbf{z}_{i_1}, ..., \mathbf{z}_{i_k} \mid \mathbf{X})$ as

$$p(\mathbf{z}_{i_1}, ..., \mathbf{z}_{i_k} \mid \mathbf{X}) = p(\mathbf{z}_{i_1} \mid \mathbf{z}_{i_2}, ..., \mathbf{z}_{i_k}, \mathbf{X}) \cdots p(\mathbf{z}_{i_k} \mid \mathbf{X}).$$

Thus to compute $p(\mathbf{z}_{i_1}, ..., \mathbf{z}_{i_k} \mid \mathbf{X})$ we need to repeatedly apply belief propagation (Bethe approximation) on the conditional MRFs. Concretely, to compute $p(\mathbf{z}_{i_1}, ..., \mathbf{z}_{i_k} \mid \mathbf{X})$ for all values of $\mathbf{z}_{i_1}, ..., \mathbf{z}_{i_k}$ we need to

$$\text{Compute } p(\mathbf{z}_{i_1} \mid \mathbf{z}_{i_2}, ..., \mathbf{z}_{i_k}, \mathbf{X}) \quad \text{for all } \mathbf{z}_{i_1}, ..., \mathbf{z}_{i_k} \in \mathcal{Z},$$
$$\text{Compute } p(\mathbf{z}_{i_2} \mid \mathbf{z}_{i_3}, ..., \mathbf{z}_{i_k}, \mathbf{X}) \quad \text{for all } \mathbf{z}_{i_2}, ..., \mathbf{z}_{i_k} \in \mathcal{Z},$$
$$\cdots$$
$$\text{Compute } p(\mathbf{z}_{i_k} \mid \mathbf{X}) \qquad\qquad\qquad \text{for all } \mathbf{z}_{i_k} \in \mathcal{Z}.$$

We now show that the $k$-node labeling can compute all the above functions. First consider computing $p(\mathbf{z}_{i_1} \mid \mathbf{z}_{i_2}, ..., \mathbf{z}_{i_k}, \mathbf{X})$. The original MRF can be written as

$$p(\mathbf{Z} \mid \mathbf{X}) = \frac{1}{Z} \prod_{i \in \mathcal{V}} \Psi_i(\mathbf{z}_i, \mathbf{x}_i) \prod_{(i,j) \in \mathcal{E}} \Psi_{ij}(\mathbf{z}_i, \mathbf{z}_j).$$

Fixing $\mathbf{z}_{i_2}, ..., \mathbf{z}_{i_k}$, we can rewrite the MRF as

$$
\begin{aligned}
p(\mathbf{Z}_{\backslash\{i_2,...,i_k\}} \mid \mathbf{z}_{i_2}, ..., \mathbf{z}_{i_k}, \mathbf{X}) =& \frac{1}{Z} \prod_{i \in \mathcal{V} \backslash \{i_2,...,i_k\}} \Psi_i(\mathbf{z}_i, \mathbf{x}_i) \prod_{(i,j) \in \mathcal{E}, i,j \notin \{i_2,...,i_k\}} \Psi_{ij}(\mathbf{z}_i, \mathbf{z}_j) \\
& \prod_{i \in \{i_2,...,i_k\}} \Psi_i(\mathbf{z}_i, \mathbf{x}_i) \prod_{i \in \{i_2,...,i_k\}, j \in \mathcal{N}(i)} \Psi_{ij}(\mathbf{z}_i, \mathbf{z}_j) \\
=& \frac{1}{Z'} \prod_{i \in \mathcal{V} \backslash \{i_2,...,i_k\}} \Psi_i(\mathbf{z}_i, \mathbf{x}_i) \prod_{(i,j) \in \mathcal{E}, i,j \notin \{i_2,...,i_k\}} \Psi_{ij}(\mathbf{z}_i, \mathbf{z}_j) \\
& \prod_{j \in \mathcal{N}(i_2)} \Psi_{i_2,j}(\mathbf{z}_{i_2}, \mathbf{z}_j) \cdots \prod_{j \in \mathcal{N}(i_k)} \Psi_{i_k,j}(\mathbf{z}_{i_k}, \mathbf{z}_j).
\end{aligned}
\tag{8}
$$

Assume each $\mathbf{z}_i \in \mathcal{Z}$ has $S$ states, then the above equations induce $S^{k-1}$ MRFs based on different values of $\mathbf{z}_{i_2}, ..., \mathbf{z}_{i_k}$. We notice that the above conditional MRFs can be regarded as normal MRFs on the modified graphs: given the original graph $G$, we first add a unique label $l_2$ for each neighbor of the node $i_2$, then delete $i_2$ from $G$; we then repeat this procedure for $i_3, ..., i_k$, obtaining the induced graph $\tilde{G}$. Then, there exists 1-WL distinguishable 2-order $\mathcal{A}_1, ..., \mathcal{A}_{S^{k-1}}$, such that the conditional MRFs on $G$ are given by $\mathcal{A}_1, ..., \mathcal{A}_{S^{k-1}}$ on $\tilde{G}$. This is simply done by setting

$$
\mathcal{A}_{s_2 + s_3 S + s_4 S^2 + ... + s_k S^{k-2}}(i, \tilde{G}) = \begin{cases} \Psi_i(\cdot) + \Psi_{i_m}(s_m, \cdot), & \text{node } i \text{ has label } l_m \text{ for } m \in [k], \\ \Psi_i(\cdot), & \text{else.} \end{cases}
$$
$$
\mathcal{A}_s(\{i, j\}, \tilde{G}) = \Psi_{ij}.
$$

According to our previous discussion on MPNNs and 1-WL expressive 2-order MRFs, MPNNs on $\tilde{G}$ can compute marginals on all $S^{k-1}$ MRFs. There's only one step before we prove the theorem. Note that after one iteration of MPNN on our $k$-node labeling framework, the neighbors of $\mathbf{z}_{i_2}, ..., \mathbf{z}_{i_k}$ are naturally tagged with unique labels, thus $\tilde{G}$ and our $k$-node labeling enhanced graph are equivalent for MPNNs. Thus, $k$-node labeling with MPNNs can also 1-2 approximates $p(\mathbf{z}_{i_1} \mid \mathbf{z}_{i_2}, ..., \mathbf{z}_{i_k}, \mathbf{X})$. Since computing $p(\mathbf{z}_{i_2} \mid \mathbf{z}_{i_3}, ..., \mathbf{z}_{i_k}, \mathbf{X}), ...$ only requires performing node labeling on $i_3, ..., i_k, ...$, the $k$-node labeling can also approximate them. Putting these together, the $k$-node labeling with MPNNs 1-2 approximates $p(\mathbf{z}_{i_1}, ..., \mathbf{z}_{i_k} \mid \mathbf{X})$. $\qquad\square$

### I.2    PROOF W.R.T. $k$-GNNS

**Proposition 12.** *$k$-GNNs cannot 1-2 approximate $p(\mathbf{z}_{i_1}, ..., \mathbf{z}_{i_k} \mid \mathbf{X})$ for arbitrary $k \geq 2$.*

*Proof.* The proof is similar with the proof in (Qian et al., 2022), where they showed that $k$-node labeling is not weaker than $k$-WL. The core is to construct graphs that are undistinguished by $k$-WL but are distinguished by $(k-1)$-node labeling. We construct CFI graphs $\mathcal{X}(\mathcal{K})$ and $\widehat{\mathcal{X}}(\mathcal{K})$. From Grohe & Otto (2012) we know that $\mathcal{X}(\mathcal{K})$ and $\widehat{\mathcal{X}}(\mathcal{K})$ cannot be distinguished by $(k-1)$-WL, whose expressive power is bounded by $(k-2)$-FWL.

Next we show that $\mathcal{X}(\mathcal{K})$ and $\widehat{\mathcal{X}}(\mathcal{K})$ can be distinguished by $(k-2)$-node labeling. To do so we need to introduce the concept of *pebble games* (Grohe & Otto, 2012; Cai et al., 1989). The readers are welcome to check Grohe & Otto (2012) for a more detailed introduction. Given two structures $\mathcal{A}, \mathcal{B}$, the *bijective $k$-pebble game* is played by two players by placing $k$ pairs of pebbles on a pair of structures $\mathcal{A}, \mathcal{B}$. The rounds of the game are as follows. Player **I** picks up one of his pebbles, and player **II** picks up her corresponding pebble. Then player **II** chooses a bijection $f$ between $\mathcal{A}$ and $\mathcal{B}$ (if no such bijection exists player **II** immediately loses). Then player **I** places his pebble on an element $a$ of $\mathcal{A}$, and player **II** places her pebble on $f(a)$. After each round there is a subset $p \subseteq \mathcal{A} \times \mathcal{B}$ consisting of the at most $k$ pairs of elements corresponding to the pebbles placed. Player **II** wins a play if every position $p$ is a local isomorphism.

**Theorem 23.** *(Cai et al. (1989))* $\mathcal{A} \equiv_C^k \mathcal{B}$ *if and only if player **II** has a winning strategy for the bijective $k$-pebble game on $\mathcal{A}, \mathcal{B}$.*

Theorem 23 indicates that $(k-1)$-FWL can distinguish $\mathcal{A}, \mathcal{B}$ if and only if player **II** has a winning strategy for the bijective $k$-pebble game on $\mathcal{A}, \mathcal{B}$. However, although it justifies $(k-1)$-FWL, it has nothing to do with the $(k-2)$-node labelind here. We propose a variant of the bijective $k$-pebble game, namely restricted bijective $k$-$i$-pebble game, described as follows. Given two structures $\mathcal{A}, \mathcal{B}$, the *restricted bijective $k$-$i$-pebble game* is also played by two players by placing $k$ pairs of pebbles on a pair of structures $\mathcal{A}, \mathcal{B}$. The rounds of the game are as follows. Player **I** picks up one of his pebbles, and player **II** picks up her corresponding pebble. Then player **II** chooses a bijection $f$ between $\mathcal{A}$ and $\mathcal{B}$ (if no such bijection exists player **II** immediately loses). Then player **I** places his pebble on an element $a$ of $\mathcal{A}$, and player **II** places her pebble on $f(a)$. The difference is, $i$ pairs of the pebbles are static, as when the players place these pebbles on the elements of $\mathcal{A}, \mathcal{B}$, they can no longer pick and replace them on other elements. Then, we have

**Theorem 24.** *There is a $(k-2)$-node labeling MPNN that distinguishes the $(k-2)$-tuples $\mathbf{u}, \mathbf{v}$ from $\mathcal{A}, \mathcal{B}$ if and only if player **II** has a winning strategy for the restricted bijective k-(k-2)-pebble game on $\mathcal{A}, \mathcal{B}$ which initially places $(k-2)$ static pebbles on $\mathbf{u}, \mathbf{v}$, if $\mathcal{A}, \mathcal{B}$ are connected graphs.*

*Proof.* Let $G, H$ be a pair of connected graphs and let $\mathbf{u}, \mathbf{v}$ be the target $k$-tuples. Let $G_{(\mathbf{u})}, H_{(\mathbf{v})}$ be the corresponding node labeling induced graphs. Since $G, H$ are connected, we need to show the following statements are equivalent.

1. 1-WL cannot distinguish $G_{(\mathbf{u})}, H_{(\mathbf{v})}$

2. 2-WL cannot distinguish $G_{(\mathbf{u})}, H_{(\mathbf{v})}$

3. No FOC$_2$ formula distinguishes $G_{(\mathbf{u})}, H_{(\mathbf{v})}$

4. Player **II** has a winning strategy for the $(k+2)$-$k$ pebble game on $G, H$ which initially places the $k$ pairs of static pebbles on $\mathbf{u}, \mathbf{v}$.

$1 \Longleftrightarrow 2 \Longleftrightarrow 3$: $2 \Longleftrightarrow 3$ is proved as a special case in Cai et al. (1989). Since $G, H$ are connected, $1 \Longleftrightarrow 2$ also holds.

$2 \Rightarrow 4$: Suppose after $r$ iterations the 2-WL still assigns the same color to $G_{(\mathbf{u})}, H_{(\mathbf{v})}$. We instead prove the following statement:

- After $r + k$ iterations 2-WL gives $(x_1, y_1) \in \mathcal{V}^2_{G_{(\mathbf{u})}}$ and $(x_2, y_2) \in \mathcal{V}^2_{H_{(\mathbf{v})}}$ the same color $\implies$ Player **II** has a winning strategy for the $(k+2)$-$k$ pebble game on $G, H$ which initially places the $k$ pairs of static pebbles on $\mathbf{u}, \mathbf{v}$ and place the other 2 pairs of static pebbles on $(x_1, y_1)$ and $(x_2, y_2)$ in $r$ moves.

We assume $W^r$ to be the color assignment of 2-WL at iteration $r + k$. Clearly player **I** can only chooses one of the pebbles on $x_1, y_1$. Without loss of generality suppose he picks up $x_1$. The player **II** answers with the bijective mapping that maps node pairs with the same $W^{r-1}$ color, that is, $f(t_1) \in \{t_2 \mid W^{r-1}(t_1, y_1) = W^{r-1}(t_2, y_2)\}$. Note that such mapping must exist because

$$W^r(x, y) = \text{Hash}\left(W^{r-1}(x, y), \{\!\{W^{r-1}(x, z) \mid z \in \mathcal{V}_G\}\!\}, \{\!\{W^{r-1}(z, y) \mid z \in \mathcal{V}_G\}\!\}\right)$$

is the same for $(x_1, y_1)$ and $(x_2, y_2)$. No matter which node player **I** places his pebble on, player **II** places her pebble on the corresponding node. Player **II** has not yet lost: the structure between $(t_1, y_1)$ and $(t_2, y_2)$ must be the same, otherwise they have different $W^{r-1}$ colors. The structure between $(t_1, \mathbf{u})$ and $(t_2, \mathbf{v})$ are also the same: This is because that at the start we added unique labels to the $k$-tuples $\mathbf{u}$ and $\mathbf{v}$. Therefore, after the first $k$ rounds of 2-WL iterations if the subgraphs induced by $t_1, y_1, \mathbf{u}$ from $G_{(\mathbf{u})}$ and $t_2, y_2, \mathbf{v}$ from $H_{(\mathbf{v})}$ are different, $(t_1, y_1)$ and $(t_2, y_2)$ will also have different 2-WL colors. Now, since $(t_1, y_1)$ and $(t_2, y_2)$ have the same $W^{r-1}$ colors, by induction on $r$ we proved the above statement.

By further induction on $r$ in the statement, we can prove 2$\Rightarrow$4 because we have showed that for any node pairs the results of 2-WL and the pebble game are always consistent.

$\neg 3 \Rightarrow \neg 4$: Suppose for some $\text{FOC}_2$ formula $\varphi$, $G_{(\mathbf{u})}\varphi$ and $H_{(\mathbf{v})}\varphi$. If $\varphi$ is a conjunction then $G_{(\mathbf{u})}, H_{(\mathbf{v})}$ must differs on at least one of the conjuncts, so we may assume $\varphi$ is of the form $\exists^N x\psi$. Without loss of generality we assume the quantifier depth of $\varphi$ is $r$. Note that there are total 2 free pairs of pebbles that can be placed to nodes, which exactly corresponds to the number of free variables in $\text{FOC}_2$ formula. Player **I** takes a pebble, corresponding to the variable $x$ in $\varphi$. Player **II** must respond with a bijective mapping $f$. Since $G_{(\mathbf{u})}\varphi$ and $H_{(\mathbf{v})}\varphi$, we know that there are at least $N$ nodes satisfying $\psi$ in $G_{(\mathbf{u})}$ but less than $N$ nodes satisfying $\psi$ in $H_{(\mathbf{v})}$. Player **I** then picks the node $w$ in $G_{(\mathbf{u})}$ such that $\psi(w)$ is true but $\psi(f(w))$ is false. By induction we can see that at quantifier depth 0 player **II** loses the game. $\qquad\square$

With Theorem 24 we can now prove that the graphs $\mathcal{X}(\mathcal{K})$ and $\widehat{\mathcal{X}}(\mathcal{K})$ can be distinguished by $(k-2)$-node labeling. The prove steps are exactly the same as in Grohe & Otto (2012), as the steps in Grohe & Otto (2012) naturally follow the constraints of the bijective $k$-$(k-2)$-pebble game.

We give a winning strategy for player **I** in the bijective $k$-$(k-2)$-pebble game. In the first $k-1$ rounds of the game, player **I** picks his $k-1$ pebbles on $v_2^\emptyset, ..., v_k^\emptyset$ and suppose $p(v_i^\emptyset) = v_i^{S_i}$ is the corresponding position for some sets $S_i$. That is,

$$p = \{v_2^\emptyset v_2^{S_2}, ..., v_k^\emptyset v_k^{S_k}\}.$$

We now assume that the pebbles at $v_3^\emptyset, ..., v_k^\emptyset$ are static. Therefore $v_3^\emptyset, ..., v_k^\emptyset$ compose all $k-2$ static pebbles and the pebble at $v_2^\emptyset$ is still movable. In the next round of the game player **I** starts by selecting this pebble, and places it on $e_{12}^0$. In the next round, player **I** starts by selecting the pebble on $e_{12}^0$ and places it on $v_1^\emptyset$. It is proved by Grohe & Otto (2012) that player **I** wins at this time. $\qquad\square$

## I.3 Proof w.r.t. $k$-FWL-MPNNs

**Proposition 13.** *$k$-FWL-MPNNs can 1-2 approximate $p(\mathbf{z}_{i_1}, ..., \mathbf{z}_{i_k} \mid \mathbf{X})$.*

*Proof.* Given $G = (\mathcal{V}, \mathcal{E})$, the $k$-FWL at each layer evaluates

$$\text{Col}^{(l+1)}(u_1, ..., u_k) = \text{Hash}\Big(\text{Col}^{(l)}(u_1, ..., u_k),$$
$$\{\!\{\Big(\text{Col}^{(l)}(v, u_2, ..., u_k), \text{Col}^{(l)}(u_1, v, ..., u_k), ..., \text{Col}^{(l)}(u_1, ..., u_{k-1}, v)\Big) \mid v \in \mathcal{V}\}\!\}\Big).$$

If we consider a fragment of the above procedure where we replace the last colors in each neighbor with the initial colors

$$
\mathrm{Col}^{(l+1)}(u_1, ..., u_k) = \mathrm{Hash}\Big(\mathrm{Col}^{(l)}(u_1, ..., u_k),
$$

$$
\Big\{\Big\{\Big(\mathrm{Col}^{(l)}(v, u_2, ..., u_k), \mathrm{Col}^{(0)}(u_1, v, ..., u_k), ..., \mathrm{Col}^{(0)}(u_1, ..., u_{k-1}, v)\Big) \mid v \in \mathcal{N}(u_1)\Big\}\Big\}\Big).
$$

Note that the neighbors $\mathcal{N}(u_1)$ are already encoded in $\mathrm{Col}^{(0)}(u_1, v, ...)$. Clearly, this variant is less expressive than the original one. We can rewrite this variant to make it strictly corresponded to $k - 1$-node labeling as

$$
f^{(l+1)}(u) = \phi\left(f^{(l)}(u), \left\{\left\{f^{(l)}(v) \mid v \in \mathcal{V}\right\}\right\}\right),
$$

where we initialize $f^{(0)}$ as $f^{(0)}(v) = \mathrm{Col}^{(0)}(v, u_2, ..., u_k)$. The computation of $f^{(l)}$ is exactly the same as $(k - 1)$-node labeling when we apply node labeling on $u_2, ..., u_k$. Since $f$ is clearly corresponded to a smaller fragment of the above equations compared with $k$-FWL, we have proved the result. $\qquad\square$

### I.4 PROOF W.R.T. LABELING TRICK

**Proposition 14.** *$k$-labeling trick MPNNs 1-2 approximate $f(\boldsymbol{z}) = p(\mathbf{z}_{i_1} = \cdots \mathbf{z}_{i_k} = \boldsymbol{z} \mid \mathbf{X})$ but not $p(\mathbf{z}_i, \mathbf{z}_j \mid \mathbf{X})$.*

*Proof.* It's simple to show that labeling trick cannot capture $p(\mathbf{z}_i, \mathbf{z}_j \mid \mathbf{X})$. Consider a graph $G$ with two isolated nodes $1, 2$ with different node features. Then obviously we can have $p(\mathbf{z}_1, \mathbf{z}_2 \mid \mathbf{X}) \neq p(\mathbf{z}_2, \mathbf{z}_1 \mid \mathbf{X})$ but labeling trick will always assign them the same representation.

Next we show that $k$-labeling trick MPNNs 1-2 approximate $f(\boldsymbol{z}) = p(\mathbf{z}_{i_1} = \cdots \mathbf{z}_{i_k} = \boldsymbol{z} \mid \mathbf{X})$. This is done by altering the MRFs in Eq. 8. We can rewrite the MRFs as

$$
p(\mathbf{Z}_{\backslash\{i_1, i_2, ..., i_k\}} \mid \mathbf{z}_{i_2} = ... = \mathbf{z}_{i_k} = \boldsymbol{z}, \mathbf{X}) = \frac{1}{Z} \prod_{i \in \mathcal{V}\backslash\{i_1, ..., i_k\}} \Psi_i(\mathbf{z}_i, \mathbf{x}_i) \prod_{(i,j) \in \mathcal{E}, i, j \notin \{i_1, ..., i_k\}} \Psi_{ij}(\mathbf{z}_i, \mathbf{z}_j)
$$

$$
\prod_{j \in \mathcal{N}(i_1)} \Psi_{i_2, j}(\boldsymbol{z}, \mathbf{z}_j) \cdots \prod_{j \in \mathcal{N}(i_k)} \Psi_{i_k, j}(\boldsymbol{z}, \mathbf{z}_j)
$$

Thus $f(\boldsymbol{z})$ is computed by only reserving the MRFs specified by $\mathcal{A}_s$ for $s \in \mathcal{Z}$ (for simplicity we assume $\mathcal{Z} = [S]$) where we add an identical label to $i_2, ..., i_k$ and set

$$
\mathcal{A}_s(i, \tilde{G}) = \begin{cases} \Psi_i(\cdot) + \Psi_{ij}(s, \cdot), & j \text{ is labeled}, \\ \Psi_i(\cdot), & \text{else}. \end{cases}
$$

$$
\mathcal{A}_s(\{i, j\}, \tilde{G}) = \begin{cases} \Psi_{ij}, & i, j \text{ are not labeled}, \\ \mathbf{1}, & \text{else}. \end{cases}
$$

This identical labeling procedure is captured by the $k$-labeling trick, thus the representation of node $i$ in $k$-labeling trick can capture $p(\mathbf{z}_i \mid \mathbf{z}_{i_1} = ... = \mathbf{z}_{i_k} = \boldsymbol{z}, \mathbf{X})$ and similarly as our discussion before, $p(\mathbf{z}_i, \mathbf{z}_{i_1} = ... = \mathbf{z}_{i_k} = \boldsymbol{z}, \mathbf{X})$. Since $k$-labeling trick aggregates all nodes in the graph, the graph representation can capture $p(\mathbf{z}_{i_1} = ... = \mathbf{z}_{i_k} = \boldsymbol{z}, \mathbf{X})$. $\qquad\square$

### I.5 PROOF W.R.T. ORDERED NODE LABELING

**Proposition 15.** *MPNNs with ordered node pair labeling 1-2 approximate $p(\mathbf{z}_i, \mathbf{z}_j \mid \mathbf{X})$.*

*Proof.* To approximate $p(\mathbf{z}_i, \mathbf{z}_j \mid \mathbf{X})$ we only need to add an label on node $j$. Since ordered node pair labeling adds labels on both $i$ and $j$, it can also approximate $p(\mathbf{z}_i, \mathbf{z}_j \mid \mathbf{X})$. $\qquad\square$

### I.6 PROOF W.R.T. SOURCE NODE LABELING

**Proposition 16.** *Source node labeling MPNNs 1-2 approximate $p(\mathbf{z}_i, \mathbf{z}_j \mid \mathbf{X})$.*

*Proof.* The source node labeling is exactly 1-node labeling applied on $j$. $\qquad\square$

# J  PROOF W.R.T. PHANTOM NODES / EDGES

## J.1  PROOF OF PROPOSITION 17

**Proposition 17.** *MPNNs with phantom nodes can 1-$\infty$ approximate $p(\mathbf{z}_i \mid \mathbf{X})$.*

*Proof.* We prove by showing that for any $n \in \mathbb{N}_+$, phantom nodes can capture 1-$n$-CWL-GNNs for graphs with no more than $n$ nodes. To separate our notations for nodes and phantom nodes, we will use $i, j, ...$ to refer to nodes and $p, q, ...$ to refer to phantom nodes. We next show that two layers of phantom node enhanced MPNNs can capture one layer of 1-$n$-CWL-GNNs:

$$
\begin{aligned}
\boldsymbol{c}_i^{t+1} &= f\left(\boldsymbol{c}_i^t, \{\{(\boldsymbol{c}_j^t \mid j \in \mathcal{N}_1(i)\}\}, ..., \{\{(\boldsymbol{c}_{\mathbf{j}}^t \mid \mathbf{j} \in \mathcal{N}_n(i)\}\}\right) \\
&= f_0\left(\boldsymbol{c}_i^t, \{\{(\boldsymbol{c}_{\mathbf{j}}^t \mid \mathbf{j} \in \mathcal{N}_1(i) \cup ... \cup \mathcal{N}_n(i)\}\}\right)
\end{aligned}
$$

Since smaller cliques are completely contained in larger cliques, we can also only preserve *maximum cliques* in the above equation:

$$
\begin{aligned}
\boldsymbol{c}_i^{t+1} &= f_1\left(\boldsymbol{c}_i^t, \{\{(\boldsymbol{c}_{\mathbf{j}}^t \mid (\mathbf{j} \cup \{i\}) \in \mathcal{C}\}\}\right) \\
&= f_2\left(\boldsymbol{c}_i^t, \{\{(\boldsymbol{c}_p^t \mid p \in \mathcal{N}(i)\}\}\right),
\end{aligned}
$$

where we further define for phantom nodes

$$
\boldsymbol{c}_p^t = \{\{\boldsymbol{c}_i^t \mid i \in \mathcal{N}(p)\}\}.
$$

Therefore, two iterations of phantom node enhanced MPNNs can capture one layer of 1-$n$-CWL-GNNs for arbitrary large $n$. $\square$

## J.2  PROOF OF PROPOSITION 18

**Proposition 18.** *MPNNs with phantom nodes of cliques no more than $k$ can 1-$k$ approximate $p(\mathbf{z}_i \mid \mathbf{X})$.*

*Proof.* The proof is equivalent to the proof of Proposition 17, except that the phantom nodes at most corresponds to $k$-cliques thus are corresponded to 1-$(k-1)$-CWL-GNNs:

$$
\begin{aligned}
\boldsymbol{c}_i^{t+1} &= f\left(\boldsymbol{c}_i^t, \{\{(\boldsymbol{c}_j^t \mid j \in \mathcal{N}_1(i)\}\}, ..., \{\{(\boldsymbol{c}_{\mathbf{j}}^t \mid \mathbf{j} \in \mathcal{N}_{k-1}(i)\}\}\right) \\
&= f_0\left(\boldsymbol{c}_i^t, \{\{(\boldsymbol{c}_{\mathbf{j}}^t \mid \mathbf{j} \in \mathcal{N}_1(i) \cup ... \cup \mathcal{N}_{k-1}(i)\}\}\right) \\
&= f_1\left(\boldsymbol{c}_i^t, \{\{(\boldsymbol{c}_{\mathbf{j}}^t \mid (\mathbf{j} \cup \{i\}) \in \mathcal{C}^k\}\}\right) \\
&= f_2\left(\boldsymbol{c}_i^t, \{\{(\boldsymbol{c}_p^t \mid p \in \mathcal{N}(i)\}\}\right).
\end{aligned}
$$

$\square$

## J.3  PROOF OF PROPOSITION 19

**Proposition 19.** *MPNNs with phantom edges 1-2 approximates $p(\mathbf{z}_u, \mathbf{z}_v)$ given by the modified Bethe free energy with altered distribution in Eq. 2.*

*Proof.* Given a MRF

$$
p(\mathbf{Z} \mid \mathbf{X}) = \frac{1}{Z} \prod_{i \in \mathcal{V}} \Psi_i(\mathbf{z}_i, \mathbf{x}_i) \prod_{(i,j) \in \mathcal{E}} \Psi_{ij}(\mathbf{z}_i, \mathbf{z}_j)
$$

and the corresponding altered distribution for Bethe free energy

$$
q(\mathbf{Z}) = \prod_{i \in \mathcal{V}} q_i(\mathbf{z}_i) \prod_{(i,j) \in \mathcal{E}} \frac{q_{ij}(\mathbf{z}_i, \mathbf{z}_j)}{q_i(\mathbf{z}_i) q_j(\mathbf{z}_j)} \prod_{(u,v) \in \hat{\mathcal{E}}} \frac{q_{uv}(\mathbf{z}_u, \mathbf{z}_v)}{q_u(\mathbf{z}_u) q_v(\mathbf{z}_v)}, \tag{9}
$$

we aim to

$$
\min_q D_{\mathrm{KL}}(q \| p).
$$

We can write an altered MRF as

$$p(\mathbf{Z} \mid \mathbf{X}) = \frac{1}{Z} \prod_{i \in \mathcal{V}} \Psi_i(\mathbf{z}_i, \mathbf{x}_i) \prod_{(i,j) \in \mathcal{E}} \Psi_{ij}(\mathbf{z}_i, \mathbf{z}_j) \prod_{(u,v) \in \hat{\mathcal{E}}} \Phi(\mathbf{z}_u, \mathbf{z}_v),$$

where $\Phi(\cdot, \cdot) \equiv 1$, thus this formulation is equivalent with the original one. Applying Bethe approximation on this formulation naturally yields the quasi distribution in Eq. 9. Obviously, by adding phantom edges for each $(u, v) \in \hat{\mathcal{E}}$ to the original $G$, the augmented graph $\hat{G}$ can naturally produce the altered MRF with

$$\mathcal{A}(i, \hat{G}) = \Psi_i,$$

$$\mathcal{A}(\{i, j\}, \hat{G}) = \begin{cases} \Psi_{ij}, & (i, j) \in \mathcal{E}, \\ \Phi, & (i, j) \in \hat{\mathcal{E}}. \end{cases}$$

Since this $\mathcal{A}$ is still within 1-WL's expressive power, MPNNs with phantom edges can 1-2 approximate the modified Bethe approximation. $\square$

## K   THE EXPRESSIVENESS OF GNNS WITHOUT THE WL HIERARCHY

### K.1   A REDEFINITION OF THE METRICS

In our previous discussions, we constrain the expressiveness of the function $\mathcal{A}$ for producing potentials in MRFs by the $k$-WL hierarchy. In this section we relax this constraint and provide a more fine-grained description of GNN variants without the $k$-WL constraints of $\mathcal{A}$. In contrast, we restrict the expressiveness of $\mathcal{A}$ to be dependent on the GNN models $\mathcal{M}$ we are to evaluate. Since obviously $\mathcal{M}$ won't be able to capture $\mathcal{A}$ if $\mathcal{A}$ is more expressive than $\mathcal{M}$, we restrict that $\mathcal{A}$ is equally or less expressive than $\mathcal{M}$, which further leads to the following redefinition of our metrics of expressiveness.

Our discussion now begins with the following definition of expressiveness.

**Definition 25.** A class of GNN models $\mathcal{M}$ can $l$-approximate some posteriori $p$ if and only if given arbitrary $\mathcal{A}$ satisfying:

- The maximum order of $\mathcal{A}$ is $l$.

- For arbitrary $G = (\boldsymbol{A}_G, \boldsymbol{X}_G)$, $H = (\boldsymbol{A}_H, \boldsymbol{X}_H)$ and $C_1, C_2$ being two nodes or cliques with no more than $l$ nodes from $G, H$ respectively, $\mathcal{A}(C_1, \boldsymbol{A}_G) \neq \mathcal{A}(C_2, \boldsymbol{A}_H)$ only when there exists a GNN instance from $\mathcal{M}$ that distinguishes $C_1$ and $C_2$.

There exists an instance of $\mathcal{M}$ such that:

- It can distinguish all graphs distinguished by iterations of belief propagation as in Theorem 3.

- It can provide marginals at least as accurate as Bethe approximation as in Theorem 4.

Clearly, the above definition relaxes the $k$-WL constraint of $\mathcal{A}$ in Definition 6 by restricting $\mathcal{A}$ to *share the equivalent expressiveness with the GNN models*. As we shall see, the results of the expressiveness of GNN models in Section 4 still holds.

### K.2   RESULTS W.R.T. NODE-LEVEL EXPRESSIVENESS

**Proposition 26.** *MPNNs can at most 2-approximate $p(\mathbf{z}_i \mid \mathbf{X})$ for arbitrary $G$ and $i \in \mathcal{V}_G$.*

**Proposition 27.** *$k$-GNNs can $k$-approximate $p(\mathbf{z}_i \mid \mathbf{X})$ for arbitrary $G$.*

**Proposition 28.** *ESANs with node marking policy can 3-approximate $p(\mathbf{z}_i \mid \mathbf{X})$ for arbitrary $G$.*

**Proposition 29.** *The GNN variants corresponding with SWL / CWL with $k$-clique simplex can $k$-approximate $p(\mathbf{z}_i \mid \mathbf{X})$.*

**Proposition 30.** *ID-GNNs (You et al., 2021) and Nested GNNs (Zhang & Li, 2021) can at most 2-approximate $p(\mathbf{z}_i \mid \mathbf{X})$.*

### K.3 PROOFS

First we note that the only difference between the above propositions and the main results in this paper is that $\mathcal{A}$ is now is limited by the GNN models themselves rather than the $k$-WL tests. In the previous proof in Section H.1 and Section I.1 it is seen that the potential functions $\Psi_a$ over factors $a$, when restricted by the $k$-WL tests, can be regarded as functions over the $k$-WL colors $\Psi_a = f(\mathrm{Col}^k(a))$, where $\mathrm{Col}^k(\cdot)$ indicates the $k$-WL color. We can then extend Theorem 21 to proof the above propositions.

**Theorem 31.** *Suppose the expressiveness of $\mathcal{A}$ is limited by some coloring function $\mathcal{M}$. We define generalized $l$-CWL, whose only difference between $k$-$l$-CWL is that $l$-CWL first apply $\mathcal{M}$ to the graph and records the colors. Then, $l$-CWL-GNNs can $(l+1)$-approximate $p(\mathbf{z}_i \mid \mathbf{X})$.*

*Proof.* The proof is the same except that we replace all $\mathrm{Col}^k(\cdot)$ emerged in the proof in Section H.1 with $\mathcal{M}(\cdot)$. □

With Theorem 31, the additional results in Section K.2 are proved with exactly the say steps in Section H, except that we replace the $k$-$l$-CWL in Section H with the $l$-CWL in Theorem 31 and set $\mathcal{M}$ to be the GNN models (i.e. MPNNs, $k$-GNNs, ESANs, etc.) in each proposition.

