# OpenReview forum: "Rethinking and Extending the Probabilistic Inference Capacity of GNNs"
_ICLR.cc/2024/Conference — ICLR 2024 poster_

### Official Review · Reviewer_G6RK · 2023-10-24

**Soundness:** 3 good
**Presentation:** 3 good
**Contribution:** 2 fair
**Rating:** 5
**Confidence:** 3

**Summary:**

The paper studies the expressiveness of different GNN variants from a probabilistic inference perspective. Specifically, under certain settings of PGMs and different ways to measure their complexity, the paper studies to what extent MPNNs can approximate node and edge marginals which are obtained from local minima of Bethe approximation. The analysis is also extended to other GNN variants. To equip GNNs with the power of estimating more complex node and edge marginals, phantom nodes and edges are proposed respectively, and experiments are conducted on several node and link prediction datasets to evaluate the proposed methods.

**Strengths:**

1. Analyzing how well GNNs can approximate marginals in PGMs is interesting (though there are limitations as will be discussed), and offers a new perspective for measuring the expressiveness of GNNs.
2. The paper is generally well written with clear background introductions and problem setup. Many details are given in the appendix, which is also appreciated.
2. A lot of analysis has been done with discussions on many different GNN variants and inference tasks. Results also seem solid (but I did not carefully check the proof).

**Weaknesses:**

1. Defining the discriminative power of potential function by WL test appears to be a somewhat contrived setup, whereby the problem still boils down to graph isomorphism. I am not sure how practically relevant this setting is, and whether it is truly “fundamentally different” with works on WL tests. A more comprehensive discussion about the connection may mitigate the issue.
2. Following the above point, the analysis does not take into account node features, whereas in many PGMs node and clique potentials are functions on node features. This further stresses the point that the setup might deviate from practice.
3. The proposed method seems not scalable as it requires identifying maximum cliques that is exponentially complex, and even relaxing it to cliques with size k has super-quadratic complexity O(n^k) (please correct me if I am wrong). This limitation has not been discussed in the paper, and experiments only include small datasets.
4. The experimental results are relatively weak. E.g. on PPI, best performance in most cases (4 out of 6 columns in the table) is achieved by GCNII+SPN. Improvement on synthetic dataset is also limited, and Planetoid dataset is outdated. Baselines are also limited. Particularly, many GNN variants are considered in the analysis but are not compared in experiments, and I wonder why? Moreover, the proposed methods are essentially based on data augmentation, but competitors in this category are missing.

Typos: invalid reference in appendix A and E.2; “a successful ? for minimizing” in 3.2; repeated references Bergen et al. and Cai
at al.

**Questions:**

1.  In 3.1, while lemma 1 is straight-forward, I do not fully understand why the authors mention “in graph machine learning fields each instance corresponds to different graphs with possibly different structures”, as the paper does not address the case of graph classification? And what does “instance” present? (the authors might wanna refer to a graph as an instance, but in the context of paper where node- and edge-level tasks are considered, each node or random variable in the PGM correspond an an instance.)
2. What is the complexity of the proposed method, and are there any ways to scale it to large graphs?
3. If possible, please also answer to questions in the weakness section.

---

I decrease score from 6 to 5 due to my lingering concerns, and there has been no response from the authors to date.

---

> ### Author Response · Authors · 2023-11-22
> **Response to the Review Comments**
>
> We thank the reviewer for the constructive comments! We have revised the manuscript to address the concerns. We are sorry for the late response.
>
> ## Question 1
>
> `In 3.1, while lemma 1 is straight-forward, I do not fully understand why the authors mention “in graph machine learning fields each instance corresponds to different graphs with possibly different structures”, as the paper does not address the case of graph classification? And what does “instance” present?`
>
> Our description follows the setting in [1] (Section 3), where each instance refers to a graph. The aim of this sentence is to point out the different assumptions in PGMs and graph machine learning. Traditionally, each individual instance in a graphical model shares the same conditional independence structure, that is, if $G_1$ and $G_2$ are two instances, $G_1$ and $G_2$ must share the same structure. However, in graph machine learning fields things are different. For example, the graphs for training and test often have different structures. In this case we say each instance has distinct structure. Therefore, we propose lemma1 to address the difference between traditional PGMs and the assumptions in graph machine learning.
>
> ## Question 2
>
> `What is the complexity of the proposed method, and are there any ways to scale it to large graphs?`
>
> We have added discussion of the complexity in Appendix C. To summarize, in worst cases where the graphs are dense (suppose $N$ nodes and $O(N^2)$ edges), $k$-order PE takes $O(N^k)$ space and $O(kN^k)$ time ($k$-GNNs take $O(kN^{k+1}) time). For sparse graphs the major complexity comes from finding cliques. It's hard to theoretically estimate the complexity of finding cliques in sparse graphs, but many works (eg. [2]) have shown that there are empirically efficient algorithms for finding k-cliques, e.g. [2] can find  cliques with order less than 10 in graphs with 68K nodes in seconds.
>
> ## Weakness 1
>
> `Defining the discriminative power of potential function by WL test appears to be a somewhat contrived setup, whereby the problem still boils down to graph isomorphism. I am not sure how practically relevant this setting is, and whether it is truly “fundamentally different” with works on WL tests. A more comprehensive discussion about the connection may mitigate the issue.`
>
> The key point of our theory is to discuss whether various GNN models can capture the variational inference of PGMs, including:
> 1. Whether GNNs can simulate local extrema of Bethe approximation.
> 2. What distribution can GNNs capture.
>
> To answer the problem 2 we propose concept of permutation-invariant PGMs, which is described by a permutation-invariant function $\mathcal{A}$ introduced in section 3.1. Therefore, it's **necessary** to restrict the expressive power of $\mathcal{A}$: for example if $\mathcal{A}$ output different potential functions for all pairs of non-isomorphic graphs, no GNNs can capture it.
>
> Since it's necessary to restrict the expressive power of $\mathcal{A}$, we choose two metrics: the order of $\mathcal{A}$ and the most common k-WL tests for restricting the expressive power. However, the WL tests play different roles in our and previous works. Previous works focus on the relation between GNNs and WL tests, while in our work WL tests in fact only serve as a restriction of the space of the joint distribution. Thus we believe our perspective is fundamentally different with previous works.
>
> In fact, many works such as [1] simply assumed PGMs as $p(X,Z)=\frac{1}{Z}\prod_{i}\Phi(x_i,z_i)\prod_{(i,j)\in\mathcal{E}}\Psi(z_i,z_j)$ where the potential functions are exactly the same for different nodes and edges, thus they are special situations of our assumptions. In other words, if it's valuable to discuss whether GNNs can capture local extrema of the above PGMs, so is our work.
>
> ## Weakness 2
>
> `Following the above point, the analysis does not take into account node features, whereas in many PGMs node and clique potentials are functions on node features. This further stresses the point that the setup might deviate from practice.`
>
> In our initial settings we believe regarding node features as observed random variables would provide a clear theoretical framework. Still, our results still hold when node and clique potentials are functions on node features. Recall that in Section 3.1 we introduce the permutation-invariant function $\mathcal{A}$ that describes PGMs, where $\mathcal{A}$ is a function on cliques and adjacency matrices $\mathcal{A}(C,\mathbf{A})$. If we let $\mathcal{A}$ further be a function on node features, i.e. $\mathcal{A}(C,G)$ where $G=(\mathbf{A},\mathbf{X})$ and $\mathbf{X}$ is the node feature matrix, the results still hold because GNN layers are also functions on $G$. Therefore, our results also hold when node and clique potentials are functions on node features.

---

> ### Author Response · Authors · 2023-11-22
> **Response to the Review Comments, part 2**
>
> ## Weakness 3
>
> `The proposed method seems not scalable as it requires identifying maximum cliques that is exponentially complex, and even relaxing it to cliques with size k has super-quadratic complexity O(n^k) (please correct me if I am wrong). This limitation has not been discussed in the paper, and experiments only include small datasets.`
>
> The maximum clique problem is a NP problem, thus is not scalable for large graphs. We have added discussions in Appendix C. In worst cases (where graphs are dense), relaxing it to k-cliques still has super-quadratic complexity. However, if we consider sparse graphs, there are empirically efficient algorithms for finding k-cliques (e.g. [2]). Although it's hard to theoretically estimate the complexity, the method in [2] can find 10-cliques in graphs with 68K nodes in seconds. Besides, even in worst cases our method is still more efficient than k-GNNs.
>
> ## Weakness 4
>
> `The experimental results are relatively weak. E.g. on PPI, best performance in most cases (4 out of 6 columns in the table) is achieved by GCNII+SPN. Improvement on synthetic dataset is also limited, and Planetoid dataset is outdated. Baselines are also limited. Particularly, many GNN variants are considered in the analysis but are not compared in experiments, and I wonder why? Moreover, the proposed methods are essentially based on data augmentation, but competitors in this category are missing.`
>
> We hypothesis three reasoning why our method hardly works better than SPN for GCNII.
>
> 1. SPN uses a node GNN to model node potentials and an edge GNN to model edge potentials. This framework might better aligns with the natural behavior of molecules.
>
> 2. We tried our best but we couldn't reproduce the results of baseline GNNs using the hyperparameters in [2]. Therefore although our methods can consistently improve the performance of GNNs, for some tasks it might not be better than SPN.
>
> 3. Better expressive power might not always lead to better empirical performance (e.g., [3]). Our framework focuses on theoretically improving the expressiveness of GNNs, rather than designing an empirically effective GNN architecture. We only apply the most simple and intuitive design.
>
> We did not include all GNN variants in the analysis because our methods focus on node classification and link prediction methods and aim to improve the expressiveness of MPNNs. Thus we mainly considers comparisons with MPNNs and other methods in the PGM-GNN literature. We are trying our best to perform additional experiments on real-world datasets and the LDPC dataset, and will update the manuscript once we finished the experiments. We are trying our best to to perform additional experiments on real-world datasets. We have performed experiments on the DBLP datasets in [4] as follows. As we can see, GAT+PN achieves the best results, while GCNII+PN achieves better results than SPN-GCNII.
>
> | GCN  | GCN+PN | GraphSage | GraphSage+PN |
> |:--------| :---------:|:---------:|:---------:|
> | 76.60 | 84.43 | 73.81 | 83.67 |
>
> | GAT | SPN-GAT | GAT+PN | GCNII | SPN-GCNII | GCNII+PN |
> |:---------:|:---------:|:---------:|:---------:|:---------:|:---------:|
> | 79.16 | 84.84 | 86.34 | 81.79 | 83.57 | 85.08 |
>
>
> [1] Dai et al. Discriminative Embeddings of Latent Variable Models for Structured Data. ICML 2016.
>
> [2] Lukas Gianinazzi et al. Parallel Algorithms for Finding Large Cliques in Sparse Graphs. In Proceedings of the 33rd ACM Symposium on Parallelism in Algorithms and Architectures.
>
> [3] Morris et al. Weisfeiler and Leman Go Neural: Higher-order Graph Neural Networks. AAAI 2019.
>
> [4] Qu et al. NEURAL STRUCTURED PREDICTION FOR INDUCTIVE NODE CLASSIFICATION. ICLR 2022.

---

> ### Author Response · Authors · 2023-11-23
> **Additional comments about the discriminative power of potential function**
>
> We thank the reviewer for the constructive comments! Here's our additional comments about why we restrict the discriminative power of potential function.
>
> First, we argue that it is **necessary** to restrict the discriminative power of potential function. The reason is that, given some GNN model $\mathcal{G}$, if the discriminative power of potential function is higher than $\mathcal{G}$, then clearly $\mathcal{G}$ cannot capture such potential function (we can simply let $\mathcal{A}$ to output different values only for graphs that are undistinguished by $\mathcal{G}$ but are distinguished by the potential functions). Therefore, it is necessary to restrict the discriminative power of potential function.
>
> Second, the reason why we chose the k-WL hierarchy is that it is the most common metric in GNN literature. However, this is not necessary. In fact, if you check the proof for the expressiveness of the GNN variants in the paper, the discriminative power of potential function is restricted only because we want the GNN variants to be able to distinguish **nodes that share different potential functions**. In other words, for the results (Proposition 9-Proposition 17), the discriminative power of potential function are not necessary restricted by k-WL. We can safely rewrite Proposition 9-Proposition 17 in the following manner:
>
> - (Original) Some GNN model $\mathcal{G}$ can $k$-$l$ approximate $p$
> - (Equivalent to) Some GNN model $\mathcal{G}$ can approximate $p$ given by $l$-order MRFs, whose discriminative power of potential function is limited by k-WL
> - (Rewrited version) Some GNN model $\mathcal{G}$ can approximate $p$ given by $l$-order MRFs, whose discriminative power of potential function is limited by $\mathcal{G}$
>
> The proof steps still holds for the rewrited version, so we can discard the limitations of $k$-WL in this manner. As shown in the proofs ( and the above rewrite), the key problem is not the discriminative power of potential function, but whether and how can GNNs simulate the local extrema of Bethe approximation and approximate node, edge and clique marginals.
>
> To summarize,
> - It is necessary to restrict the expressiveness of potential functions because triditional MRFs are not permutation invariant.
> - We can choose different metrics for the expressiveness, which are not limited to the k-WL hierarchy.

---

### Official Review · Reviewer_G9eq · 2023-10-31

**Soundness:** 3 good
**Presentation:** 3 good
**Contribution:** 2 fair
**Rating:** 6
**Confidence:** 3

**Summary:**

This paper delves into the expressive power of Graph Neural Networks (GNNs) within the context of approximate inference in graphical models, going beyond the common association of GNNs with graph convolutions and Weisfeiler-Lehman (WL) graph isomorphism tests. The paper challenges the prevailing notion that GNNs necessitate integration with graphical models to enhance their probabilistic inference capacity, asserting that GNNs intrinsically possess robust approximation capabilities for posterior distributions. The research introduces a new expressive power hierarchy using Markov Random Fields (MRFs) with increasingly complex distributions and inference targets, providing insights into various GNN variants, including MPNNs, higher-order GNNs, subgraph GNNs, and labeling tricks in the contexts of node classification and link prediction. Furthermore, the paper presents a systematic framework for extending the capabilities of GNNs in modeling complex distributions and inference targets, with a particular focus on phantom nodes and phantom edges, showcasing their empirical improvements in real-world applications of GNNs.

**Strengths:**

* Theoretical analysis is solid and reasonable on the inference capacity of various GNNs.
* The paper identifies that GNNs can do Probabilistic inference on its own, which has novelty than other methods combining GNN with graphical models.
* The paper proposes methods like phantom nodes and edges based on their inference analysis framework and show experiments of their methods.

**Weaknesses:**

* Experiments on large link prediction datasets are lacked.
* While the theoretical analysis has included a lot of GNN methods, the experiments don't include them all.
* Experiments on various node classification datasets(including homogeneous and heterogeneous graphs) are lacked.
* Improvements of better GNNs(like GCNII) in node classification is limited.
* Experiments details are lacked, like how to choose hyperparameters.

**Questions:**

1. For link prediction tasks, how is the experiments done? Why don't choose metric of link prediction to be HITS@10 like in the BUDDY paper. Also, why not include other baselines in the BUDDY paper, especially the GNN ones? Also, maybe lack of other experiments on large link prediction datsets
2. For node classification task, why not choose traditional homogeneous datasets and heterogeneous datasets like previous works. How will phantom nodes have impact when meeting graphs with different heterophily? Also, why For GCNII it hardly work better than SPN?
3. What's the complexity of GNNs with phantom nodes or phantom edges, for node classification and link prediction ,respectively?

---

> ### Author Response · Authors · 2023-11-22
> **Response to the Review Comments**
>
> We thank the reviewer for the constructive comments! We have revised the manuscript to address the concerns.
>
> ## Questions 1
>
> `For link prediction tasks, how is the experiments done?`
>
> The experimental setup follows [1]. For all datasets, at training time
> the message passing links are equal to the supervision links, while at test and validation time, disjoint sets of links are held out for supervision that are never seen at training time. We randomly generate 70-10-20 percent train-val-test splits.
>
> `Why don't choose metric of link prediction to be HITS@10 like in the BUDDY paper.`
>
> To make the experiments more challenging we choose H@10 as our metrics. If we set the metrics to be H@100 we can see that the results are much close:
>
> |  | Cora  |	Citeseer |
> |:--------| :---------:|--------:|
> | GCN | 0.853 | 0.892 |
> | GCN+PE | 0.883 | 0.907 |
> | GCNII | 0.868 | 0.911 |
> | GCNII + PE | 0.872 | 0.932 |
> | ELPH | 0.877 | 0.934 |
> | BUDDY | 0.880 | 0.929 |
>
> `Also, why not include other baselines in the BUDDY paper, especially the GNN ones? Also, maybe lack of other experiments on large link prediction datsets`
>
> We are sorry we did not test all baselines and did not conduct experiments on other datasets. Since this work mainly focus on theoretically analyse the probabilistic inference capacity of GNNs, our prior target is to ensure the theoretical results in this paper sound and clear. Nevertheless, we will keep trying our best to enhance the experiment sections in this manuscript.
>
> ## Questions 2
>
> `For node classification task, why not choose traditional homogeneous datasets and heterogeneous datasets like previous works.`
>
> For node classification task we follow the setting in [2]. The PPI datasets in [2] have 121 distinct labels thus suit node property prediction tasks well, and have been shown to benefit well from PGMs. Therefore, we choose them for node classification tasks.
>
> `How will phantom nodes have impact when meeting graphs with different heterophily?`
>
> The phantom nodes play as ensemble for cliques, enabling MPNNs to directly model closely connected nodes as an ensemble. The influence is especially significant when the input graphs are made up of several closely connected clusters.
>
> `Also, why For GCNII it hardly work better than SPN?`
>
> We hypothesis three reasoning why our method hardly works better than SPN for GCNII.
>
> 1. SPN uses a node GNN to model node potentials and an edge GNN to model edge potentials. This framework might better aligns with the natural behavior of molecules.
>
> 2. We tried our best but we couldn't reproduce the results of baseline GNNs in [2]. Therefore although our methods can consistently improve the performance of GNNs, for some tasks it might not be better than SPN.
>
> 3. Better expressive power might not always lead to better empirical performance[3]. Our framework focuses on theoretically improving the expressiveness of GNNs, rather than designing an empirically effective GNN architecture. We only apply the most simple and intuitive design.
>
> ## Questions 3
>
> `What's the complexity of GNNs with phantom nodes or phantom edges, for node classification and link prediction, respectively?`
>
> We have added discussions of complexity in Appendix C. Please refer to the paragraphs "Algorithm complexities" in Appendix C (colored blue) for detailed discussion.
>
> ## Weakness
>
> We are trying our best to perform additional experiments on real-world datasets and the LDPC dataset, and will update the manuscript once we finished the experiments. We did not include all GNN variants in the analysis because our methods focus on node classification and link prediction methods and aim to improve the expressiveness of MPNNs. Thus we mainly considers comparisons with MPNNs and other methods in the PGM-GNN literature.
>
> For hyperparameters, for node classification tasks we select the same hyperparameters in [1],[2] for fair comparision. When applying our phantom nodes / edges methods we keep the hyperparameters (learning rate, hidden dimension, etc.) unchanged. For link prediction tasks the hyperparameters are selected by performing search on validation data. Similarly, once we have chosen the hyperparameters of baseline MPNNs, we use the same hyperparameters for our PE-enhanced methods. We have conducted additional experiments on DBLP datasets in [2]. The results are as follows.
>
> | GCN  | GCN+PN | GraphSage | GraphSage+PN |
> |:--------| :---------:|:---------:|:---------:|
> | 76.60 | 84.43 | 73.81 | 83.67 |
>
> | GAT | SPN-GAT | GAT+PN | GCNII | SPN-GCNII | GCNII+PN |
> |:---------:|:---------:|:---------:|:---------:|:---------:|:---------:|
> | 79.16 | 84.84 | 86.34 | 81.79 | 83.57 | 85.08 |
>
>
> [1] Chamberlain et al. Graph Neural Networks for Link Prediction with Subgraph
> Sketching. ICLR 2023.
>
> [2] Qu et al. NEURAL STRUCTURED PREDICTION FOR INDUCTIVE NODE CLASSIFICATION. ICLR 2022.
>
> [3] Morris et al. Weisfeiler and Leman Go Neural: Higher-order Graph Neural Networks. AAAI 2019.

---

> > ### Comment · Reviewer_G9eq · 2023-11-23
> > **Response to the Authors**
> >
> > I appreciate the authors' responses and they address my concerns. I'll raise my score to 6.

---

> > > ### Author Response · Authors · 2023-11-23
> > > **Thanks for your response**
> > >
> > > We sincerely thank you for your response. We will keep improving the manuscript and conduct more experiments on various real-world datasets. Please let us know if you have further questions.

---

### Official Review · Reviewer_jjG1 · 2023-11-04

**Soundness:** 3 good
**Presentation:** 3 good
**Contribution:** 3 good
**Rating:** 8
**Confidence:** 3

**Summary:**

The paper studies the expressivity of the Graph neural networks from the perspective of probabilistic graphical methods. Although multiple works have previously established connections between GNNs and PGMs, this paper studies it from the perspective of expressive power of the GNNs. After formulating the correspondence between WL power with PGMs, the authors present several results in providing a new perspective of seeing the expressivity of GNNs in its ability to learn the complex higher-order distributions formalized as clique based MRFs.

**Strengths:**

1. The GNN’s connection to the PGMs is developed in an interesting and principled way.. Specifically, the formulation overcomes the permutation invariance inherent to the graph neural networks but missing in the PGMs.
1. The results progressively elaborate on the connections between GNN’s expressive power in terms of the learnable capacity of node marginals.
1. Some of the results are surprising although many of them are intuitively known. The correspondence between k-wl and clique orders is intuitive and is
1. The paper is well written and the presentation flow is reasonably clear.

**Weaknesses:**

1. The first paragraph is problematic. “implicitly assume that node representations learnt by GNNs are independent conditioned on node features and edges, thereby ignoring the joint dependency among nodes...” does not represent the related works accurately. These works do not ignore dependency among the nodes, which is captured via multiple rounds of message passing similar to loopy belief propagation. I find the first paragraph could be phrased in a different way to make the distinctions accurate.
1. The introduction of phantom nodes and edges is not a novel development and closely resembles the other methods like CIN. However, the problem of inefficiency remains in such methods i.e. the computational complexity of finding maximal cliques which can be used for phantom nodes to guarantee the inferential capacity.
1. Certain related works are missing in the paper. Recent works on Factor Graph Neural Networks (FGNN) [1] are highly related works in establishing connections between PGMs and GNNs. A small discussion on the relevance would be pertinent.
1. The experimental section could be more elaborate to study the effectiveness of GNNs in learning higher-order distributions. For example, comparison with Zhen et al. (2023)  on inference of higher-order LDPC codes.

**Overall,**
I find the paper making a good contribution to the understanding of the GNN's expressivity in a new perspective from the lens of Probabilistic graphical models. Although, I haven't carefully checked all the proofs, the results are mostly not surprising.

**References:**

 [1] Zhen Zhang, Mohammed Haroon Dupty, Fan Wu, Javen Qinfeng Shi, Wee Sun Lee. "Factor Graph Neural Networks" Journal of machine Learning research 24(181):1−54, 2023

**Questions:**

Please address the weaknesses

---

> ### Author Response · Authors · 2023-11-22
> **Response to the Review Comments**
>
> We thank the reviewer for the constructive comments! We have revised the manuscript to address the concerns.
>
> ## Weakness 1
>
> `The first paragraph is problematic. “implicitly assume that node representations learnt by GNNs are independent conditioned on node features and edges, thereby ignoring the joint dependency among nodes...” does not represent the related works accurately. These works do not ignore dependency among the nodes, which is captured via multiple rounds of message passing similar to loopy belief propagation. I find the first paragraph could be phrased in a different way to make the distinctions accurate.`
>
> Thanks for pointing them out. We have rewrited the sentence to make the presentation more accurate.
>
> ## Weakness 2
>
> `The introduction of phantom nodes and edges is not a novel development and closely resembles the other methods like CIN. However, the problem of inefficiency remains in such methods i.e. the computational complexity of finding maximal cliques which can be used for phantom nodes to guarantee the inferential capacity.`
>
> The introduction of phantom nodes does share spirits with CIN, that is to capture cliques in the graphs. We agree that finding cliques might be computational complex and this problem of inefficiency remains in our methods. But our method is more efficient than CIN: even if we only preserve the clique complex in CIN, CIN(CWL) needs to pass messages:
> - from nodes to nodes
> - between higher-order cliques and lower-order cliques (lower adjacent and higher adjacent)
> - between cliques of the same order (co-boundary cells)
> - between cliques and nodes (boundary cells)
>
> Our method, although share the same spirits, in fact point out the necessary component to capture higher-order dependencies and show that by adding messages between cliques and nodes in the cliques we can already extend MPNNs to capture higher-order PGMs without re-designing the WL scheme.
>
> We believe that the phantom edges are not similar to other methods like CIN. We focus on the link prediction scenarios, while CIN focuses on graph classification.
>
> ## Weakness 3
>
> `Certain related works are missing in the paper. Recent works on Factor Graph Neural Networks (FGNN) [1] are highly related works in establishing connections between PGMs and GNNs. A small discussion on the relevance would be pertinent.`
>
> We are sorry that we did not discuss recent works that combine PGMs and GNNs. We have added a paragraph in Appendix A (colored blue) to discuss these related works in the revised manuscript.
>
> ## Weakness 4
>
> `The experimental section could be more elaborate to study the effectiveness of GNNs in learning higher-order distributions. For example, comparison with Zhen et al. (2023) on inference of higher-order LDPC codes.`
>
> We are trying our best to perform additional experiments on more real-world datasets and the LDPC dataset, and will update the manuscript once we finished the experiments.

---

### Official Review · Reviewer_rfPW · 2023-11-05

**Soundness:** 3 good
**Presentation:** 3 good
**Contribution:** 3 good
**Rating:** 6
**Confidence:** 2

**Summary:**

The paper proposes a new approach to evaluate the expressive power of graph neural networks from a probabilistic perspective instead of Weisfeiler-Lehman (WL) tests, which are generally used for evaluating the expressiveness of GNNs. By introducing the central inference problems of probabilistic graphical models (PGMs), the authors analyze GNNs. In addition, the authors design two methods (phantom nodes and phantom edges) for the expressive power.

**Strengths:**

- The paper is well written.
- The research topic about the expressive power of GNNs is important and interesting.
- The paper seems novel to me. Different from existing methods on the expressive power of GNNs, which use WL test, this paper analyze the expressive power of GNNs in the perspective of the probabilistic view.

**Weaknesses:**

- Do you have any ideas for the graph classification tasks? Generally, the papers about the  expressiveness power of GNNs use graph classification tasks to demonstrate the effectiveness of their methods.
- Could you apply your methods on large-scale graphs?

**Questions:**

Please refer to the weaknesses.

---

> ### Author Response · Authors · 2023-11-22
> **Response to the Review Comments**
>
> We thank the reviewer for the constructive comments! We have revised the manuscript to address the concerns.
>
> ## Weakness 1
> `Do you have any ideas for the graph classification tasks? Generally, the papers about the expressiveness power of GNNs use graph classification tasks to demonstrate the effectiveness of their methods.`
>
> The inference of MRFs naturally results in node and edge marginals, which corresponds to node classification and link prediction tasks. To discuss graph-level expressive power from a probabilistic perspective, we need to first extend the MRFs to model the distribution of the whole graph. This can be done by adding an invented node that's connected to the rest of the nodes in the graph. For example, given a graph $G$, we manually add a node $v_0$ to the graph and connect it to the rest of the nodes. Suppose the graph classification task is modeled by $p(z_{v_0}~|~X)$, and GNNs compute the graph representation by aggregating all node representations. Then the results in Section 4.2.1 still holds for connected graphs.
>
> The proof is as follows. Recall that the two 1-WL variants with aggregation $c^{t+1}_u=hash(c^t_u,\{c^t_v ~ | ~ v\in N(u)\})$ and $c^{t+1}_u=hash(c^t_u,\{c^t_v ~ | ~ v\in N(u)\},\{c^t_v ~ | ~ v\notin N(u)\})$ share the same expressiveness for connected graphs. Thus by adding a node $v_0$ connecting to all nodes in $G$ does not influence the expressiveness of MPNNs (messages over $v_0$ can be seen as the aggregation of all nodes). Therefore, the representation of $v_0$ shares the same discriminative power with the aggregation of all nodes in $G$. Therefore, we have: a GNN variant can capture $k$-order MRFs in node level $\Rightarrow$ a GNN variant can capture MPNNs with $k$-order phantom nodes in node level $\Rightarrow$ a GNN variant can capture MPNNs with $k$-order phantom nodes in graph level by aggregating all nodes $\Rightarrow$ a GNN variant can capture $k$-order MRFs in graph level when the graph is connected.
>
> ## Weakness 2
> `Could you apply your methods on large-scale graphs?`
>
> By restricting the maximum order of cliques to be $k$ a fixed value, our methods can apply on large-scale graphs. We are trying our best to perform additional experiments on real-world datasets and the LDPC dataset, and will update the manuscript once we finished the experiments.
>
> We have conducted experiments on DBLP dataset, and the results are as follows.
>
> | GCN  | GCN+PN | GraphSage | GraphSage+PN |
> |:--------| :---------:|:---------:|:---------:|
> | 76.60 | 84.43 | 73.81 | 83.67 |
>
> | GAT | SPN-GAT | GAT+PN | GCNII | SPN-GCNII | GCNII+PN |
> |:---------:|:---------:|:---------:|:---------:|:---------:|:---------:|
> | 79.16 | 84.84 | 86.34 | 81.79 | 83.57 | 85.08 |

---

### Official Review · Reviewer_ubyE · 2023-11-08

**Soundness:** 3 good
**Presentation:** 2 fair
**Contribution:** 3 good
**Rating:** 8
**Confidence:** 2

**Summary:**

The authors report a slate of new theoretical results regarding the expressive power of a variety of GNNs w.r.t. MRFs. Utilizing this theoretical basis, they then propose a novel extension of MPNNs utilizing so-called "phantom nodes/edges". These proposed algorithms are tested using synthetic and real world data, comparing against state-of-the-art algorithms.

In node classification tasks, the phantom node lifting method improved state of the art when applied on GCN, and compared well to recent works when utilized with SAGE and GCNII methodologies. In link prediction, phantom edges proved to improve all compared results using standard datasets and metrics.

In summary, recasting GNN's expressibility via MRF, the authors were able to improve the understanding of MPNNs and used this probabilistic viewpoint to introduce a novel methodology that matches or outperforms the current state-of-the-art in standard tests.

**Strengths:**

The paper makes two contributions: (1) improving the theoretical understanding of GNNs utilizing a probabilistic viewpoint to classify expressibility with respect to (and beyond) Weisfeiler-Lehman tests and (2) utilizing this probabilistic viewpoint to propose a novel methodology to lift the expressive power of MPNNs.

Recasting the question of expressiveness from limited WL tests into a probabilistic frame is natural in the setting of ML, and as shown, lucrative.

**Weaknesses:**

While the results are compelling - I find the exposition of the novel phantom node/edge methodology lacking. While proofs are given on their approximation power, and one can assume motivation from framed previous works, the paper would be strengthened with the motivation driving the methodology.

Further, after the long theoretical exposition, the analysis of the proposed method is brief, and restricted mainly to results. A longer discussion of the modified properties (and potential limitations) of the graph would allow more informed adoption.

**Questions:**

1. Question: It seems that DropGNN (perhaps applied to a slightly modified graph) is closely related to the proposed PN method. How do the two compare in inference tasks?

2. Suggestion: I find the paper would be strengthened by a more direct discussion of the origins of the methodology (i.e., how does this differ/extend from previous works? E.g. node labeling)

---

> ### Author Response · Authors · 2023-11-22
> **Response to the Review Comments**
>
> We thank the reviewer for the constructive comments! We have revised the manuscript to address the concerns.
>
> ## Question 1
> `Question: It seems that DropGNN (perhaps applied to a slightly modified graph) is closely related to the proposed PN method. How do the two compare in inference tasks?`
>
> During each run, DropGNN randomly remove some nodes from the original graph. Although both methods aim to improve the expressiveness, we believe that DropGNN and our method do this in different manners. DropGNN improve the expressiveness by manually adding noise to the input graphs. Two non-isomorphic graphs that used to confuse MPNNs would now likely to behave differently because the structures induced by dropped nodes will now likely to be different. Thus, the motivation behind DropGNN is to obtain slightly perturbed variants of multi-hop neighborhoods of nodes. The phantom nodes instead serve as ensemble for cliques, enabling MPNNs to directly model closely connected nodes as an ensemble. The influence is especially significant when the input graphs are made up of several closely connected clusters. Our method is deterministic, while DropGNN is stochastic. Since even when applied on the same input, DropGNN might produce different results, we cannot ensure that DropGNN can capture (or shares the same distingshing power) with marginals of $k$-order PGMs.
>
> ## Question 2
> `Suggestion: I find the paper would be strengthened by a more direct discussion of the origins of the methodology (i.e., how does this differ/extend from previous works? E.g. node labeling)`
>
> We have added the discussion in Appendix C.2 (colored in blue). To summarize our method and the partial node labeling can be seen as methods that extend the labeling trick from two different perspective, each with its own advantages and disadvantages.
>
> ## Weakness
>
> We are sorry that due to the space limits of the initial manuscript, we weren't able to provide more motivation and analysis of the two proposed methods. We have added more discission of our methods including algorithm complexities, etc. in Appendix C (colored in blue).

---

### Meta-Review · Area_Chair_b61g · 2023-12-05

**Metareview:**

The paper studies the expressivity of the GNN from the perspective of probabilistic graphical methods. Within this framework the authors show how to systematically extend the capabilities of GNNs in modeling complex distributions and inference targets, with a particular focus on phantom nodes and phantom edges, showcasing their empirical improvements in real-world applications of GNNs.

The reviewers agree that the new perspective is interesting. Although, some of the reviewers raised concerns about the computational complexity and the lack of scalability of the methodology, which in return limits the experiments to small datasets.

The response of the authors during the rebuttal period addressed several comments made by the reviewers.

Given the interesting new perspective and compelling results, I recommend it for publication.

**Justification For Why Not Higher Score:**

Although the perspective is new, due to the lack of scalability it is only applicable to small graphs. Thus, it is not applicable to a large community.

**Justification For Why Not Lower Score:**

The paper presents a new perspective, which the reviewers agree that it is novel and interesting.

---

### Decision · Program_Chairs · 2024-01-16

Accept (poster)